# DEMONSTRATION-REGULARIZED RL

**Daniil Tiapkin**[1,2]    **Denis Belomestny**[3,2]    **Daniele Calandriello**[4]    **Éric Moulines**[1,5]
**Remi Munos**[4]    **Alexey Naumov**[2]    **Pierre Perrault**[6]    **Michal Valko**[4]    **Pierre Ménard**[7]
[1]CMAP, École Polytechnique    [2]HSE University    [3]Duisburg-Essen University
[4]Google DeepMind    [5]Mohamed Bin Zayed University of AI, UAE    [6]IDEMIA    [7]ENS Lyon
`{daniil.tiapkin,eric.moulines}@polytechnique.edu`
`denis.belomestny@uni-due.de`   `{dcalandriello,munos,valkom}@google.com`
`anaumov@hse.ru`   `pierre.perrault@outlook.com`   `pierre.menard@ens-lyon.fr`

## ABSTRACT

Incorporating expert demonstrations has empirically helped to improve the sample efficiency of reinforcement learning (RL). This paper quantifies theoretically to what extent this extra information reduces RL's sample complexity. In particular, we study the demonstration-regularized reinforcement learning that leverages the expert demonstrations by KL-regularization for a policy learned by behavior cloning. Our findings reveal that using $N^{\mathrm{E}}$ expert demonstrations enables the identification of an optimal policy at a sample complexity of order $\widetilde{\mathcal{O}}(\mathrm{Poly}(S, A, H)/(\varepsilon^2 N^{\mathrm{E}}))$ in finite and $\widetilde{\mathcal{O}}(\mathrm{Poly}(d, H)/(\varepsilon^2 N^{\mathrm{E}}))$ in linear Markov decision processes, where $\varepsilon$ is the target precision, $H$ the horizon, $A$ the number of action, $S$ the number of states in the finite case and $d$ the dimension of the feature space in the linear case. As a by-product, we provide tight convergence guarantees for the behavior cloning procedure under general assumptions on the policy classes. Additionally, we establish that demonstration-regularized methods are provably efficient for reinforcement learning from human feedback (RLHF). In this respect, we provide theoretical evidence showing the benefits of KL-regularization for RLHF in tabular and linear MDPs. Interestingly, we avoid pessimism injection by employing computationally feasible regularization to handle reward estimation uncertainty, thus setting our approach apart from the prior works.

## 1 INTRODUCTION

In reinforcement learning (RL, Sutton & Barto 1998), agents interact with an environment to maximize the cumulative reward they collect. While RL has shown remarkable success in mastering complex games (Mnih et al., 2013; Silver et al., 2018; Berner et al., 2019), controlling physical systems (Degrave et al., 2022), and enhancing computer science algorithms (Mankowitz et al., 2023), it does face several challenges. In particular, RL algorithms suffer from a large sample complexity, which is a hindrance in scenarios where simulations are impractical and struggle in environments with sparse rewards (Goecks et al., 2020).

A remedy found to handle these limitations is to incorporate information from a pre-collected offline dataset in the learning process. Specifically, leveraging demonstrations from experts—trajectories without rewards—has proven highly effective in reducing sample complexity, especially in fields like robotics (Zhu et al., 2018; Nair et al., 2020) and guiding exploration (Nair et al., 2018; Aytar et al., 2018; Goecks et al., 2020).

However, from a theoretical perspective, little is known about the impact of this approach. Previous research has often focused on either offline RL (Rashidinejad et al., 2021; Xie et al., 2021; Yin et al., 2021; Shi et al., 2022) or online RL (Jaksch et al., 2010; Azar et al., 2017; Fruit et al., 2018; Dann et al., 2017; Zanette & Brunskill, 2019b; Jin et al., 2018). In this study, we aim to quantify how prior demonstrations from experts influence the sample complexity of various RL tasks, specifically two scenarios: best policy identification (BPI, Domingues et al., 2021a; Al Marjani et al., 2021) and reinforcement learning from human feedback (RLHF), within the context of finite or linear Markov decision processes (Jin et al., 2020).

**Imitation learning** The case where the agent only observes expert demonstrations without further interaction with the environment corresponds to the well-known imitation learning problem. There

are two primary approaches in this setting: *inverse reinforcement learning* (Ng & Russell, 2000; Abbeel & Ng, 2004; Ho & Ermon, 2016) where the agent first infers a reward from demonstrations then finds an optimal policy for this reward; and *behavior cloning* (Pomerleau, 1988; Ross & Bagnell, 2010; Ross et al., 2011; Rajeswaran et al., 2018), a simpler method that employs supervised learning to imitate the expert. However collecting demonstration could be expansive and, furthermore, imitation learning suffers from the compounding errors effect, where the agent can diverge from the expert's policy in unvisited states (Ross & Bagnell, 2010; Rajaraman et al., 2020). Hence, imitation learning is often combined with an online learning phase where the agent directly interacts with the environment.

**BPI with demonstrations** In BPI with demonstrations, the agent observes expert demonstrations like in imitation learning but also has the opportunity to collect new trajectories, including reward information, by directly interacting with the environment. There are three main method categories[1] for BPI with demonstration: one employs an off-policy algorithm augmented with a supervised learning loss and a replay buffer pre-filled the demonstrations (Hosu & Rebedea, 2016; Lakshminarayanan et al., 2016; Vecerík et al., 2017; Hester et al., 2018); while a second uses reinforcement learning with a modified reward supplemented by auxiliary rewards obtained by inverse reinforcement learning (Zhu et al., 2018; Kang et al., 2018). The third class, demonstration-regularized RL, which is the one we study in this paper, leverages behavior cloning to learn a policy that imitates the expert and then applies reinforcement learning with regularization toward this behavior cloning policy (Rajeswaran et al., 2018; Nair et al., 2018; Goecks et al., 2020; Pertsch et al., 2021).

**Demonstration-regularized RL** We introduce a particular demonstration-regularized RL method that consists of several steps. We start by learning with maximum likelihood estimation from the demonstrations of a behavior policy. This transfers the prior information from the demonstrations to a more practical representation: the behavior cloning policy. During the online phase, we aim to solve a trajectory Kullback-Leibler divergence regularized MDP (Neu et al., 2017; Vieillard et al., 2020; Tiapkin et al., 2023), penalizing the policy for deviating too far from the behavior cloning policy. We use the solution of this regularized MDP as an estimate for the optimal policy in the unregularized MDP, effectively reducing BPI with demonstrations to regularized BPI.

Consequently, we propose two new algorithms for BPI in regularized MDPs: The `UCBVI-Ent+` algorithm, a variant of the `UCBVI-Ent` algorithm by Tiapkin et al. (2023) with improved sample complexity, and the `LSVI-UCB-Ent` algorithm, its adaptation to the linear setting. When incorporated into the demonstration-regularized RL method, these algorithms yield sample complexity rates for BPI with $N^{\mathrm{E}}$ demonstrations of order[2] $\widetilde{\mathcal{O}}(\mathrm{Poly}(S, A, H)/(\varepsilon^2 N^{\mathrm{E}}))$ in finite and $\widetilde{\mathcal{O}}(\mathrm{Poly}(d, H)/(\varepsilon^2 N^{\mathrm{E}}))$ in linear MDPs, where $\varepsilon$ is the target precision, $H$ the horizon, $A$ the number of action, $S$ the number of states in the finite case and $d$ the dimension of the feature space in the linear case. Notably, these rates show that leveraging demonstrations can significantly improve upon the rates of BPI without demonstrations, which are of order $\widetilde{\mathcal{O}}(\mathrm{Poly}(S, A, H)/\varepsilon^2)$ in finite MDPs (Kaufmann et al., 2021; Ménard et al., 2021) and $\widetilde{\mathcal{O}}(\mathrm{Poly}(d, H)/\varepsilon^2)$ in linear MDPs (Taupin et al., 2023). This work, up to our knowledge, represents the first instance of sample complexity rates for BPI with demonstrations, establishing the provable efficiency of demonstration-regularized RL.

**Preference-based BPI with demonstration** In RL with demonstrations, the assumption typically entails the observation of rewards in the online learning phase. However, in reinforcement learning from human feedback, such that recommendation system (Chaves et al., 2022), robotics (Jain et al., 2013; Christiano et al., 2017), clinical trials (Zhao et al., 2011) or large language models fine-tuning (Ziegler et al., 2019; Stiennon et al., 2020; Ouyang et al., 2022), the reward is implicitly defined by human values. Our focus centers on preference-based RL (PbRL, Busa-Fekete et al. 2014; Wirth et al. 2017; Novoseller et al. 2020; Saha et al. 2023) where the observed preferences between two trajectories are essentially noisy reflections of the value of a link function evaluated at the difference between cumulative rewards for these trajectories.

Existing literature on PbRL focuses either on the offline setting where the agent observes a pre-collected dataset of trajectories and preferences (Zhu et al., 2023; Zhan et al., 2023a) or on the

---

[1]The boundary between the above families of methods is not strict, since for example, one can see the regularization in the third family as a particular choice of auxiliary reward learned by inverse reinforcement learning that appears in the second class of methods.

[2]In the $\widetilde{\mathcal{O}}(\cdot)$ notation we ignore terms poly-log in $H, S, A, d, 1/\delta, 1/\varepsilon$ and the notation Poly indicates polynomial dependencies.

online setting where the agent sequentially samples a pair of trajectories and observes the associated preference (Saha et al., 2023; Xu et al., 2020; Wang et al., 2023).

In this work, we explore a hybrid setting that aligns more closely with what is done in practice (Ouyang et al., 2022). In this framework, which we call preference-based BPI with demonstration, the agent selects a sampling policy based on expert-provided demonstrations used to generate trajectories and associated preferences. The offline collection of preference holds particular appeal in RLHF due to the substantial cost and latency associated with obtaining preference feedback. Finally, in our setting, the agent engages with the environment by sequentially collecting *reward-free* trajectories and returns an estimate for the optimal policy.

**Demonstration-regularized RLHF** To address this novel setting, we follow a similar approach that was used in RL with demonstrations. We employ the dataset of preferences sampled using the behavior cloning policy to estimate rewards. Then, we solve the MDP regularized towards the behavior cloning policy, equipped with the estimated reward. Using the same regularized BPI solvers, we establish a sample complexity for the demonstration-regularized RLHF method of order $\widetilde{\mathcal{O}}((\mathrm{Poly}(S, A, H)/(\varepsilon^2 N^{\mathrm{E}}))$ in finite MDPs and $\widetilde{\mathcal{O}}((\mathrm{Poly}(d, H)/(\varepsilon^2 N^{\mathrm{E}}))$ in linear MDPs. Intriguingly, these rates mirror those of RL with demonstrations, illustrating that RLHF with demonstrations does not pose a greater challenge than RL with demonstrations. Notably, these findings expand upon the similar observation made by Wang et al. (2023) in the absence of prior information.

We highlight our main contributions:

- We establish that demonstration-regularized RL is an efficient solution method for RL with $N^{\mathrm{E}}$ demonstrations, exhibiting a sample complexity of order $\widetilde{\mathcal{O}}(\mathrm{Poly}(S, A, H)/(\varepsilon^2 N^{\mathrm{E}}))$ in finite MDPs and $\widetilde{\mathcal{O}}(\mathrm{Poly}(d, H)/(\varepsilon^2 N^{\mathrm{E}}))$ in linear MDPs.
- We provide evidence that demonstration-regularized methods can effectively address reinforcement learning from human feedback (RLHF) by collecting preferences offline and eliminating the necessity for pessimism (Zhan et al., 2023a). Interestingly, they achieve sample complexities similar to those in RL with demonstrations.
- We prove performance guarantees for the behavior cloning procedure in terms of Kullback-Leibler divergence from the expert policy. They are of order $\widetilde{\mathcal{O}}(\mathrm{Poly}(S, A, H)/N^{\mathrm{E}})$ for finite MDPs and $\widetilde{\mathcal{O}}(\mathrm{Poly}(d, H)/N^{\mathrm{E}})$ for linear MDPs.
- We provide novel algorithms for regularized BPI in finite and linear MDPs with sample complexities $\widetilde{\mathcal{O}}(H^5 S^2 A/(\lambda \varepsilon))$ and $\widetilde{\mathcal{O}}(H^5 d^2/(\lambda \varepsilon))$, correspondingly, where $\lambda$ is a regularization parameter.

## 2 SETTING

**MDPs** We consider an episodic MDP $\mathcal{M} = (\mathcal{S}, s_1, \mathcal{A}, H, \{p_h\}_{h \in [H]}, \{r_h\}_{h \in [H]})$, where $\mathcal{S}$ is the set of states with $s_1$ the fixed initial state, $\mathcal{A}$ is the finite set of actions of size $A$, $H$ is the number of steps in one episode, $p_h(s'|s, a)$ is the probability transition from state $s$ to state $s'$ by performing action $a$ in step $h$. And $r_h(s, a) \in [0, 1]$ is the reward obtained by taking action $a$ in state $s$ at step $h$.

We will consider two particular classes of MDPs.

**Definition 1.** (Finite MDP) An MDP $\mathcal{M}$ is *finite* if the state space $\mathcal{S}$ is finite with size denoted by $S$.

**Definition 2.** (Linear MDP) An MDP $\mathcal{M} = (\mathcal{S}, s_1, \mathcal{A}, H, \{p_h\}_{h \in [H]}, \{r_h\}_{h \in [H]})$ is *linear* if the state space $\mathcal{S}$ is a measurable for a certain $\sigma$-algebra $\mathcal{F}_{\mathcal{S}}$, and there exists known feature map $\psi: \mathcal{S} \times \mathcal{A} \to \mathbb{R}^d$, and *unknown* parameters $\theta_h \in \mathbb{R}^d$ and an *unknown* family of signed measure $\mu_{h,i}, h \in [H], i \in [d]$ with its vector form $\mu_h: \mathcal{F}_{\mathcal{S}} \to \mathbb{R}^d$ such that for all $(h, s, a) \in [H] \times \mathcal{S} \times \mathcal{A}$ and for any measurable set $B \in \mathcal{F}_{\mathcal{S}}$, it holds $r_h(s, a) = \psi(s, a)^{\mathsf{T}} \theta_h$, and $p_h(B|s, a) = \sum_{i=1}^d \psi(s, a)_i \mu_{h,i}(B) = \psi(s, a)^{\mathsf{T}} \mu_h(B)$. Without loss of generality, we assume $\|\psi(s, a)\|_2 \leq 1$ for all $(s, a) \in \mathcal{S} \times \mathcal{A}$ and $\max\{\|\mu_h(\mathcal{S})\|_2, \|\theta_h\|_2\} \leq \sqrt{d}$ for all $h \in [H]$.

**Policy & value functions** A policy $\pi$ is a collection of functions $\pi_h : \mathcal{S} \to \Delta_{\mathcal{A}}$ for all $h \in [H]$, where every $\pi_h$ maps each state to a probability over the action set. We denote by $\Pi$ the set of policies. The value functions of policy $\pi$ at step $h$ and state $s$ is denoted by $V_h^{\pi}$, and the optimal value functions, denoted by $V_h^{\star}$, are given by the Bellman respectively optimal Bellman equations

$$Q_h^{\star}(s, a) = r_h(s, a) + p_h V_{h+1}^{\star}(s, a) \qquad V_h^{\star}(s) = \max_a Q_h^{\star}(s, a)$$

where by definition, $V_{H+1}^{\star} \triangleq 0$. Furthermore, $p_h f(s, a) \triangleq \mathbb{E}_{s' \sim p_h(\cdot|s,a)}[f(s')]$ denotes the expectation operator with respect to the transition probabilities $p_h$ and $\pi_h g(s) \triangleq \mathbb{E}_{a \sim \pi_h(\cdot|s)}[g(s, a)]$ denotes the composition with the policy $\pi$ at step $h$.

**Trajectory Kullback-Leibler divergence** We define the trajectory Kullback-Leibler divergence between policy $\pi$ and policy $\pi'$ as the average of the Kullback-Leibler divergence between policies

at each step along a trajectory sampled with $\pi$,

$$\mathrm{KL}_{\mathrm{traj}}(\pi\|\pi') = \mathbb{E}_\pi\left[\sum_{h=1}^{H}\mathrm{KL}(\pi_h(s_h)\|\pi'_h(s_h))\right].$$

## 3 BEHAVIOR CLONING

In this section, we analyze the complexity of behavior cloning for imitation learning in finite and linear MDPs.

**Imitation learning** In imitation learning we are provided a dataset $\mathcal{D}_{\mathrm{E}} \triangleq \{\mathring{\tau}_i = (s_1^i, a_1^i, \ldots, s_H^i, a_H^i), i \in [N^{\mathrm{E}}]\}$ of $N^{\mathrm{E}}$ independent *reward-free* trajectories sampled from a fixed unknown expert policy $\pi^{\mathrm{E}}$. The objective is to learn from these demonstrations a policy close to optimal. In order to get useful demonstrations we assume that the expert policy is close to optimal, that is, $V_1^\star(s_1) - V_1^{\pi^{\mathrm{E}}}(s_1) \leq \varepsilon_{\mathrm{E}}$ for some small $\varepsilon_{\mathrm{E}} > 0$.

**Behavior cloning** The simplest method for imitation learning is to directly learn to replicate the expert policy in a supervised fashion. Precisely the behavior cloning policy $\pi^{\mathrm{BC}}$ is obtained by minimizing the negative-loglikelihood over a class of policies $\mathcal{F} = \{\pi \in \Pi : \pi_h \in \mathcal{F}_h\}$ with $\mathcal{F}_h$ being a class of conditional distributions $\mathcal{S} \to \mathcal{P}(\mathcal{A})$ and where $\mathcal{R}_h$ is some regularizer,

$$\pi^{\mathrm{BC}} \in \arg\min_{\pi\in\mathcal{F}} \sum_{h=1}^{H}\left(\sum_{i=1}^{N^{\mathrm{E}}}\log\frac{1}{\pi_h(a_h^i|s_h^i)} + \mathcal{R}_h(\pi_h)\right). \tag{1}$$

In order to provide convergence guarantees for behavior cloning, we make the following assumptions. First, we assume some regularity conditions on the class of policies defined in terms of covering numbers of the class, see Appendix A for a definition.

**Assumption 1.** For all $h \in [H]$, there are two positive constants $d_\mathcal{F}, R_\mathcal{F} > 0$ such that
$$\forall h \in [H], \forall \varepsilon \in (0,1) : \log\mathcal{N}(\varepsilon, \mathcal{F}_h, \|\cdot\|_\infty) \leq d_\mathcal{F}\log(R_\mathcal{F}/\varepsilon).$$

Moreover, there is a constant $\gamma > 0$ such that for any $h \in [H]$, $\pi_h \in \mathcal{F}_h$ it holds $\pi_h(a|s) \geq \gamma$ for any $(s,a) \in \mathcal{S} \times \mathcal{A}$.

The Assumption 1 is a typical parametric assumption in density estimation, see e.g. Zhang (2002), with $d_\mathcal{F}$ being a covering dimension of the underlying parameter space. The part of the assumption on a minimal probability is needed to control KL-divergences (Zhang, 2006).

Next, we assume that a smooth version of the expert policy belongs to the class of hypotheses.

**Assumption 2.** There is a constant $\kappa \in (0, 1/2)$ such that a $\kappa$-greedy version of the expert policy defined by $\pi_h^{\mathrm{E},\kappa}(a|s) = (1-\kappa)\pi_h^{\mathrm{E}}(a|s) + \kappa/A$ belongs to the hypothesis class of policies: $\pi^{\mathrm{E},\kappa} \in \mathcal{F}$.

Note that a deterministic expert policy verifies Assumption 2 provided that $\gamma$ is small enough and the policy class is rich enough. For $\kappa = 0$, this assumption is never satisfied for any $\gamma > 0$.

In the sequel, we provide examples of the policy class $\mathcal{F}$ and regularizers $(\mathcal{R}_h)_{h\in[H]}$ for finite or linear MDPs such that the above assumptions are satisfied. We are now ready to state general performance guarantees for behavior cloning with KL regularization.

**Theorem 1.** *Let Assumptions 1-2 be satisfied and let $0 \leq \mathcal{R}_h(\pi_h) \leq M$ for all $h \in [H]$ and any policy $\pi \in \mathcal{F}_h$. Then with probability at least $1 - \delta$, the behavior policy $\pi^{\mathrm{BC}}$ satisfies*

$$\mathrm{KL}_{\mathrm{traj}}(\pi^{\mathrm{E}}\|\pi^{\mathrm{BC}}) \leq \frac{6d_\mathcal{F}H \cdot (\log(Ae^3/(A\gamma \wedge \kappa))) \cdot \log(2HN^{\mathrm{E}}R_\mathcal{F}/(\gamma\delta))}{N^{\mathrm{E}}} + \frac{2HM}{N^{\mathrm{E}}} + \frac{18\kappa}{1-\kappa}.$$

This result shows that if the number of demonstrations $N^{\mathrm{E}}$ is large enough and $\gamma = 1/N^{\mathrm{E}}$, $\kappa = A/N^{\mathrm{E}}$ then the behavior cloning policy $\pi^{\mathrm{BC}}$ converges to the expert policy $\pi^{\mathrm{E}}$ at a fast rate of order $\widetilde{\mathcal{O}}((d_\mathcal{F}H + A)/N^{\mathrm{E}})$ where we measure the "distance" between two policies by the trajectory Kullback-Leibler divergence. The proof of this theorem is postponed to Appendix B and it heavily relies on verifying the so-called Bernstein condition (Bartlett & Mendelson, 2006).

### 3.1 FINITE MDPS

For finite MDPs, we chose a logarithmic regularizer $\mathcal{R}_h(\pi_h) = \sum_{s,a}\log(1/\pi_h(a|s))$ and the class of policies $\mathcal{F} = \{\pi \in \Pi : \pi_h(a|s) \geq 1/(N^{\mathrm{E}} + A)\}$. One can check that Assumptions 1-2 hold and $0 \leq \mathcal{R}_h(\pi_h) \leq SA\log(N^{\mathrm{E}} + A)$. We can apply Theorem 1 to obtain the following bound for finite MDPs (see Appendix B.2 for additional details).

**Corollary 1.** *For all $N^{\mathrm{E}} \geq A$, for function class $\mathcal{F}$ and regularizer $(\mathcal{R}_h)_{h\in[H]}$ defined above, it holds with probability at least $1 - \delta$,*

$$\mathrm{KL}_{\mathrm{traj}}(\pi^{\mathrm{E}}\|\pi^{\mathrm{BC}}) \leq \frac{6SAH \cdot \log(2e^4N^{\mathrm{E}}) \cdot \log(12H(N^{\mathrm{E}})^2/\delta)}{N^{\mathrm{E}}} + \frac{18AH}{N^{\mathrm{E}}}.$$

Note that, imitation learning with a logarithmic regularizer is closely related to the statistical problem of conditional density estimation with Kullback-Leibler divergence loss, see for example Section 4.3 by van der Hoeven et al. (2023) and references therein. Additionally, we would like to emphasize that the presented upper bound is optimal up to poly-logarithmic terms, see Appendix B.5 for a corresponding lower bound.

**Remark 1.** In fact, the constraint added by the class of policies $\mathcal{F}$ is redundant with the effect of regularization and one can directly optimize over the whole set of policies in (1). It is then easy to obtain a closed formula for the behavior cloning policy $\pi_h^{\mathrm{BC}}(a|s) = (N_h^{\mathrm{E}}(s,a) + 1)/(N_h^{\mathrm{E}}(s) + A)$, where we define the counts by $N_h^{\mathrm{E}}(s) = \sum_{a \in \mathcal{A}} N_h^{\mathrm{E}}(s,a)$ and $N_h^{\mathrm{E}}(s,a) = \sum_{i=1}^{N^{\mathrm{E}}} \mathbb{1}\{(s_h^i, a_h^i) = (s,a)\}$.

**Remark 2.** Contrary to Ross & Bagnell (2010) and Rajaraman et al. (2020), our bound does not feature the optimality gap of the behavior policy but measures how close it is to the expert policy which is crucial to obtain the results of the next sections. Nevertheless, we can recover from our bound some of the results of the aforementioned references, see Appendix B.6 for details.

## 3.2 LINEAR MDPs

For the linear setting, we need the expert policy to belong to some well-behaved class of parametric policies. A first possibility would be to consider a greedy policy with respect to $Q$-value, linear in the feature space $\pi_h(s) \in \arg\max_{\pi \in \Delta_{\mathcal{A}}} (\pi\psi)(s)^{\mathsf{T}} w_h$ for some parameters $w_h$ as it is done in the existing imitation learning literature (Rajaraman et al., 2021). However, under such a parametrization it would be almost impossible to learn an expert policy with a high quality since a small perturbation in the parameters $w_h$ could lead to a completely different policy. We emphasize that if we assume that the expert policy is an optimal one, then Rajaraman et al. (2021) proposes a way to achieve a $\varepsilon$-optimal policy but not how to reconstruct the expert policy itself. That is why we consider another natural parametrization where the log probability of the expert policy is linear in the feature space.

**Assumption 3.** For all $h \in [H]$, there exists an *unknown* parameter $w_h^{\mathrm{E}} \in \mathbb{R}^d$ with $\|w_h^{\mathrm{E}}\|_2 \leq R$ for some known $R \geq 0$ such that $\pi_h^{\mathrm{E}}(a|s) = \exp(\psi(s,a)^{\mathsf{T}} w_h^{\mathrm{E}})/(\sum_{a' \in \mathcal{A}} \exp(\psi(s,a')^{\mathsf{T}} w_h^{\mathrm{E}}))$.

For instance, this assumption is satisfied for optimal policy in entropy-regularized linear MDPs, see Lemma 1 in Appendix B.3. Under Assumption 3, a suitable choice of policy class is given by $\mathcal{F} = \{\pi \in \Pi : \pi_h \in \mathcal{F}_h\}$ where

$$\mathcal{F}_h = \left\{ \pi_h(a|s) = \frac{\kappa}{A} + (1-\kappa)\frac{\exp(\psi(s,a)^{\mathsf{T}} w_h)}{\sum_{a' \in \mathcal{A}} \exp(\psi(s,a')^{\mathsf{T}} w_h)} : w_h \in \mathbb{R}^d, \|w_h\|_2 \leq R \right\} \quad (2)$$

and $\kappa = A/(N^{\mathrm{E}} + A)$. Furthermore, for the linear setting, we do not need regularization, that is, $\mathcal{R}_h(\pi) = 0$. Equipped with this class of policies we can prove a similar result as in the finite setting with the number of states replaced by the dimension $d$ of the feature space.

**Corollary 2.** *Under Assumption 3, function class $\mathcal{F}$ defined above and regularizer $\mathcal{R}_h = 0$ for all $h \in [H]$, it holds for all $N^{\mathrm{E}} \geq A$ with probability at least $1 - \delta$,*

$$\mathrm{KL}_{\mathrm{traj}}(\pi^{\mathrm{E}} \| \pi^{\mathrm{BC}}) \leq \frac{8dH \cdot (\log(2\mathrm{e}^3 A N^{\mathrm{E}}) \cdot (\log(48(N^{\mathrm{E}})^2 R) + \log(H/\delta)))}{N^{\mathrm{E}}} + \frac{18AH}{N^{\mathrm{E}}}.$$

Taking into account the fact that finite MDPs are a specific case within the broader category of linear MDPs, the lower bound presented in Appendix B.5 also shows the optimality of this result.

## 4 DEMONSTRATION-REGULARIZED RL

In this section, we study reinforcement learning when demonstrations from an expert are also available. First, we describe the regularized best policy identification framework that will be useful later.

**Regularized best policy identification (BPI)** Given some reference policy $\widetilde{\pi}$ and some regularization parameter $\lambda > 0$, we consider the trajectory Kullback-Leibler divergence regularized value function $V_{\widetilde{\pi},\lambda,1}^{\pi}(s_1) \triangleq V_1^{\pi}(s_1) - \lambda \mathrm{KL}_{\mathrm{traj}}(\pi, \widetilde{\pi})$. In this value function, the policy $\pi$ is penalized for moving too far from the reference policy $\widetilde{\pi}$. Interestingly, we can compute the value of policy $\pi$ with the regularized Bellman equations, (Neu et al., 2017; Vieillard et al., 2020)

$$Q_{\widetilde{\pi},\lambda,h}^{\pi}(s,a) = r_h(s,a) + p_h V_{\widetilde{\pi},\lambda,h+1}^{\pi}(s,a), \quad V_{\widetilde{\pi},\lambda,h}^{\pi}(s) = \pi_h Q_{\widetilde{\pi},\lambda,h}^{\pi}(s) - \lambda \mathrm{KL}(\pi_h(s) \| \widetilde{\pi}_h(s)),$$

where $V_{\widetilde{\pi},\lambda,H+1}^{\pi} = 0$. We are interested in the best policy identification for this regularized value. Precisely, in regularized BPI, the agent interacts with MDP as follows: at the beginning of episode $t$, the agent picks up a policy $\pi^t$ based only on the transitions collected up to episode $t - 1$. Then a new trajectory (with rewards) is sampled following the policy $\pi^t$ and is observed by the agent. At the end of each episode, the agent can decide to stop collecting new data, according to a random stopping time $\iota$ ($\iota = t$ if the agent stops after the $t$-th episode), and output a policy $\widehat{\pi}$ based on the observed transitions. An agent for regularized BPI is therefore made of a triplet $((\pi^t)_{t \in \mathbb{N}}, \iota, \widehat{\pi})$.

**Definition 3.** (PAC algorithm for regularized BPI) An algorithm $((\pi^t)_{t\in\mathbb{N}}, \iota, \widehat{\pi})$ is $(\varepsilon, \delta)$-PAC for BPI regularized with policy $\widetilde{\pi}$ and parameter $\lambda$ with sample complexity $\mathcal{C}(\varepsilon, \lambda, \delta)$ if

$$\mathbb{P}\Big(V_{\widetilde{\pi},\lambda,1}^{\star}(s_1) - V_{\widetilde{\pi},\lambda,1}^{\widehat{\pi}}(s_1) \leq \varepsilon, \quad \iota \leq \mathcal{C}(\varepsilon, \lambda, \delta)\Big) \geq 1 - \delta.$$

We can now describe the setting studied in this section.

**BPI with demonstration** We assume, as in Section 3, that first the agent observes $N^{\mathrm{E}}$ independent trajectories $\mathcal{D}_{\mathrm{E}}$ sampled from an expert policy $\pi^{\mathrm{E}}$. Then the setting is the same as in BPI. Precisely, the agent interacts with the MDP as follows: at episode $t$, the agent selects a policy $\pi^t$ based on *the collected transitions and the demonstrations*. Then a new trajectory (with rewards) is sampled following the policy $\pi^t$ and observed by the agent. At the end of each episode, the agent stops according to a stopping rule $\iota$ ($\iota = t$ if the agent stops after the $t$-th episode), and outputs a policy $\pi^{\mathrm{RL}}$.

**Definition 4.** (PAC algorithm for BPI with demonstration) An algorithm $((\pi^t)_{t\in\mathbb{N}}, \iota, \pi^{\mathrm{RL}})$ is $(\varepsilon, \delta)$-PAC for BPI with demonstration with sample complexity $\mathcal{C}(\varepsilon, N^{\mathrm{E}}, \delta)$ if

$$\mathbb{P}\Big(V_1^{\star}(s_1) - V_1^{\pi^{\mathrm{RL}}}(s_1) \leq \varepsilon, \quad \iota \leq \mathcal{C}(\varepsilon, N^{\mathrm{E}}, \delta)\Big) \geq 1 - \delta.$$

To tackle BPI with demonstration we focus on the following natural and simple approach.

**Demonstration-regularized RL** The main idea behind this method is to reduce BPI with demonstration to regularized BPI. Indeed, in demonstration-regularized RL, the agent starts by learning through behavior cloning from the demonstration of a policy $\pi^{\mathrm{BC}}$ that imitates the expert policy, refer to Section 3 for details. Then the agent computes a policy $\pi^{\mathrm{RL}}$ by performing regularized BPI with policy $\pi^{\mathrm{BC}}$ and some well-chosen parameter $\lambda$. The policy $\pi^{\mathrm{RL}}$ is then returned as the guess for an optimal policy. The whole procedure is described in Algorithm 1. Intuitively the prior information contained in the demonstration is compressed into a handful representation namely the policy $\pi^{\mathrm{BC}}$. Then this information is injected into the BPI procedure by encouraging the agent to output a policy close to the behavior policy.

---

**Algorithm 1** Demonstration-regularized RL

1: **Input:** Precision parameter $\varepsilon_{\mathrm{RL}}$, probability parameter $\delta_{\mathrm{RL}}$, demonstrations $\mathcal{D}_{\mathrm{E}}$, regularization parameter $\lambda$.
2: Compute behavior cloning policy $\pi^{\mathrm{BC}} = \mathtt{BehaviorCloning}(\mathcal{D}_{\mathrm{E}})$.
3: Perform regularized BPI $\pi^{\mathrm{RL}} = \mathtt{RegBPI}(\pi^{\mathrm{BC}}, \lambda, \varepsilon_{\mathrm{RL}}, \delta_{\mathrm{RL}})$
4: **Output:** policy $\pi^{\mathrm{RL}}$.

---

Using the previous results for regularized BPI, we next derive guarantees for demonstration-regularized RL. We start from a general black-box result that shows how the final policy error depends on the behavior cloning error, parameter $\lambda$, and the quality of regularized BPI.

**Theorem 2.** *Assume that there are an expert policy $\pi^{\mathrm{E}}$ such that $V_1^{\star}(s_1) - V_1^{\pi^{\mathrm{E}}}(s_1) \leq \varepsilon_{\mathrm{E}}$ and a behavior cloning policy $\pi^{\mathrm{BC}}$ satisfying $\sqrt{\mathrm{KL}_{\mathrm{traj}}(\pi^{\mathrm{E}}\|\pi^{\mathrm{BC}})} \leq \varepsilon_{\mathrm{KL}}$. Let $\pi^{\mathrm{RL}}$ be $\varepsilon_{\mathrm{RL}}$-optimal policy in $\lambda$-regularized MDP with respect to $\pi^{\mathrm{BC}}$, that is, $V_{\pi^{\mathrm{BC}},\lambda,1}^{\star}(s_1) - V_{\pi^{\mathrm{BC}},\lambda,1}^{\pi^{\mathrm{RL}}} \leq \varepsilon_{\mathrm{RL}}$. Then $\pi^{\mathrm{RL}}$ fulfills*

$$V_1^{\star}(s_1) - V_1^{\pi^{\mathrm{RL}}}(s_1) \leq \varepsilon_{\mathrm{E}} + \varepsilon_{\mathrm{RL}} + \lambda\varepsilon_{\mathrm{KL}}^2.$$

*In particular, under the choice $\lambda^{\star} = \varepsilon_{\mathrm{RL}}/\varepsilon_{\mathrm{KL}}^2$, the policy $\pi^{\mathrm{RL}}$ is $(2\varepsilon_{\mathrm{RL}} + \varepsilon_{\mathrm{E}})$-optimal in the original (non-regularized) MDP.*

**Remark 3.** We define an error in trajectory KL-divergence under the square root because the KL-divergence behaves quadratically in terms of the total variation distance by Pinsker's inequality.

**Remark 4** (BPI with prior policy). We would like to underline that we exploit all the prior information only through one fixed behavior cloning policy. However, as it is observable from the bounds of Theorem 2, our guarantees are not restricted to such type of policies and potentially could work with any prior policy close enough to a near-optimal one in trajectory Kullback-Leibler divergence.

The proof of the theorem above is postponed to Appendix C. To apply this result and derive sample complexity for demonstration-regularized BPI, we present the `UCBVI-Ent+` algorithm, a modification of the algorithm `UCBVI-Ent` proposed by Tiapkin et al. (2023), that achieves better rates for regularized BPI in the finite setting. In Appendix E, we also present the `LSVI-UCB-Ent` algorithm, a direct adaptation of the `UCBVI-Ent+` to the linear setting.

Notably, the use of `UCBVI-Ent` by Tiapkin et al. (2023) within the framework of demonstration-regularized methods, fails to yield acceleration through expert data incorporation due to its $\widetilde{\mathcal{O}}(1/\varepsilon^2)$ sample complexity. In contrast, the enhanced variant, `UCBVI-Ent+`, exhibits a more favorable complexity of $\widetilde{\mathcal{O}}(1/(\varepsilon\lambda))$, where $\lambda = \varepsilon/\varepsilon_{\mathrm{KL}}^2 \gg \varepsilon$ under the conditions stipulated in Theorem 2, for a $\varepsilon_{\mathrm{KL}}$ sufficiently small. It is noteworthy that an alternative approach employing the `RL-Explore-Ent` algorithm, introduced by Tiapkin et al. (2023), can also achieve such acceleration. However, `RL-Explore-Ent` is associated with inferior rates in terms of $S$ and $H$ and is challenging to extend beyond finite settings.

The `UCBVI-Ent+` algorithm works by sampling trajectories according to an exploratory version of an optimistic solution for the regularized MDP which is characterized by the following rules.

`UCBVI-Ent+` **sampling rule** To obtain the sampling rule at episode $t$, we first compute a policy $\bar{\pi}^t$ by optimistic planning in the regularized MDP,

$$\overline{Q}_h^t(s,a) = \mathrm{clip}\Big(r_h(s,a) + \widehat{p}_h^t \overline{V}_{h+1}^t(s,a) + b_h^{p,t}(s,a), 0, H\Big),$$

$$\bar{\pi}_h^{t+1}(s) = \arg\max_{\pi \in \Delta_{\mathcal{A}}}\Big\{\pi \overline{Q}_h^t(s) - \lambda \mathrm{KL}(\pi\|\widetilde{\pi}_h(s))\Big\}, \quad \overline{V}_h^t(s) = \bar{\pi}_h^{t+1}\overline{Q}_h^t(s) - \lambda \mathrm{KL}(\bar{\pi}_h^{t+1}(s)\|\widetilde{\pi}_h(s))$$

with $\overline{V}_{H+1}^t = 0$ by convention, where $\widetilde{\pi}$ is a reference policy, $\widehat{p}^t$ is an estimate of the transition probabilities, and $b^t$ some bonus term taking into account estimation error for transition probabilities. Then we define a family of policies that aim to explore actions for which $Q$-value is not well estimated at a particular step. That is, for $h' \in [0, H]$, the policy $\pi^{t,(h')}$ first follows the optimistic policy $\bar{\pi}^t$ until step $h$ where it selects an action leading to the largest width of a confidence interval for the optimal $Q$-value,

$$\pi_h^{t,(h')}(a|s) = \begin{cases} \bar{\pi}_h^t(a|s) & \text{if } h \neq h', \\ \mathbb{1}\Big\{a = \arg\max_{a' \in \mathcal{A}}(\overline{Q}_h^t(s,a') - \underline{Q}_h^t(s,a'))\Big\} & \text{if } h = h', \end{cases}$$

where $\underline{Q}^t$ is a lower bound on the optimal regularized $Q$-value function, see Appendix D.4. In particular, for $h' = 0$ we have $\pi^{t,(0)} = \bar{\pi}^t$. The sampling rule is obtained by picking uniformly at random one policy among the family $\pi^t = \pi^{t,(h')}$, $h' \in [0, H]$ in each episode. Note that it is equivalent to sampling from a uniform mixture policy $\pi^{\mathrm{mix},t}$ over all $h' \in [0, H]$, see Appendix D for more details. This algorithmic choice allows us to exploit strong convexity of the KL-divergence and control the properties of a stopping rule, defined in Appendix D.4, that depends on the gap $(\overline{Q}_h^t(s,a) - \underline{Q}_h^t(s,a))^2$.

The complete procedure is described in Algorithm 3 in Appendix D. We prove that for the well-calibrated bonus functions $b^{p,t}$ and a stopping rule defined in Appendix D.4, the `UCBVI-Ent+` algorithm is $(\varepsilon, \delta)$-PAC for regularized BPI and provide a high-probability upper bound on its sample complexity. Additionally, a similar result holds for `LSVI-UCB-Ent` algorithm. The next result is proved in Appendix D.5 and Appendix E.5.

**Theorem 3.** *For all $\varepsilon > 0$, $\delta \in (0, 1)$, the `UCBVI-Ent+` / `LSVI-UCB-Ent` algorithms defined in Appendix D.4 /Appendix E.4 are $(\varepsilon, \delta)$-PAC for the regularized BPI with sample complexity*

$$\mathcal{C}(\varepsilon, \delta) = \widetilde{\mathcal{O}}\Big(\frac{H^5 S^2 A}{\lambda\varepsilon}\Big) \text{ (finite)} \qquad \mathcal{C}(\varepsilon, \delta) = \widetilde{\mathcal{O}}\Big(\frac{H^5 d^2}{\lambda\varepsilon}\Big) \text{ (linear)}.$$

*Additionally, assume that the expert policy is $\varepsilon_{\mathrm{E}} = \varepsilon/2$-optimal and satisfies Assumption 3 in the linear case. Let $\pi^{\mathrm{BC}}$ be the behavior cloning policy obtained using corresponding function sets described in Section 3. Then demonstration-regularized RL based on `UCBVI-Ent+`/`LSVI-UCB-Ent` with parameters $\varepsilon_{\mathrm{RL}} = \varepsilon/4$, $\delta_{\mathrm{RL}} = \delta/2$ and $\lambda = \widetilde{\mathcal{O}}\big(N^{\mathrm{E}}\varepsilon/(SAH)\big)/\widetilde{\mathcal{O}}\big(N^{\mathrm{E}}\varepsilon/(dH)\big)$ is $(\varepsilon, \delta)$-PAC for BPI with demonstration in finite / linear MDPs and has sample complexity of order*

$$\mathcal{C}(\varepsilon, N^{\mathrm{E}}, \delta) = \widetilde{\mathcal{O}}\Big(\frac{H^6 S^3 A^2}{N^{\mathrm{E}}\varepsilon^2}\Big) \text{ (finite)} \qquad \mathcal{C}(\varepsilon, N^{\mathrm{E}}, \delta) = \widetilde{\mathcal{O}}\Big(\frac{H^6 d^3}{N^{\mathrm{E}}\varepsilon^2}\Big) \text{ (linear)}.$$

In the finite setting, `UCBVI-Ent+` improves the previous fast-rate sample complexity result of order $\widetilde{\mathcal{O}}(H^8 S^4 A/(\lambda\varepsilon))$ by Tiapkin et al. (2023). For the linear setting, we would like to acknowledge that `LSVI-UCB-Ent` is the first algorithm that achieves fast rates for exploration in regularized linear MDPs.

## 5 DEMONSTRATION-REGULARIZED RLHF

In this section, we consider the problem of reinforcement learning with human feedback. We assume that the MDP is finite, i.e., $|\mathcal{S}| < +\infty$ to simplify the manipulations with the trajectory space.

However, the state space could be arbitrarily large. In this setting, we do not observe the true reward function $r^\star$ but have access to an oracle that provides a preference feedback between two trajectories. We assume that the preference is a random variable with parameters that depend on the cumulative rewards of the trajectories as detailed in Assumption 4. Given a reward function $r = \{r_h\}_{h=1}^H$, we define the reward of a trajectory $\tau \in (\mathcal{S} \times \mathcal{A})^H$ as the sum of rewards collected over this trajectory $r(\tau) \triangleq \sum_{h=1}^H r_h(s_h, a_h)$.

**Assumption 4** (Preference-based model). Let $\tau_0, \tau_1$ be two trajectories. The preference for $\tau_1$ over $\tau_0$ is a Bernoulli random variable $o$ with a parameter $q_\star(\tau_0, \tau_1) = \sigma(r^\star(\tau_1) - r^\star(\tau_0))$, where $\sigma \colon \mathbb{R} \to [0, 1]$ is a monotone increasing link function that satisfies $\inf_{x \in [-H, H]} \sigma'(x) = 1/\zeta$ for $\zeta > 0$.

The main example of the link function is a sigmoid function $\sigma(x) = 1/(1 + \exp(-x))$ that leads to the Bradley-Terry-Luce (BTL) model (Bradley & Terry, 1952) widely used in the literature (Wirth et al., 2017; Saha et al., 2023). We now introduce the learning framework.

**Preference-based BPI with demonstration** We assume, as in Section 3, that the agent observes $N^{\mathrm{E}}$ independent trajectories $\mathcal{D}_{\mathrm{E}}$ sampled from an expert policy $\pi^{\mathrm{E}}$. Then the learning is divided in two phases:

1) *Preference collection.* Based on the observed expert trajectories $\mathcal{D}_{\mathrm{E}}$, the agent selects a sampling policy $\pi^{\mathrm{S}}$ to generate a *data set of preferences* $\mathcal{D}_{\mathrm{RM}} = \{(\tau_0^k, \tau_1^k, o^k)\}_{k=1}^{N^{\mathrm{RM}}}$ consisting of pairs of trajectories and the sampled preferences. Specifically, both trajectories of the pair $(\tau_0^k, \tau_1^k)$ are sampled with the policy $\pi^{\mathrm{S}}$ and the associated preference $o^k$ is obtained according to the preference-based model described in Assumption 4.

2) *Reward-free interaction.* Next, the agent interacts with the reward-free MDP as follows: at episode $t$, the agent selects a policy $\pi^t$ based on *the collected transitions up to time $t$, demonstrations and preferences*. Then a new trajectory (reward-free) is sampled following the policy $\pi^t$ and is observed by the agent. At the end of each episode, the agent can decide to stop according to a stopping rule $\iota$ and outputs a policy $\pi^{\mathrm{RLHF}}$.

**Definition 5** (PAC algorithm for preference-based BPI with demonstration). An algorithm $((\pi^t)_{t \in \mathbb{N}}, \pi^{\mathrm{S}}, \iota, \pi^{\mathrm{RLHF}})$ is $(\varepsilon, \delta)$-PAC for preference-based BPI with demonstrations and sample complexity $\mathcal{C}(\varepsilon, N^{\mathrm{E}}, \delta)$ if $\mathbb{P}\left(V_1^\star(s_1) - V_1^{\pi^{\mathrm{RLHF}}}(s_1) \le \varepsilon, \ \iota \le \mathcal{C}(\varepsilon, N^{\mathrm{E}}, \delta)\right) \ge 1 - \delta$, where the unknown true reward function $r^\star$ is used in the value-function $V^\star$.

For the above setting, we provide a natural approach that combines demonstration-regularized RL with the maximum likelihood estimation of the reward given preferences dataset.

**Demonstration-regularized RLHF** During the preference collection phase, the agent generates a dataset comprising trajectories and observed preferences, denoted as $\mathcal{D}_{\mathrm{RM}} = \{(\tau_0^k, \tau_1^k, o^k)\}_{k=1}^{N^{\mathrm{RM}}}$ by executing the previously computed policy $\pi^{\mathrm{BC}}$. Using this dataset, the agent can infer the reward via maximum likelihood estimation (MLE).

---

**Algorithm 2** Demonstration-regularized RLHF

---

1: **Input:** Precision parameter $\varepsilon_{\mathrm{RLHF}}$, probability parameter $\delta_{\mathrm{RLHF}}$, demonstrations $\mathcal{D}_{\mathrm{E}}$, preferences budget $N^{\mathrm{RM}}$, regularization parameter $\lambda$.
2: Compute behavior cloning policy $\pi^{\mathrm{BC}} = \texttt{BehaviorCloning}(\mathcal{D}_{\mathrm{E}})$;
3: Select sampling policy $\pi^{\mathrm{S}} = \pi^{\mathrm{BC}}$ and collect preference dataset $\mathcal{D}_{\mathrm{RM}}$;
4: Compute reward estimate $\hat{r} = \texttt{RewardMLE}(\mathcal{G}_r, \mathcal{D}_{\mathrm{RM}})$;
5: Perform regularized BPI using $\hat{r}$ as reward: $\pi^{\mathrm{RLHF}} = \texttt{RegBPI}(\pi^{\mathrm{BC}}, \lambda, \varepsilon_{\mathrm{RLHF}}, \delta_{\mathrm{RLHF}}; \hat{r})$
6: **Output:** policy $\pi^{\mathrm{RLHF}}$.

---

The core idea behind this approach is to simplify the problem by transforming it into a regularized BPI problem. The agent starts with behavior cloning applied to the expert dataset, resulting in the policy $\pi^{\mathrm{BC}}$. During the preference collection phase, the agent generates a dataset comprising trajectories and observed preferences, denoted as $\mathcal{D}_{\mathrm{RM}} = \{(\tau_0^k, \tau_1^k, o^k)\}_{k=1}^{N^{\mathrm{RM}}}$ by executing the previously computed policy $\pi^{\mathrm{BC}}$. Using this dataset, the agent can infer the reward via MLE:

$$\hat{r} \triangleq \arg\max_{r \in \mathcal{G}} \sum_{k=1}^{N^{\mathrm{RM}}} o^k \log\left(\sigma\big(r(\tau_1^k) - r(\tau_0^k)\big)\right) + (1 - o^k) \log\left(1 - \sigma\big(r(\tau_1^k) - r(\tau_0^k)\big)\right),$$

where $\mathcal{G}$ is a function class for trajectory reward functions[3]. Finally, the agent computes $\pi^{\mathrm{RL}}$ by performing regularized BPI with policy $\pi^{\mathrm{BC}}$, a properly chosen regularization parameter $\lambda$ and the estimated reward $\hat{r}$. The complete procedure is outlined in Algorithm 2.

---

[3] For the theoretical guarantees on MLE estimate of rewards $\hat{r}$ we refer to Appendix F.1.

For this algorithm, we use the behavior cloning policy $\pi^{\mathrm{BC}}$ for two purposes. First, it allows efficient offline collection of the preference dataset $\mathcal{D}_{\mathrm{RM}}$, from which a high-quality estimate of the reward can be derived. Second, a regularization towards the behavior cloning policy $\pi^{\mathrm{BC}}$ enables the injection of information obtained from the demonstrations, while also avoiding the direct introduction of pessimism in the estimated reward as in the previous works that handle offline datasets (Zhu et al., 2023; Zhan et al., 2023a).

**Remark 5.** Zhan et al. (2023b) propose a similar two-stage setting of preference collection and reward-free interaction without prior demonstrations and propose an algorithm for this setup. However, as compared to their result, our pipeline is adapted to any parametric function approximation of rewards and does not require solving any (non-convex) optimization problem during the preference collection phase.

**Remark 6.** Our approach to solve BPI with demonstration within the preference-based model framework draws inspiration from well-established methods for large language model RL fine-tuning (Stiennon et al., 2020; Ouyang et al., 2022; Lee et al., 2023). Specifically, our algorithm's policy learning phase is similar to solving an RL problem with policy-dependent rewards

$$r_h^{\mathrm{RLHF}}(s,a) = \hat{r}_h(s,a) - \lambda \log\big(\pi_h^{\mathrm{RLHF}}(a|s)/\pi_h^{\mathrm{BC}}(a|s)\big).$$

This formulation, coupled with our prior stages of behavior cloning, akin to supervised fine-tuning (SFT), and reward estimation through MLE based on trajectories generated by the SFT policy, mirrors a simplified version of the three-phase RLHF pipeline.

The following sample complexity bounds for tabular and linear MDPs is a simple corollary of Theorem 8 and Theorem 3 and its proof is postponed to Appendix F.3.

**Corollary 3** (Demonstration-regularized RLHF). *Let Assumption 4 hold. For $\varepsilon > 0$ and $\delta \in (0,1)$, assume that an expert policy $\varepsilon_{\mathrm{E}}$ is $\varepsilon/8$-optimal and satisfies Assumption 3 in the linear case. Let $\pi^{\mathrm{BC}}$ be the behavioral cloning policy obtained using function sets described in Section 3 and let the set $\mathcal{G}$ be defined in Lemma 19 for finite and in Lemma 20 for linear setting, respectively.*

*If the following two conditions hold*

$$(1)\ N^{\mathrm{E}} \cdot N^{\mathrm{RM}} \geq \widetilde{\Omega}\big(\zeta^2 H^2 \widetilde{D}^2/\varepsilon^2\big); \quad (2)\ N^{\mathrm{E}} \geq \widetilde{\Omega}\big(H^2 \widetilde{D}/\varepsilon\big)\ or\ N^{\mathrm{RM}} \geq \widetilde{\Omega}\big(C_r \zeta^2 H \widetilde{D}/\varepsilon^2\big)$$

*for $\widetilde{D} = SA/d$ in finite/linear MDPs and $C_r = C_r(\mathcal{G}, \pi^{\mathrm{E}}, \pi^{\mathrm{BC}})$ is a single-policy concentrability coefficient defined in (20), Appendix F.3 (see also Zhan et al. 2023a), then demonstration-regularized RLHF based on* `UCBVI-Ent+`/`LSVI-UCB-Ent` *with parameters $\varepsilon_{\mathrm{RL}} = \varepsilon/16$, $\delta_{\mathrm{RL}} = \delta/3$ and $\lambda = \lambda^\star \geq \widetilde{\mathcal{O}}\big(N^{\mathrm{E}}\varepsilon/(SAH)\big)/\widetilde{\mathcal{O}}\big(N^{\mathrm{E}}\varepsilon/(dH)\big)$ is $(\varepsilon,\delta)$-PAC for BPI with demonstration in finite/linear MDPs with sample complexity*

$$\mathcal{C}(\varepsilon, N^{\mathrm{E}}, \delta) = \widetilde{\mathcal{O}}\left(\frac{H^6 S^3 A^2}{N^{\mathrm{E}}\varepsilon^2}\right)\ \text{(finite)} \qquad \mathcal{C}(\varepsilon, N^{\mathrm{E}}, \delta) = \widetilde{\mathcal{O}}\left(\frac{H^6 d^3}{N^{\mathrm{E}}\varepsilon^2}\right)\ \text{(linear)}.$$

The conditions (1) and (2) control two different terms in the reward estimation error presented in Theorem 8. Condition (1) shows that the small size of the expert dataset should be compensated by a larger dataset used for reward estimation and vice versa.

At the same time, condition (2) requires that at least one of these datasets is large enough to overcome the sub-exponential behavior of the error in the reward estimation problem. We remark that the second part of the condition (2) $N^{\mathrm{RM}} \geq C_r/\varepsilon^2$ is unavoidable in the general case of offline learning even if the transitions are known due to a lower bound in Theorem 3 by Zhan et al. (2023a). However, as soon as the reward estimation error is small enough, we obtain the same sample complexity guarantees as in the demonstration-regularized RL (see Section 4).

## 6 CONCLUSION

In this study, we introduced the BPI with demonstration framework and showed that demonstration-regularized RL, a widely employed technique, is not just practical but also theoretically efficient for this problem. Additionally, we proposed a novel preference-based BPI with demonstration approach, where the agent gathers demonstrations offline. Notably, we proved that a demonstration-regularized RL method can also solve this problem efficiently without explicit pessimism injection. A compelling direction for future research could involve expanding the feedback mechanism in the preference-based setting, transitioning from pairwise comparison to preference ranking (Zhu et al., 2023). Additionally, it would be interesting to explore scenarios where the assumption of a white-box preference-based model, as proposed by Wang et al. (2023), is relaxed.

## ACKNOWLEDGMENTS

D. Belomestny acknowledges the financial support from Deutsche Forschungsgemeinschaft (DFG), Grant Nr.497300407. The work of D. Belomestny and A. Naumov was supported by the grant for research centers in the field of AI provided by the Analytical Center for the Government of the Russian Federation (ACRF) in accordance with the agreement on the provision of subsidies (identifier of the agreement 000000D730321P5Q0002) and the agreement with HSE University No. 70-2021-00139. The work of D. Tiapkin has been supported by the Paris Île-de-France Région in the framework of DIM AI4IDF.

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

# Appendix

## Table of Contents

# A  NOTATION

Table 1: Table of notation use throughout the paper

| Notation | Meaning |
|---|---|
| $\mathcal{S}$ | state space of size $S$ |
| $\mathcal{A}$ | action space of size $A$ |
| $d$ | dimension of linear MDP |
| $H$ | length of one episode |
| $s_1$ | initial state |
| $\iota$ | stopping time |
| $\mathcal{T}$ | trajectory space, $\mathcal{T} \triangleq (\mathcal{S} \times \mathcal{A})^H$ |
| $\varepsilon$ | desired accuracy of solving the problem |
| $\delta$ | desired upper bound on failure probability |
| $p_h(s'|s,a)$ | probability transition |
| $r_h(s,a)$ | reward function |
| $V_h^\pi, V_h^\star$ | value of policy $\pi$ and optimal value |
| $Q_h^\pi, Q_h^\star$ | Q-value of policy $\pi$ and optimal Q-value |
| $V_{\widetilde{\pi},\lambda,h}^\pi, V_{\widetilde{\pi},\lambda,h}^\star$ | regularized value of policy $\pi$ and optimal regularized value |
| $Q_{\widetilde{\pi},\lambda,h}^\pi, Q_{\widetilde{\pi},\lambda,h}^\star$ | regularized Q-value of policy $\pi$ and optimal regularized Q-value |
| $\Pi, \Pi_h$ | space of all policies and space of policies on step $h$ |
| $\Pi_\gamma, \Pi_{h,\gamma}$ | space of all policies and policies on step $h$ with minimal probability $\gamma$ |
| $\mathcal{D}_{\mathrm{E}}$ | expert dataset of size $N^{\mathrm{E}}$: $\mathcal{D}_{\mathrm{E}} \triangleq \{\mathring{\tau}_i = (s_1^i, a_1^i, \ldots, s_H^i, a_H^i), i \in [N^{\mathrm{E}}]\}$ |
| $\pi^{\mathrm{E}}$ | expert policy |
| $\pi^{\mathrm{E},\kappa}$ | $\kappa$-greedy version of the expert policy |
| $\varepsilon_{\mathrm{E}}$ | sub-optimality gap of the expert policy: $V_1^\star(s_1) - V_1^{\pi^{\mathrm{E}}}(s_1) \leq \varepsilon_{\mathrm{E}}$ |
| $\pi^{\mathrm{BC}}$ | behavior cloning policy |
| $\mathcal{R}_h$ | regularizer for behavior cloning |
| $\mathcal{F}$ | class of policies for behavior cloning |
| $d_{\mathcal{F}}$ | covering dimension of one-step policy class for behavior cloning |
| $s_h^t$ | state that was visited at $h$ step during $t$ episode |
| $a_h^t$ | action that was picked at $h$ step during $t$ episode |
| $r^\star$ | true reward function in a preference-based model |
| $\sigma$ | link function, see Assumption 4 |
| $\zeta$ | linearity measure of link function, see Assumption 4 |
| $\pi^{\mathrm{S}}$ | sampling policy for generation preference dataset |
| $\mathcal{D}_{\mathrm{RM}}$ | preference dataset of size $N^{\mathrm{RM}}$: $\mathcal{D}_{\mathrm{RM}} \triangleq \{(\tau_0^k, \tau_1^k, o^k)\}$ |
| $C_r(\mathcal{G}, \pi^{\mathrm{E}}, \pi^{\mathrm{BC}})$ | single-policy concentrability coefficient, see (20) |
| $\mathcal{G}$ | class of trajectory rewards for reward modeling |
| $d_{\mathcal{G}}$ | bracketing dimension of the induced preference models |
| $\pi^{\mathrm{RL}}$ | policy for BPI with demonstration |
| $\pi^{\mathrm{RLHF}}$ | policy for preference-based BPI with demonstration |
| $\mathcal{C}(\varepsilon, N^{\mathrm{E}}, \delta)$ | sample complexity for BPI with demonstration |
| $\mathcal{C}(\varepsilon, \lambda, \delta)$ | sample complexity for regularized BPI |
| $\mathbb{R}_+$ | non-negative real numbers |
| $\mathbb{N}_+$ | positive natural numbers |
| $[n]$ | set $\{1, 2, \ldots, n\}$ |
| e | Euler's number |
| $\Delta_d$ | $d-1$-dimensional probability simplex: $\Delta_d \triangleq \{x \in \mathbb{R}_+^d : \sum_{j=1}^d x_j = 1\}$ |
| $\Delta_{\mathcal{X}}$ | set of distributions over a finite set $\mathcal{X}$ : $\Delta_{\mathcal{X}} = \Delta_{|\mathcal{X}|}$. |
| $\mathrm{clip}(x, m, M)$ | clipping procedure $\mathrm{clip}(x, m, M) \triangleq \max(\min(x, M), m)$ |

Let $(\mathsf{X}, \mathcal{X})$ be a measurable space and $\mathcal{P}(\mathsf{X})$ be the set of all probability measures on this space. For $p \in \mathcal{P}(\mathsf{X})$, we denote by $\mathbb{E}_p$ the expectation w.r.t. $p$. For a random mapping $\xi : \mathsf{X} \to \mathbb{R}$ notation $\xi \sim p$ means $\mathrm{Law}(\xi) = p$. For any measures $p, q \in \mathcal{P}(\mathsf{X})$, we denote their product measure by

$p \otimes q$. We also write $\mathbb{E}_{\xi \sim p}$ instead of $\mathbb{E}_p$. For any $p, q \in \mathcal{P}(\mathsf{X})$, the Kullback-Leibler divergence between $p$ and $q$ is given by

$$\mathrm{KL}(p, q) = \begin{cases} \mathbb{E}_p[\log \frac{\mathrm{d}p}{\mathrm{d}q}], & p \ll q, \\ +\infty, & \text{otherwise}. \end{cases}$$

For any $p \in \mathcal{P}(\mathsf{X})$ and $f: \mathsf{X} \to \mathbb{R}$, we denote $pf = \mathbb{E}_p[f]$. In particular, for any $p \in \Delta_d$ and $f: \{0, \ldots, d\} \to \mathbb{R}$, we use $pf = \sum_{\ell=0}^{d} f(\ell)p(\ell)$. Define $\mathrm{Var}_p(f) = \mathbb{E}_{s' \sim p}\big[(f(s') - pf)^2\big] = p[f^2] - (pf)^2$. For any $(s, a) \in \mathcal{S}$, transition kernel $p(s, a) \in \mathcal{P}(\mathcal{S})$ and $f: \mathcal{S} \to \mathbb{R}$, define $pf(s, a) = \mathbb{E}_{p(s,a)}[f]$ and $\mathrm{Var}_p[f](s, a) = \mathrm{Var}_{p(s,a)}[f]$. For any $s \in \mathcal{S}$, policy $\pi(s) \in \mathcal{P}(\mathcal{S})$ and $f: \mathcal{S} \times \mathcal{A} \to \mathbb{R}$, set $\pi f(s) = \mathbb{E}_{a \sim \pi(s)}[f(s, a)]$ and $\mathrm{Var}_\pi f(s) = \mathrm{Var}_{a \sim \pi(s)}[f(s, a)]$. For a MDP $\mathcal{M}$, a policy $\pi$ and a sequence of function $(f_h, h \in [H])$, define $\mathbb{E}_\pi[\sum_{h'=h}^{H} f(s_{h'}, a_{h'})|s_h]$ as a conditional expectation of $\sum_{h'=h}^{H} f(s_{h'}, a_{h'})$ with respect to the sigma-algebra $\mathcal{F}_h = \sigma\{(s_{h'}, a_{h'})|h' \le h\}$, where for any $h \in [H]$, we have $a_h \sim \pi(s_h), s_{h+1} \sim p_h(s_h, a_h)$.

We define trajectory KL-divergence between two policies $\pi = \{\pi_h\}_{h \in [H]}, \pi' = \{\pi_h\}_{h \in [H]}$ as follows

$$\mathrm{KL}_{\mathrm{traj}}(\pi, \pi') = \mathbb{E}_\pi\left[\sum_{h=1}^{H} \mathrm{KL}(\pi_h(s_h), \pi'_h(s_h))\right].$$

We write $f(S, A, H, \varepsilon) = \mathcal{O}(g(S, A, H, \varepsilon, \delta))$ if there exist $S_0, A_0, H_0, \varepsilon_0, \delta_0$ and constant $C_{f,g}$ such that for any $S \ge S_0, A \ge A_0, H \ge H_0, \varepsilon < \varepsilon_0, \delta < \delta_0, f(S, A, H, T, \delta) \le C_{f,g} \cdot g(S, A, H, T, \delta)$. We write $f(S, A, H, \varepsilon, \delta) = \widetilde{\mathcal{O}}(g(S, A, H, \varepsilon, \delta))$ if $C_{f,g}$ in the previous definition is poly-logarithmic in $S, A, H, 1/\varepsilon, 1/\delta$.

For any symmetric positive definite matrix $A$, we define the corresponding $A$-scalar product and $A$-norm as follows

$$\langle x, y \rangle_A = \langle x, Ay \rangle, \qquad \|x\|_A = \sqrt{\langle x, x \rangle_A}.$$

Notice that if $\|A\|_2 \le c$, then $\|x\|_A \le \sqrt{c}\|x\|_2$.

**Coverings, packings, and bracketings** A pair $(\mathcal{X}, \rho)$ is called pseudometric space with a metric $\rho: \mathcal{X} \times \mathcal{X} \to \mathbb{R}_+$ if $\rho$ satisfies $\rho(x, x) = 0$ for all $x \in \mathcal{X}$, $\rho$ is symmetric, that is, $\forall x, y \in \mathcal{X}: \rho(x, y) = \rho(y, x)$, and $\rho$ satisfies triangle inequality $\forall x, y, z: \rho(x, y) + \rho(y, z) \ge \rho(x, z)$.

**Definition 6** ($\varepsilon$-covering and packing). Let $(\mathcal{X}, \rho)$ be a (pseudo)metric space with a metric $\rho: \mathcal{X} \times \mathcal{X} \to \mathbb{R}_+$. The $\varepsilon$-covering number $\mathcal{N}(\varepsilon, \mathcal{X}, \rho)$ is the size of the minimal $\varepsilon$-cover of $(\mathcal{X}, \rho)$, that is,

$$\mathcal{N}(\varepsilon, \mathcal{X}, \rho) = \min_{X \subseteq \mathcal{X}}\{|X| : \forall y \in \mathcal{X}\ \exists x \in X : \rho(y, x) \le \varepsilon\}.$$

The $\varepsilon$-packing number $\mathcal{P}(\varepsilon, \mathcal{X}, \rho)$ is the size of the maximal $\varepsilon$-separated set of $(\mathcal{X}, \rho)$,

$$\mathcal{P}(\varepsilon, \mathcal{X}, \rho) = \max_{X \subseteq \mathcal{X}}\{|X| : \forall x \ne y \in X : \rho(x, y) > \varepsilon\}.$$

**Definition 7** ($\varepsilon$-bracketing). Let $\mathcal{F}: \mathcal{X} \to \mathbb{R}$ be a function class endowed with a norm $\|\cdot\|$. Given two functions $\ell, u: \mathcal{X} \to \mathbb{R}$, a bracket $[\ell, u]$ is a set of all functions $f \in \mathcal{F}$ such that $\ell(x) \le f(x) \le u(x)$ for all $x \in \mathcal{X}$. A $\varepsilon$-bracket is a bracket $[\ell, u]$ such that $\|\ell - u\| \le \varepsilon$. The $\varepsilon$-bracketing number $\mathcal{N}_{[]}(\varepsilon, \mathcal{F}, \|\cdot\|)$ is the cardinality of the minimal set of $\varepsilon$-brackets needed to cover $\mathcal{F}$,

$$\mathcal{N}_{[]}(\mathcal{F}, \|\cdot\|) = \min_N\{|N| \mid \forall f \in \mathcal{F}\ \exists [\ell, u] \in N : \ell(x) \le f(x) \le u(x) \forall x \in \mathcal{X}, \|\ell - u\| \le \varepsilon\}.$$

## B  BEHAVIOR CLONING

In this appendix, we gather the proofs of the results for behavior cloning presented in Section 3.

### B.1  PROOF FOR GENERAL SETTING

In this appendix, we provide the proof of Theorem 1.

**Theorem** (Restatement of Theorem 1). *Assume Assumptions 1-2 and that $0 \leq \mathcal{R}_h(\pi_h) \leq M$ for all $h \in [H]$, for any policy $\pi \in \mathcal{F}_h$. Let $\pi^{\mathrm{BC}}$ be a solution to (1). Then with probability at least $1 - \delta$ the behavior policy $\pi^{\mathrm{BC}}$ satisfies*

$$\mathrm{KL}_{\mathrm{traj}}(\pi^{\mathrm{E}} \| \pi^{\mathrm{BC}}) \leq \frac{6 d_{\mathcal{F}} H \cdot (\log(A e^3 / (A\gamma \wedge \kappa))) \cdot \log(2 H N^{\mathrm{E}} R_{\mathcal{F}} / (\gamma\delta))}{N^{\mathrm{E}}} + \frac{2 H M}{N^{\mathrm{E}}} + \frac{18\kappa}{1 - \kappa}.$$

*Proof.* We commence by defining the one-step trajectory KL-divergence as follows:

$$\mathrm{KL}_{\mathrm{traj}}(\pi_h^{\mathrm{E}} \| \pi_h^{\mathrm{BC}}) = \mathbb{E}_{\pi^{\mathrm{E}}}\left[\log\left(\frac{\pi_h^{\mathrm{E}}(a_h|s_h)}{\pi_h^{\mathrm{BC}}(a_h|s_h)}\right)\right].$$

In particular, by the linearity of expectation, the following holds

$$\mathrm{KL}_{\mathrm{traj}}(\pi^{\mathrm{E}} \| \pi^{\mathrm{BC}}) = \sum_{h=1}^{H} \mathrm{KL}_{\mathrm{traj}}(\pi_h^{\mathrm{E}} \| \pi_h^{\mathrm{BC}}).$$

Recall the definition of the $\kappa$-greedy version of the expert policy

$$\pi_h^{\mathrm{E},\kappa}(a|s) = (1 - \kappa)\pi_h^{\mathrm{E}}(a|s) + \frac{\kappa}{A} = (1 - \kappa) \cdot \left(\pi_h^{\mathrm{E}}(a|s) + \frac{\kappa}{(1 - \kappa)A}\right).$$

Next, we can decompose the one-step trajectory KL-divergence as follows

$$\mathrm{KL}_{\mathrm{traj}}(\pi_h^{\mathrm{E}} \| \pi_h^{\mathrm{BC}}) = \mathbb{E}_{\pi^{\mathrm{E}}}\left[\log\left(\frac{\pi_h^{\mathrm{E},\kappa}(a_h|s_h)}{\pi_h^{\mathrm{BC}}(a_h|s_h)}\right)\right] + \mathbb{E}_{\pi^{\mathrm{E}}}\left[\log\left(\frac{\pi_h^{\mathrm{E}}(a_h|s_h)}{\pi_h^{\mathrm{E},\kappa}(a_h|s_h)}\right)\right].$$

For the second term, we have

$$\mathbb{E}_{\pi^{\mathrm{E}}}\left[\log\left(\frac{\pi_h^{\mathrm{E}}(a_h|s_h)}{\pi_h^{\mathrm{E},\kappa}(a_h|s_h)}\right)\right] = \mathbb{E}_{\pi^{\mathrm{E}}}\Bigg[\underbrace{\log\left(\frac{\pi_h^{\mathrm{E}}(a_h|s_h)}{\pi_h^{\mathrm{E}}(a_h|s_h) + \kappa/(A(1 - \kappa))}\right)}_{\leq 0}\Bigg] - \log(1 - \kappa) \leq \frac{\kappa}{1 - \kappa},$$

where the last inequality follows from the fact that $(1 - x)\log(1 - x) \geq -x$ for any $x < 1$, by convexity of the function $x \mapsto x \log x$. Next, we decompose the smoothed version of the one-step trajectory KL to the sum of stochastic and empirical terms,

$$\mathbb{E}_{\pi^{\mathrm{E}}}\left[\log\left(\frac{\pi_h^{\mathrm{E},\kappa}(a_h|s_h)}{\pi_h^{\mathrm{BC}}(a_h|s_h)}\right)\right] = \frac{1}{N^{\mathrm{E}}} \sum_{t=1}^{N^{\mathrm{E}}}\left(\mathbb{E}_{\pi^{\mathrm{E}}}\left[\log\left(\frac{\pi_h^{\mathrm{E},\kappa}(a_h|s_h)}{\pi_h^{\mathrm{BC}}(a_h|s_h)}\right)\right] - \log\left(\frac{\pi_h^{\mathrm{E},\kappa}(a_h^t|s_h^t)}{\pi_h^{\mathrm{BC}}(a_h^t|s_h^t)}\right)\right)$$
$$+ \frac{1}{N^{\mathrm{E}}} \sum_{t=1}^{N^{\mathrm{E}}} \log\left(\frac{\pi_h^{\mathrm{E},\kappa}(a_h^t|s_h^t)}{\pi_h^{\mathrm{BC}}(a_h^t|s_h^t)}\right).$$

To upper bound the first term we apply Lemma 3 and obtain with probability at least $1 - \delta$

$$\frac{1}{N^{\mathrm{E}}} \sum_{t=1}^{N^{\mathrm{E}}}\left(\mathbb{E}_{\pi^{\mathrm{E}}}\left[\log\left(\frac{\pi_h^{\mathrm{E},\kappa}(a_h|s_h)}{\pi_h^{\mathrm{BC}}(a_h|s_h)}\right)\right] - \log\left(\frac{\pi_h^{\mathrm{E},\kappa}(a_h^t|s_h^t)}{\pi_h^{\mathrm{BC}}(a_h^t|s_h^t)}\right)\right)$$
$$\leq \sqrt{\frac{2\log(\mathrm{e}^2/\gamma)\,\mathrm{KL}_{\mathrm{traj}}(\pi_h^{\mathrm{E}} \| \pi_h^{\mathrm{BC}}) \cdot d(\log(2 N^{\mathrm{E}} R_{\mathcal{F}}/\gamma) + \log(1/\delta))}{N^{\mathrm{E}}}}$$
$$+ \frac{5(\log(A e^3 / (A\gamma \wedge \kappa))) \cdot d_{\mathcal{F}}(\log(2 N^{\mathrm{E}} R_{\mathcal{F}}/\gamma) + \log(1/\delta)))}{3 N^{\mathrm{E}}} + \frac{8\kappa}{1 - \kappa}.$$

To control the second term, we first notice that since $\mathcal{F}$ has a product structure, then by a simple observation

$$\{\pi_h\}_{h=1}^H = \underset{\pi_1 \in \mathcal{F}_1, \ldots, \pi_H \in \mathcal{F}_H}{\arg\min} \sum_{h=1}^H \mathcal{L}_h(\pi_h) \iff \forall h \in [H] : \pi_h = \underset{\pi_h \in \mathcal{F}_h}{\arg\min} \mathcal{L}_h(\pi_h)$$

for any functions $\{\mathcal{L}_h\}_{h=1}^H$, the MLE estimation (1) implies

$$\pi_h^{\mathrm{BC}} \in \underset{\pi_h \in \mathcal{F}_h}{\arg\min} \sum_{i=1}^{N^{\mathrm{E}}} \log \frac{1}{\pi_h(a_h^i | s_h^i)} + \mathcal{R}_h(\pi_h),$$

therefore the following holds

$$\sum_{t=1}^{N^{\mathrm{E}}} \log\left( \frac{\pi_h^{\mathrm{E},\kappa}(a_h^t | s_h^t)}{\pi_h^{\mathrm{BC}}(a_h^t | s_h^t)} \right) \leq M + \left\{ \sum_{t=1}^{N^{\mathrm{E}}} \log\left( \frac{1}{\pi_h^{\mathrm{BC}}(a_h^t | s_h^t)} \right) + \mathcal{R}_h(\pi_h^{\mathrm{BC}}) \right\}$$

$$- \left\{ \sum_{t=1}^{N^{\mathrm{E}}} \log\left( \frac{1}{\pi_h^{\mathrm{E},\kappa}(a_h^t | s_h^t)} \right) + \mathcal{R}_h(\pi_h^{\mathrm{E},\kappa}) \right\} \leq M.$$

Thus, we have

$$\mathrm{KL}_{\mathrm{traj}}(\pi_h^{\mathrm{E}} \| \pi_h^{\mathrm{BC}}) \leq \sqrt{ \frac{2 \log(\mathrm{e}^2/\gamma) \, \mathrm{KL}_{\mathrm{traj}}(\pi_h^{\mathrm{E}} \| \pi_h^{\mathrm{BC}}) \cdot d_{\mathcal{F}}(\log(2N^{\mathrm{E}} R_{\mathcal{F}}/\gamma) + \log(1/\delta))}{N^{\mathrm{E}}} }$$

$$+ \frac{5(\log(A\mathrm{e}^3/(A\gamma \wedge \kappa)) \cdot d_{\mathcal{F}}(\log(2N^{\mathrm{E}} R_{\mathcal{F}}/\gamma) + \log(1/\delta)))}{3N^{\mathrm{E}}} + \frac{M}{N^{\mathrm{E}}} + \frac{9\kappa}{1-\kappa}.$$

This means that $\sqrt{\mathrm{KL}_{\mathrm{traj}}(\pi_h^{\mathrm{E}} \| \pi_h^{\mathrm{BC}})}$ satisfies a quadratic inequality of the form $x^2 \leq ax + b$. Since $ax \leq (a^2 + x^2)/2$, we further have $x^2 \leq a^2 + 2b$. As a result

$$\mathrm{KL}_{\mathrm{traj}}(\pi_h^{\mathrm{E}} \| \pi_h^{\mathrm{BC}}) \leq \frac{6d_{\mathcal{F}} \log(A\mathrm{e}^3/(A\gamma \wedge \kappa)) \cdot \log(2N^{\mathrm{E}} R_{\mathcal{F}}/(\gamma\delta))}{N^{\mathrm{E}}} + \frac{2M}{N^{\mathrm{E}}} + \frac{18\kappa}{1-\kappa}.$$

To conclude the statement, we apply a union bound over $h \in [H]$ and sum over the final upper bound. □

## B.2 Proofs for Finite setting

We recall that for finite MDPs we chose a logarithmic regularizer $\mathcal{R}_h(\pi_h) = \sum_{s,a} \log(1/\pi_h(a|s))$ and the policy class $\mathcal{F} = \{\pi \in \Pi : \pi_h(a|s) \geq 1/(N^{\mathrm{E}} + A)\}$. One can check that Assumptions 1-2 holds for these choices and that $0 \leq \mathcal{R}_h(\pi_h) \leq SA \log(A)$. Then we can apply Theorem 1 to obtain the following bound for finite MDPs.

**Corollary** (Restatement of Corollary 1). *For all $N^{\mathrm{E}} \geq A$, for function class $\mathcal{F}$ and regularizer $(\mathcal{R}_h)_{h \in [H]}$ defined above, with probability at least $1 - \delta$,*

$$\mathrm{KL}_{\mathrm{traj}}(\pi^{\mathrm{E}} \| \pi^{\mathrm{BC}}) \leq \frac{6SAH \cdot \log(2\mathrm{e}^4 N^{\mathrm{E}}) \cdot \log(12H(N^{\mathrm{E}})^2/\delta)}{N^{\mathrm{E}}} + \frac{18AH}{N^{\mathrm{E}}}.$$

*Proof.* Let us start from a simple observation that $\mathcal{F}_h \subseteq \Delta_{\mathcal{A}}^{\mathcal{S}}$ is a subset of a unit ball in $\ell_\infty$-norm. Therefore by a standard result in the covering numbers for finite-dimensional Banach spaces (see i.e. Problem 5.5 by van Handel (2016))

$$\log \mathcal{N}(\varepsilon, \mathcal{F}_h, \|\cdot\|_\infty) \leq SA \log(3/\varepsilon).$$

Thus, the parametric classes $\{\mathcal{F}_h\}_{h \in [H]}$ satisfies Assumption 1 with constants $d_{\mathcal{F}} = SA, R_{\mathcal{F}} = 3, \gamma = 1/(N^{\mathrm{E}} + A)$. Next we notice that for *any* expert policy, Assumption 2 is satisfied with $\kappa = A/(N^{\mathrm{E}} + A)$ for this parametric family. Thus, we can apply Theorem 1 and get

$$\mathrm{KL}_{\mathrm{traj}}(\pi^{\mathrm{E}} \| \pi^{\mathrm{BC}}) \leq \frac{6SAH \cdot \log((N^{\mathrm{E}} + A)\mathrm{e}^3) \cdot \log(6HN^{\mathrm{E}}(N^{\mathrm{E}} + A)/\delta)}{N^{\mathrm{E}}}$$

$$+ \frac{2SAH \log(A)}{N^{\mathrm{E}}} + \frac{18AH}{N^{\mathrm{E}}}.$$

By upper bounding the first and the second terms under the assumption $N^{\mathrm{E}} \geq A$ we conclude the statement. □

### B.3 Proofs for Linear setting

We start from a natural example when Assumption 3 is fulfilled and the sub-optimality error $\varepsilon_{\mathrm{E}}$ is small.

**Lemma 1.** *Assume that the MDP $\mathcal{M}$ is linear (see Definition 2) and consider the regularized MDP with uniform policy $\widetilde{\pi}(a|s) = \mathcal{U}\mathrm{nif}[A]$ and with a coefficient $\lambda$ (see Appendix E for more exposition). Then the optimal regularied policy $\pi^{\star}_{\widetilde{\pi},\lambda,h}$ satisfies Assumption 3 with a constant $R = H\sqrt{d}/\lambda$. Moreover, this policy is $\lambda H \log(A)$-optimal.*

*Proof.* At first, by Proposition 2, it holds that for an optimal policy $\pi^{\star}_{\widetilde{\pi},\lambda,h}$ there are some weights $w^{\star}_h$ such that $Q^{\star}_h(s,a) = \langle \psi(s,a), w^{\star}_h\rangle$ and, moreover, $\|w^{\star}_h\| \leq H\sqrt{d}$.

Then we notice that from the regularized Bellman equations it holds

$$\pi^{\star}_{\widetilde{\pi},\lambda,h}(a|s) = \arg\max_{\pi}\big\{\pi Q^{\star}_{\widetilde{\pi},\lambda,h}(s) - \lambda\,\mathrm{KL}(\pi\|\mathcal{U}\mathrm{nif}[A])\big\}$$

$$= \arg\max_{\pi}\left\{\pi\left[\frac{1}{\lambda}Q^{\star}_{\widetilde{\pi},\lambda,h}\right](s) - \mathrm{KL}(\pi\|\mathcal{U}\mathrm{nif}[A])\right\} = \frac{\exp\big(\langle\psi(s,a),\frac{1}{\lambda}w^{\star}_h\rangle\big)}{Z(s)}.$$

Therefore, Assumption 3 is satisfied with $R = H\sqrt{d}/\lambda$ for $\pi^{\mathrm{E}} = \pi^{\star}_\lambda$.

To verify the suboptimality of this policy, we notice that $\pi^{\star}_{\widetilde{\pi},\lambda}$ satisfies

$$\pi^{\star}_{\widetilde{\pi},\lambda} = \arg\max_{\pi\in\Pi}\{V^{\pi}_1(s_1) - \lambda\,\mathrm{KL}_{\mathrm{traj}}(\pi\|\widetilde{\pi})\},$$

therefore

$$V^{\star}_1(s_1) - \lambda\,\mathrm{KL}_{\mathrm{traj}}(\pi^{\star}\|\widetilde{\pi})\} \leq V^{\pi^{\star}_{\widetilde{\pi},\lambda}}_1(s_1) - \lambda\,\mathrm{KL}_{\mathrm{traj}}(\pi^{\star}_{\widetilde{\pi},\lambda}\|\mathcal{U}\mathrm{nif}[A])$$

$$\Rightarrow V^{\star}_1(s_1) - V^{\pi^{\star}_{\widetilde{\pi},\lambda}}_1(s_1) \leq \lambda H \log(A).$$

$\square$

Next, we provide the result for linear MDPs under Assumption 3, using the parametric assumption given in (2).

**Corollary** (Restatement of Corollary 2)**.** *Under Assumption 3, for all $N^{\mathrm{E}} \geq A$, for the function class $\mathcal{F}$ defined in (2) and regularizer $\mathcal{R}_h = 0$, for all $h \in [H]$, with probability at least $1 - \delta$,*

$$\mathrm{KL}_{\mathrm{traj}}(\pi^{\mathrm{E}}\|\pi^{\mathrm{BC}}) \leq \frac{6dH \cdot \log(2\mathrm{e}^3 N^{\mathrm{E}}) \cdot \log(48H(N^{\mathrm{E}})^2 R/\delta)}{N^{\mathrm{E}}} + \frac{18AH}{N^{\mathrm{E}}}.$$

*Proof.* We start by checking that Assumption 1 holds. By construction of $\mathcal{F}_h$ in (2), we have

$$\inf_{\pi_h\in\mathcal{F}_h}\inf_{(s,a)\in\mathcal{S}\times\mathcal{A}}\pi_h(a|s) \geq \frac{1}{N^{\mathrm{E}} + A}.$$

Next, we have to consider the covering dimension of the hypothesis set. First, we notice that for any two policies $\pi_h, \mu_h \in \mathcal{F}_h$ we have

$$|\pi_h(a|s) - \mu_h(a|s)| = |(1-\kappa)\pi'_h(a|s) - (1-\kappa)\mu'_h(a|s)| = (1-\kappa)|\pi'_h(a|s) - \mu'_h(a|s)|,$$

where $\pi'_h, \mu'_h \in \mathcal{F}'_h$ for $\mathcal{F}'_h$ defined as follows

$$\mathcal{F}'_h = \left\{\pi_h(a|s) = \frac{\exp(\psi(s,a)^{\top}w_h)}{\sum_{a'\in\mathcal{A}}\exp(\psi(s,a')^{\top}w_h)} : \|w_h\|_2 \leq R\right\}.$$

Thus, it is sufficient to compute the covering number for $\mathcal{F}'_h$. Let us define

$$\Phi(a|s, w_h) = \exp\{\langle\psi(s,a), w_h\rangle\}, \quad Z(s, w_h) = \sum_{a\in\mathcal{A}}\Phi(a|s, w_h).$$

Then let $w_h, w'_h$ be weight vectors that correspond to $\pi'_h$ and $\mu'_h$ respectively. Then

$$|\pi'_h(a|s) - \mu'_h(a|s)| = \left| \frac{\Phi(a|s, w_h)}{Z(s, w_h)} - \frac{\Phi(a|s, w'_h)}{Z(s, w'_h)} \right|$$

$$= \left| \frac{\Phi(a|s, w_h) - \Phi(a|s, w'_h)}{Z(s, w_h)} - \Phi(a|s, w'_h) \left[ \frac{1}{Z(s, w'_h)} - \frac{1}{Z(s, w_h)} \right] \right|$$

$$\leq \frac{\Phi(a|s, w_h)}{Z(s, w_h)} \left| 1 - \frac{\Phi(a|s, w'_h)}{\Phi(a|s, w_h)} \right| + \frac{\Phi(a|s, w'_h)}{Z(s, w'_h)} \left| 1 - \frac{Z(s, w'_h)}{Z(s, w_h)} \right|.$$

Next, we analyze both terms separately. For the first term, we have

$$\left| 1 - \frac{\Phi(a|s, w'_h)}{\Phi(a|s, w_h)} \right| = |1 - \exp\{\langle \psi(s, a), w'_h - w_h \rangle\}|.$$

We notice that the absolute value of the expression under exponent is upper-bounded by $\|w_h - w'_h\|_2$. Let us assume that $\|w_h - w'_h\|_2 \leq 1$, then by the inequality $|1 - e^x| \leq 2|x|$ for any $|x| \leq 1$, we have

$$\left| 1 - \frac{\Phi(a|s, w'_h)}{\Phi(a|s, w_h)} \right| \leq 2\|w_h - w'_h\|_2. \tag{3}$$

For the second term, we have by the definition of the normalization constant

$$\left| 1 - \frac{Z(s, w'_h)}{Z(s, w_h)} \right| = \left| \frac{\sum_{a'} \Phi(a'|s, w_h)[1 - \Phi(a'|s, w'_h)/\Phi(a'|s, w_h)]}{Z(s, w_h)} \right|$$

$$\leq \frac{\sum_{a'} \Phi(a'|s, w_h) \cdot |1 - \Phi(a'|s, w'_h)/\Phi(a'|s, w_h)|}{Z(s, w_h)} \leq 2\|w_h - w'_h\|_2,$$

where in the end we applied (3). Finally, we have, for any policies $\pi'_h$ and $\mu'_h$ such that the corresponding weights $w_h$ and $w'_h$ satisfies $\|w_h - w'_h\|_2 \leq 1$, that

$$|\pi_h(a|s) - \mu_h(a|s)| \leq |\pi'_h(a|s) - \mu'_h(a|s)| \leq 4\|w_h - w'_h\|_2. \tag{4}$$

Now we construct an $\varepsilon$-net for $\varepsilon \in (0, 1)$. Let $\mathcal{N}_{\varepsilon/4}(W, \|\cdot\|_2)$ be a $\varepsilon/4$-net in the space of weights $W = \{w_h \in \mathbb{R}^d : \|w_h\|_2 \leq R\}$. It satisfies (see i.e. van Handel (2016))

$$\log \mathcal{N}(\varepsilon/4, W, \|\cdot\|_2)| \leq d \log(12R/\varepsilon).$$

Next, we show that policies with weights that correspond to a covering of size $\mathcal{N}(\varepsilon/4, W_h, \|\cdot\|_2)$ forms an $\varepsilon$-net in $\mathcal{F}_h$. Let $\pi_h \in \mathcal{F}_h$ be an arbitrary policy with parameter $w_h$. Let $w'_h$ be in the covering of size $\mathcal{N}(\varepsilon/4, W, \|\cdot\|_2)$ be a parameter that satisfies $\|w_h - w'_h\|_2 \leq \varepsilon/4 \leq 1$. Let us fix $\mu_h$ as a policy that corresponds to $w'_h$. Since $\|w_h - w'_h\|_2 \leq 1$, (4) is applicable. Thus

$$\|\pi_h - \mu_h\|_\infty = \sup_{(s,a) \in \mathcal{S} \times \mathcal{A}} |\pi_h(a|s) - \mu_h(a|s)| \leq 4\|w_h - w'_h\|_2 \leq \varepsilon.$$

Therefore, policies that correspond to an $\varepsilon/4$-net in $w_h$ form an $\varepsilon$-net in $\mathcal{F}$ and we have an upper bound on the size of the $\varepsilon$-net. As a result, $\mathcal{F}_h$ satisfies Assumption 1 with a dimension $d_{\mathcal{F}} = d$, a scaling factor $R_{\mathcal{F}} = 12R$ and $\gamma = 1/(N^E + A)$. Additionally, by construction of $\mathcal{F}_h$ and Assumption 3, the last Assumption 2 holds with $\kappa = A/(N^E + A)$. Therefore, we can apply Theorem 1 and obtain with probability at least $1 - \delta$

$$\text{KL}_{\text{traj}}(\pi^E \| \pi^{BC}) \leq \frac{6dH \cdot (\log(e^3(N^E + A)) \cdot (\log(24HN^E(N^E + A)R/\delta)))}{N^E} + \frac{18AH}{N^E}.$$

Using of $A \leq N^E$ concludes the statement. $\square$

## B.4 Concentration Results

In this section, we state important results on the concentration of the stochastic error for the risk estimates.

Recall the definition of the $\kappa$-greedy version of the expert policy as follows

$$\pi_h^{E,\kappa}(a|s) = (1 - \kappa)\pi_h^E(a|s) + \frac{\kappa}{A} = (1 - \kappa) \cdot \left( \pi_h^E(a|s) + \frac{\kappa}{(1 - \kappa)A} \right).$$

**Lemma 2.** *Let $\pi^E$ be a fixed expert policy. Let $(s_h^t, a_h^t)_{t=1}^N$ be an i.i.d. sequence of state-action pairs generated by following the policy $\pi^E$ at step $h$. For $\gamma \in (0, 1/A)$ let $\pi$ a policy such that for all $(s, a) \in \mathcal{S} \times \mathcal{A}$ it holds $\pi_h(a|s) \geq \gamma$. Then for any $\delta \in (0, 1)$ and any $\kappa < 1/2$ with probability at least $1 - \delta$*

$$\left| \frac{1}{N} \sum_{t=1}^N \log \left( \frac{\pi_{h,\sigma}^E(a_h^t|s_h^t)}{\pi_h(a_h^t|s_h^t)} \right) - \mathbb{E}_{\pi^E} \left[ \log \left( \frac{\pi_h^{E,\kappa}(a_h|s_h)}{\pi_h(a_h|s_h)} \right) \right] \right| \leq \sqrt{\frac{2 \log(e^2/\gamma) \, \mathrm{KL}_{\mathrm{traj}}(\pi_h^E \| \pi_h) \log(2/\delta)}{N}}$$

$$+ \frac{2 \log(Ae^3/(A\gamma \wedge \kappa)) \cdot \log(2/\delta)}{3N} + \frac{5\kappa}{1 - \kappa}.$$

*Proof.* As a first step, we can apply Bernstein inequality

$$\left| \frac{1}{N} \sum_{t=1}^N \log \left( \frac{\pi_h^{E,\kappa}(a_h^t|s_h^t)}{\pi_h(a_h^t|s_h^t)} \right) - \mathbb{E}_{\pi^E} \left[ \log \left( \frac{\pi_h^{E,\kappa}(a_h|s_h)}{\pi_h(a_h|s_h)} \right) \right] \right| \leq \sqrt{\frac{2\mathrm{Var}_{\pi^E} \left[ \log \left( \frac{\pi_h^{E,\kappa}(a_h|s_h)}{\pi_h(a_h|s_h)} \right) \right] \log(2/\delta)}{N}}$$

$$+ \frac{2(\log(1/\gamma) \vee \log(A/\kappa)) \cdot \log(2/\delta)}{3N}.$$

Next, we want to upper bound a variance in terms of $\mathrm{KL}_{\mathrm{traj}}(\pi_h^E \| \pi_h)$. We start from a bound of square root variance in terms of the second moment and Minkowski inequality

$$\sqrt{\mathrm{Var}_{\pi^E} \left[ \log \left( \frac{\pi_h^{E,\kappa}(a_h|s_h)}{\pi_h(a_h|s_h)} \right) \right]} \leq \sqrt{\mathbb{E}_{\pi^E} \left[ \left( \log \left( \frac{\pi_h^{E,\kappa}(a_h|s_h)}{\pi_h(a_h|s_h)} \right) \right)^2 \right]}$$

$$= \sqrt{\mathbb{E}_{\pi^E} \left[ \left( \log \left( \frac{\pi_h^E(a|s)}{\pi_h(a|s)} \right) + \log \left( \frac{\pi_h^{E,\kappa}(a|s)}{\pi_h^E(a|s)} \right) \right)^2 \right]}$$

$$\leq \sqrt{\mathbb{E}_{\pi^E} \left[ \left( \log \left( \frac{\pi_h^E(a_h|s_h)}{\pi_h(a_h|s_h)} \right) \right)^2 \right]} \qquad = \sqrt{(\mathbf{A})}$$

$$+ \sqrt{\mathbb{E}_{\pi^E} \left[ \left( \log \left( 1 + \frac{\kappa}{(1-\kappa)A\pi_h^E(a|s)} \right) + \log(1 - \kappa) \right)^2 \right]}. \qquad = \sqrt{(\mathbf{B})}$$

**Term $(\mathbf{A})$.** The result below directly follows from Lemma 4 of [Yang & Barron (1998)](#). However, for completeness, we prove it here.

First, we notice that

$$(\mathbf{A}) = \mathbb{E}_{\pi^E} \left[ \sum_{a \in \mathcal{A}} \pi_h^E(a|s_h) \log^2 \left( \frac{\pi_h^E(a|s_h)}{\pi_h(a|s_h)} \right) \right].$$

To analyze this term, we define an $f$-divergence ([Sason & Verdú, 2016](#)) for a function $f$ as follows

$$D_f(\pi_h^E(s) \| \pi_h(s)) = \sum_{a \in \mathcal{A}} f \left( \frac{\pi_h^E(a|s)}{\pi_h(a|s)} \right) \pi_h(a|s).$$

In particular, $\mathrm{KL}(\pi_h^E(s), \pi_h(s)) = D_g(\pi_h^E(s) \| \pi_h(s))$ for $g(t) = t \log t + (1 - t)$ and, moreover for $f(t) = t \log^2(t)$

$$D_f(\pi_h^E(s) \| \pi_h(s)) = \sum_{a \in \mathcal{A}} \pi_h^E(a|s) \log^2 \left( \frac{\pi_h^E(a|s)}{\pi_h(a|s)} \right).$$

Then we notice that $g$ and $f$ are non-negative function and, moreover, its argument $t$ takes values in $(0, 1) \cup (1, 1/\gamma]$ since for $t = 1$ both functions are zero. First, we analyze the ratio for $f$ and $g$ for any $t \in (0, 1)$

$$r(t) = \frac{f(t)}{g(t)} = \frac{t \log^2(t)}{t \log t + (1 - t)}.$$

To bound this function, let us prove that is monotone for all $t \in (0, 1)$

$$r'(t) = \frac{\overbrace{\log(t)}^{\leq 0} \cdot \overbrace{((t+1)\log(t) + 2(1-t))}^{\leq 0}}{(t \log t + (1-t))^2} \geq 0.$$

Thus for any $t \in (0, 1)$

$$r(t) \leq \lim_{t \to 1} \frac{t \log^2(t)}{t \log t + (1-t)} = 2.$$

Next, we analyze the segment $t \in (1, 1/\gamma]$.

$$r(t) = \frac{t \log^2(t)}{t \log t + (1-t)} \leq \log(t) + 2 \leq \log(1/\gamma) + 2 = \log(e^2/\gamma).$$

since

$$t \log^2(t) \leq t \log^2(t) + (1-t) \log t + 2t \log(t) + 2(1-t) \iff (t+1)\log(t) + 2(1-t) \geq 0 \quad \forall t > 1.$$

Therefore we have for any $t \in (0, 1) \cup (1, 1/\gamma]$ and as a simple corollary

$$f(t) \leq \log(e^2/\gamma) \cdot g(t) \Rightarrow D_f(\pi_h^E(s) \| \pi_h(s)) \leq \log(e^2/\gamma) \cdot \mathrm{KL}(\pi_h^E(s) \| \pi_h(s)).$$

Finally, we have

$$(\mathbf{A}) \leq \log(e^2/\gamma) \mathbb{E}_{\pi^E} \left[ \mathrm{KL}(\pi_h^E(s_h), \pi_h(s_h)) \right] = \log(e^2/\gamma) \, \mathrm{KL}_{\mathrm{traj}}(\pi_h^E \| \pi_h).$$

**Term $(\mathbf{B})$.** We can rewrite this term as follows using inequality $(a + b)^2 \leq 2a^2 + 2b^2$

$$(\mathbf{B}) \leq 2\mathbb{E}_{\pi^E} \left[ \sum_{a:\pi_h^E(a|s_h)>0} \pi_h^E(a|s_h) \log^2\left(1 + \frac{\kappa}{(1-\kappa)A\pi_h^E(a_h|s_h)}\right) \right] + 2\left(\frac{\kappa}{1-\kappa}\right)^2.$$

Next we analyze the function $g(x) = x \log^2(1 + \varepsilon/x)$. Its derivative is equal to

$$g'(x) = \log\left(1 + \frac{\varepsilon}{x}\right) \cdot \left(\log\left(1 + \frac{\varepsilon}{x}\right) - \frac{2\varepsilon}{x+\varepsilon}\right).$$

Since $\varepsilon > 0$. we can define $x^\star$ as a root of equation $g'(x) = 0$ for $x > 0$. Notice that it will be maximum of $g(x)$, thus for $\varepsilon > 0$

$$g(x) \leq g(x^\star) = x^\star \left(\log\left(1 + \frac{\varepsilon}{x^\star}\right)\right)^2 = x^\star \frac{4\varepsilon^2}{(x^\star + \varepsilon)^2} \leq \frac{4\varepsilon^2}{x^\star + \varepsilon} \leq 4\varepsilon.$$

Therefore

$$(\mathbf{B}) \leq \frac{8\kappa}{1-\kappa} + 2\left(\frac{\kappa}{1-\kappa}\right)^2.$$

**Final bound on variance** Combining these two bounds, we have

$$\sqrt{\mathrm{Var}_{\pi^E}\left[\log\left(\frac{\pi_h^{E,\kappa}(a_h|s_h)}{\pi_h(a_h|s_h)}\right)\right]} \leq \sqrt{\log(e^2/\gamma) \cdot \mathrm{KL}_{\mathrm{traj}}(\pi_h^E \| \pi_h)} + \sqrt{8\kappa/(1-\kappa) + 2\kappa^2/(1-\kappa)^2}.$$

Using this bound on variance, we can bound the main stochastic term

$$\sqrt{\frac{2\mathrm{Var}_{\pi^E}\left[\log\left(\frac{\pi_h^{E,\kappa}(a_h|s_h)}{\pi_h(a_h|s_h)}\right)\right]\log(2/\delta)}{N}} \leq \sqrt{\frac{2\log(e^2/\gamma)\,\mathrm{KL}_{\mathrm{traj}}(\pi_h^E \| \pi_h)\log(2/\delta)}{N}}$$

$$+ 2\sqrt{\frac{(4\kappa/(1-\kappa) + \kappa^2/(1-\kappa)^2)\log(2/\delta)}{N}}.$$

Next, we use an inequality $2ab \leq a^2 + b^2$ to obtain

$$2\sqrt{(4\kappa/(1-\kappa) + \kappa^2/(1-\kappa)^2) \cdot \frac{\log(2/\delta)}{N}} \leq (4\kappa/(1-\kappa) + \kappa^2/(1-\kappa)^2) + \frac{2\log(2/\delta)}{N}.$$

Finally, since $k/(1-\kappa) \leq 1$ we conclude the statement. $\qquad \square$

Next we define $\Pi_{\gamma,h}$ a set of all one-step policies $\pi_h(a|s)$ such that

$$\inf_{(s,a)\in\mathcal{S}\times\mathcal{A}} \pi_h(a|s) \geq \gamma.$$

This set forms a metric space with a metric induced by $\ell_\infty$-norm

$$\|\pi_h - \pi'_h\|_\infty = \sup_{(s,a)\times\mathcal{S}\times\mathcal{A}} |\pi_h(a|s) - \pi'_h(a|s)|.$$

**Lemma 3.** *Let $\mathcal{F}$ be sub-space of $\Pi_{\gamma,h}$ with an induced metric, such that it satisfies for all $\varepsilon \in (0,1)$*

$$\log|\mathcal{N}(\varepsilon, \mathcal{F}, \|\cdot\|_\infty)| \leq d\log(R/\varepsilon).$$

*for some positive constants $R, d > 0$. Then with probability at least $1 - \delta$ the following holds for all $\pi_h \in \mathcal{F}$ simultaneously*

$$\left| \frac{1}{N} \sum_{t=1}^{N} \left( \log\left( \frac{\pi_h^{\mathrm{E},\kappa}(a_h^t|s_h^t)}{\pi_h(a_h^t|s_h^t)} \right) - \mathbb{E}_{\pi^{\mathrm{E}}}\left[ \log\left( \frac{\pi_h^{\mathrm{E},\kappa}(a_h|s_h)}{\pi_h(a_h|s_h)} \right) \right] \right) \right|$$
$$\leq \sqrt{ \frac{2\log(\mathrm{e}^2/\gamma)\,\mathrm{KL}_{\mathrm{traj}}(\pi_h^{\mathrm{E}}\|\pi_h)\cdot d(\log(2NR/\gamma) + \log(1/\delta))}{N} }$$
$$+ \frac{5(\log(A\mathrm{e}^3/(\gamma\wedge\sigma))\cdot d(\log(2NR/\gamma) + \log(1/\delta)))}{3N} + \frac{8\kappa}{1-\kappa}.$$

*Proof.* Let $\mathcal{N}_\varepsilon$ be a minimal $\varepsilon$-net of $\mathcal{F}$ for $\varepsilon$ that will be specified later. Combining Lemma 2 with a union bound over $\mathcal{N}_\varepsilon$ we have for any $\pi'_h \in \mathcal{N}_\varepsilon$ with probability at least $1 - \delta$

$$\left| \frac{1}{N} \sum_{t=1}^{N} \left( \log\left( \frac{\pi_h^{\mathrm{E},\kappa}(a_h^t|s_h^t)}{\pi'_h(a_h^t|s_h^t)} \right) - \mathbb{E}_{\pi^{\mathrm{E}}}\left[ \log\left( \frac{\pi_h^{\mathrm{E},\kappa}(a_h|s_h)}{\pi'_h(a_h|s_h)} \right) \right] \right) \right|$$
$$\leq \sqrt{ \frac{2\log(\mathrm{e}^2/\gamma)\,\mathrm{KL}_{\mathrm{traj}}(\pi_h^{\mathrm{E}}\|\pi'_h)\cdot d\log(2R/(\varepsilon\delta))}{N} } \qquad (5)$$
$$+ \frac{2(\log(A\mathrm{e}^3/(A\gamma\wedge\kappa))\cdot d\log(2R/(\varepsilon\delta))}{3N} + \frac{5\kappa}{1-\kappa}.$$

Next, we select an arbitrary policy $\pi_h \in \mathcal{F}$ and let $\pi'_h \in \mathcal{N}_\varepsilon$ be $\varepsilon$-close policy to $\pi_h$. Then

$$\left| \frac{1}{N} \sum_{t=1}^{N} \left( \log\left( \frac{\pi_h^{\mathrm{E},\kappa}(a_h^t|s_h^t)}{\pi_h(a_h^t|s_h^t)} \right) - \mathbb{E}_{\pi^{\mathrm{E}}}\left[ \log\left( \frac{\pi_h^{\mathrm{E},\kappa}(a_h|s_h)}{\pi_h(a_h|s_h)} \right) \right] \right) \right|$$
$$\leq \left| \frac{1}{N} \sum_{t=1}^{N} \left( \log\left( \frac{\pi_h^{\mathrm{E},\kappa}(a_h^t|s_h^t)}{\pi'_h(a_h^t|s_h^t)} \right) - \mathbb{E}_{\pi^{\mathrm{E}}}\left[ \log\left( \frac{\pi_h^{\mathrm{E},\kappa}(a_h|s_h)}{\pi'_h(a_h|s_h)} \right) \right] \right) \right|$$
$$+ \left| \frac{1}{N} \sum_{t=1}^{N} \left( \log\left( \frac{\pi'_h(a_h^t|s_h^t)}{\pi_h(a_h^t|s_h^t)} \right) - \mathbb{E}_{\pi^{\mathrm{E}}}\left[ \log\left( \frac{\pi'_h(a_h|s_h)}{\pi_h(a_h|s_h)} \right) \right] \right) \right|.$$

We start from bounding the second term, which could be done as follows

$$\left| \frac{1}{N} \sum_{t=1}^{N} \left( \log\left( \frac{\pi'_h(a_h^t|s_h^t)}{\pi_h(a_h^t|s_h^t)} \right) - \mathbb{E}_{\pi^{\mathrm{E}}}\left[ \log\left( \frac{\pi'_h(a_h|s_h)}{\pi_h(a_h|s_h)} \right) \right] \right) \right| \leq 2\max_{s,a}\left| \log\left( \frac{\pi'_h(a|s)}{\pi_h(a|s)} \right) \right|.$$

Next, we use simple inequalities

$$\log\left( \frac{\pi'_h(a|s)}{\pi_h(a|s)} \right) = \log\left( 1 + \frac{\pi'_h(a|s) - \pi_h(a|s)}{\pi_h(a|s)} \right) \leq \frac{|\pi'_h(a|s) - \pi_h(a|s)|}{\gamma} \leq \frac{\varepsilon}{\gamma},$$

and, in the opposite direction, we can use the same reasoning

$$\log\left( \frac{\pi'_h(a|s)}{\pi_h(a|s)} \right) = -\log\left( \frac{\pi_h(a|s)}{\pi'_h(a|s)} \right) \geq -\frac{\varepsilon}{\gamma}.$$

Thus, applying (5) for the first term we obtain

$$
\left| \frac{1}{N} \sum_{t=1}^{N} \left( \log\left( \frac{\pi_h^{\mathrm{E},\kappa}(a_h^t|s_h^t)}{\pi_h(a_h^t|s_h^t)} \right) - \mathbb{E}_{\pi^{\mathrm{E}}}\left[ \log\left( \frac{\pi_h^{\mathrm{E},\kappa}(a_h|s_h)}{\pi_h(a_h|s_h)} \right) \right] \right) \right|
$$
$$
\leq \sqrt{ \frac{2\log(\mathrm{e}^2/\gamma)\,\mathrm{KL}_{\mathrm{traj}}(\pi_h^{\mathrm{E}}\|\pi_h')\cdot d\log(2R/(\varepsilon\delta))}{N} }
$$
$$
+ \frac{2(\log(A\mathrm{e}^3/(A\gamma\wedge\kappa))\cdot d\log(2R/(\varepsilon\delta))}{3N} + \frac{5\kappa}{1-\kappa} + 2\varepsilon/\gamma.
$$

Next, we use a similar inequality to obtain

$$
\mathrm{KL}_{\mathrm{traj}}(\pi_h^{\mathrm{E}}\|\pi_h') = \mathbb{E}_{\pi^{\mathrm{E}}}\left[ \log\left( \frac{\pi_h^{\mathrm{E}}(a_h|s_h)}{\pi_h'(a_h|s_h)} \right) \right]
$$
$$
= \mathrm{KL}_{\mathrm{traj}}(\pi_h^{\mathrm{E}}\|\pi_h) + \mathbb{E}_{\pi^{\mathrm{E}}}\left[ \log\left( \frac{\pi_h(a_h|s_h)}{\pi_h'(a_h|s_h)} \right) \right]
$$
$$
\leq \mathrm{KL}_{\mathrm{traj}}(\pi_h^{\mathrm{E}}\|\pi_h) + \frac{\varepsilon}{\gamma}.
$$

Finally, applying inequalities $\sqrt{a+b} \leq \sqrt{a} + \sqrt{b}$ and $2ab \leq a^2 + b^2$

$$
\left| \frac{1}{N} \sum_{t=1}^{N} \left( \log\left( \frac{\pi_h^{\mathrm{E},\kappa}(a_h^t|s_h^t)}{\pi_h(a_h^t|s_h^t)} \right) - \mathbb{E}_{\pi^{\mathrm{E}}}\left[ \log\left( \frac{\pi_h^{\mathrm{E},\kappa}(a_h|s_h)}{\pi_h(a_h|s_h)} \right) \right] \right) \right|
$$
$$
\leq \sqrt{ \frac{2d\log(\mathrm{e}^2/\gamma)\,\mathrm{KL}_{\mathrm{traj}}(\pi_h^{\mathrm{E}}\|\pi_h)\log(2R/(\varepsilon\delta))}{N} }
$$
$$
+ \frac{d\log(\mathrm{e}^2/\gamma)\cdot\log(2R/(\varepsilon\delta))}{N} + \frac{\varepsilon}{\gamma}
$$
$$
+ \frac{2(\log(A\mathrm{e}^3/(A\gamma\wedge\kappa))\cdot d\log(2R/(\varepsilon\delta))}{3N} + \frac{5\kappa}{1-\kappa} + 2\varepsilon/\gamma.
$$

By rearranging the terms and taking $\varepsilon = \gamma\cdot\kappa/(1-\kappa)$ we conclude the statement.

$\square$

### B.5 PROOF OF LOWER BOUNDS

#### B.5.1 GENERAL SETUP

In this section we provide a lower bound on estimation in KL-divergence using a framework of Chapter 2 by Tsybakov (2008). Our goal is to obtain a lower bound on minimax risk that is defined as follows

$$
\inf_{\widehat{\pi}} \sup_{\pi\in\mathcal{F}} \mathbb{E}_{\tau_1,\dots,\tau_N\sim\pi}[\mathrm{KL}_{\mathrm{traj}}(\pi\|\widehat{\pi})],
$$

where infimum is taken over all estimators that map the sampled trajectories $(\tau_1,\dots,\tau_N)$ to a policy from the hypothesis class $\mathcal{F} \triangleq \mathcal{F}_1 \times \dots \mathcal{F}_H$.

Let us consider a specific type of MDPs where the transition kernel $p_h(s,a)$ does not depend on a state-action pair $(s,a)$: $\forall(s,a,h)\in\mathcal{S}\times\mathcal{A}\times[H], \forall A\in\mathcal{F}_{\mathcal{S}}: p_h(A|s,a) = \mu_h(A)$ for fixed measures $\mu_h$. In particular, for $h=1$ we always have $\mu_h = \delta_{s_1}$ is a Dirac measure at initial state $s_1$.

Then we define over the space of all policies the following specific distance defined through the Hellinger distance

$$
\rho_h(\pi_h,\pi_h') = \sqrt{\mathbb{E}_{s\sim\mu_h}[d_{\mathcal{H}}^2(\pi_h,\pi_h')]}, \qquad d_{\mathcal{H}}^2(\pi_h,\pi_h') = \sum_{a\in\mathcal{A}}\left( \sqrt{\pi_h(a|s)} - \sqrt{\pi_h'(a|s)} \right)^2
$$

and we define the following distance for the space of full policies (the triangle inequality follows from Minkowski inequality)

$$\rho(\pi, \pi') = \sqrt{\sum_{h=1}^{H} \rho_h^2(\pi_h, \pi'_h)}. \tag{6}$$

Next we impose the following metric-specific assumption for our hypothesis classes

**Assumption 5.** For all $h \in [H]$ a of the function class $\mathcal{F}_h$ with respect to the metric $\rho_h$ satisfies

$$\forall \varepsilon \in (0,1) : \log \mathcal{P}(\varepsilon, \mathcal{F}_h, \rho_h) \geq d_h \log(R/\varepsilon)$$

for constants $d_h \geq 0$ and $R > 0$.

In particular, Lemma 6 implies that

$$\log \mathcal{P}(\varepsilon, \mathcal{F}, \rho) \geq \sum_{h=1}^{H} d_h \log(R/\varepsilon).$$

**Theorem 4.** *Let Assumption 5 holds and let us define $D = \sum_{h=1}^{H} d_h$. Also we assume that for any $\pi \in \mathcal{F}$ it holds $\pi_h(s,a) \geq \gamma$ for $\gamma \in (0, 1/A)$. Let us assume $D \geq 5$ and $n \geq \mathrm{e}^2 D/R^2$. Then the following minimax lower bound holds*

$$\min_{\widehat{\pi}} \max_{\pi \in \mathcal{F}} \mathbb{E}[\mathrm{KL}_{\mathrm{traj}}(\pi \| \widehat{\pi})] \geq \frac{D}{16N \log(\mathrm{e}^2/\gamma)}.$$

*Proof.* First, we notice by the first part of Lemma 7

$$\min_{\widehat{\pi}} \max_{\pi \in \mathcal{F}} \mathbb{E}\left[ \frac{\log(\mathrm{e}^2/\gamma)N}{D} \mathrm{KL}_{\mathrm{traj}}(\pi \| \widehat{\pi}) \right] \geq \min_{\widehat{\pi}} \max_{\pi \in \mathcal{F}} \mathbb{E}\left[ \frac{\log(\mathrm{e}^2/\gamma)N}{D} \rho^2(\pi, \widehat{\pi}) \right],$$

where the expectation is taken with respect to a sample $\tau_1, \dots, \tau_N$. Next, we can follow the general reduction scheme, see Chapter 2.2 by Tsybakov (2008). By Markov inequality

$$\min_{\widehat{\pi}} \max_{\pi \in \mathcal{F}} \mathbb{E}\left[ \frac{n \log(\mathrm{e}^2/\gamma)}{D} \rho^2(\pi, \widehat{\pi}) \right] \geq \frac{1}{4} \min_{\widehat{\pi}} \max_{\pi \in \mathcal{F}} \mathbb{P}\left[ \rho(\pi, \widehat{\pi}) \geq \sqrt{\frac{D}{4N \log(\mathrm{e}^2/\gamma)}} \right].$$

Next, we use the reduction to a finite hypothesis class. Define $\zeta = \sqrt{D/(4 \log(\mathrm{e}^2/\gamma)N)}$ and take $P$ is a maximal $\zeta$-separated set of size $M + 1 = \mathcal{P}(\zeta, \mathcal{F}, \rho)$ and enumerate all the policies in it as $\pi_0, \dots, \pi_M$.

Therefore we obtain

$$\min_{\widehat{\pi}} \max_{\pi \in \mathcal{F}} \mathbb{P}_{\tau_1, \dots, \tau_N \sim \pi}\left[ \rho(\pi, \widehat{\pi}) \geq \sqrt{\frac{D}{4N \cdot \log(\mathrm{e}^2/\gamma)}} \right]$$

$$\geq \min_{\widehat{\pi}} \max_{j \in \{0, \dots, M\}} \mathbb{P}_{\tau_1, \dots, \tau_N \sim \pi_j}\left[ \rho(\pi_j, \widehat{\pi}) \geq \sqrt{\frac{D}{4N \cdot \log(\mathrm{e}^2/\gamma)}} \right].$$

Let us define $\psi^\star = \arg\min_{j=0,\dots,M} \rho(\pi_j, \widehat{\pi})$. Then we have that if $\psi^\star \neq j$, then

$$2\rho(\pi_j, \widehat{\pi}) \geq \rho(\pi_j, \widehat{\pi}) + \rho(\pi_{\psi^\star}, \widehat{\pi}) \geq \rho(\pi_j, \pi_{\psi^\star}).$$

Since $j \neq \pi^\star$, then by definition of $\zeta$-separable set we have $\rho(\pi_j, \widehat{\pi}) \geq \zeta/2$. As a result

$$\min_{\widehat{\pi}} \max_{j \in \{0,\dots,M\}} \mathbb{P}_{\tau_1,\dots,\tau_N \sim \pi_j}\left[ \rho(\pi_j, \widehat{\pi}) \geq \sqrt{\frac{D}{4N \cdot \log(\mathrm{e}^2/\gamma)}} \right]$$

$$\geq \min_{\widehat{\pi}} \max_{j \in \{0,\dots,M\}} \mathbb{P}_{\tau_1,\dots,\tau_N \sim \pi_j}[\psi^\star \neq j].$$

Finally, taking infimum over all hypothesis tests we obtain

$$\min_{\widehat{\pi}} \max_{\pi \in \mathcal{F}} \mathbb{P}\left[\rho(\pi, \widehat{\pi}) \geq \sqrt{\frac{D}{4N \cdot \log(\mathrm{e}^2/\gamma)}}\right] \geq \inf_{\psi} \max_{j=0,\ldots,M} \mathbb{P}_j[\psi \neq j] \triangleq p_{e,M}.$$

To lower bound the right-hand side, we apply Proposition 2.3 by Tsybakov (2008). Notice that the maximal $\varepsilon$-packing is $\varepsilon$-net (see Lemma 4.2.6 by Vershynin (2018)). Therefore, by Lemma 7 we have

$$\frac{1}{M} \sum_{i=1}^{M} \mathrm{KL}(\mathbb{P}_{\tau_1,\ldots,\tau_N \sim \pi_i} \| \mathbb{P}_{\tau_1,\ldots,\tau_N \sim \pi_0}) = \frac{N}{M} \sum_{i=1}^{M} \mathrm{KL}_{\mathrm{traj}}(\pi_i \| \pi_0)$$

$$\leq n \log(\mathrm{e}^2/\gamma) \frac{1}{M} \sum_{i=1}^{M} \rho^2(\pi_i, \pi_0) \leq D \triangleq \alpha_\star.$$

Thus by Proposition 2.3 by Tsybakov (2008)

$$p_{e,M} \geq \sup_{0 < \tau < 1}\left[\frac{\tau M}{1 + \tau M}\left(1 + \frac{\alpha_\star + \sqrt{\alpha_\star}}{\log(\tau)}\right)\right].$$

Next, we select $\tau_\star$ in a way such that

$$\frac{\alpha_\star + \sqrt{\alpha_\star/2}}{\log(\tau_\star)} = -\frac{1}{2} \iff \log(\tau_\star) = -\frac{1}{2}\left(\alpha_\star + \sqrt{\alpha_\star/2}\right).$$

Therefore we have

$$p_{e,M} \geq \frac{1}{2} \frac{\exp(\log(M) - 1/2(\alpha_\star + \sqrt{\alpha_\star/2}))}{1 + \exp(\log(M) - 1/2(\alpha_\star + \sqrt{\alpha_\star/2}))}.$$

Notice that the function $f(x) = \exp(x)/(1 + \exp(x))$ monotonically increasing, therefore it is enough to bound the expression under the exponent from below.

Let us assume that $\alpha_\star \geq 5 \iff D \geq 5$. Then we have

$$\log(M) - 1/2(\alpha^\star + \sqrt{\alpha_\star/2}) \geq \log(M+1) - \frac{2+\sqrt{2}}{4}\alpha_\star - \log(2)$$

$$\geq D \log\left(R\sqrt{\frac{4\log(\mathrm{e}^2/\gamma)N}{D}}\right) - D$$

$$\geq \frac{D}{2}\left(\log\left(\frac{4\log(\mathrm{e}^2/\gamma)R^2 \cdot N}{D}\right) - 2\right).$$

To show that the expression above is non-negative, it is enough to guarantee

$$\log(N) + \log(R^2/D) \geq 2 \iff N \geq \mathrm{e}^2 D/R^2.$$

Under this condition we have $p_{e,M} \geq 1/4$ concluding the statement. $\qquad\square$

### B.5.2 FINITE MDPS

For the case of finite MDPs, we additionally specialize the distributions $\mu_h$ as a uniform over $\mathcal{S}$ for all $h > 1$ and $\mu_1 = \delta_{s_1}$.

**Lemma 4.** *Let* $\Delta_{A,\gamma} = \{x \in \mathbb{R}^A : \sum_{i=1}^{n} x_i = 1, x_i \geq \gamma\}$ *with* $\gamma < 1/A$. *Then we have for any* $\varepsilon \in (0,1)$

$$\log \mathcal{P}(\varepsilon, \Delta_{A,\gamma}, d_{\mathcal{H}})| \geq (A-1)\log((1 - A\gamma)/(2\varepsilon)).$$

*Proof.* Let $S = \{x \in \mathbb{R}^A : \sum_{i=1}^{n} x_i^2 = 1, x_i \geq 0\}$. Then there is a mapping $\varphi \colon (S, \|\cdot\|_2) \to (\Delta_A, d_{\mathcal{H}})$ that defines an isometry between these two metric spaces:

$$\forall x, y \in S : \|x - y\|_2 = d_{\mathcal{H}}(\varphi(x), \varphi(y)), \quad \varphi(x) = \sqrt{x},$$

where the square root is applied component-wise. Therefore, it is enough to estimate the packing number of the preimage of $\Delta_{A,\gamma}$ that is defined as follows $S_\gamma(1) = \{x \in \mathbb{R}^A : \sum_{i=1}^A x_i^2 = 1, x_i \geq \sqrt{\gamma}\}$ with the same Euclidean metric.

The next step is to proceed with a shift $x \mapsto x + \sqrt{\gamma}$ that will be isometry between $S_\gamma(1)$ and $S_0(1 - A\gamma)$.

Next, we can lower bound the $\ell_2$-distance over the sphere by the $\ell_2$ distance over the first $A - 1$ coordinates and therefore it is enough to consider the packing number of $S_0^\circ(1 - A\gamma) = \mathcal{B}(0, 1) \cap \{y \in \mathbb{R}^{A-1} : \sum_{i=1}^{A-1} y_i^2 \leq 1 - A\gamma\}$.

Finally, we apply the volume argument. In particular, it is enough to compute the volume of $S_0'(1 - A\gamma)$. To do it, we notice that we can represent the ball of radius $1 - A\gamma$ by $2^{A-1}$ copy of $S_0'(1 - A\gamma)$. Thus

$$\text{vol}(S_0'(1 - A\gamma)) = \frac{(1 - A\gamma)^{A-1}}{2^{A-1}} \cdot \text{vol}(B_2^{A-1}).$$

Finally we have by Proposition 4.2.12 by Vershynin (2018)

$$\mathcal{P}(\varepsilon, \Delta_{A,\gamma}, d_\mathcal{H})| \geq \left(\frac{1 - A\gamma}{2\varepsilon}\right)^{A-1}.$$

$\square$

**Lemma 5.** *Let $(\mathcal{X}, \rho)_{i=1}^K$ be a metric space such that $\log |\mathcal{P}_\varepsilon(\mathcal{X}, \rho)| \geq d \log(R/\varepsilon)$ for $d \geq 1$. and define on the space $\mathcal{X}^K$ the following metric $\rho(x, y) = \frac{1}{K} \sum_{i=1}^K \rho(x_i, y_i)$. Then*

$$\log \mathcal{P}(\varepsilon, \mathcal{X}^K, \rho) \geq dK/2 \cdot \log(R/(8\varepsilon)).$$

*Proof.* Consider the maximal $\varepsilon$-separable set $P$ of the space $K$. This set could be considered as a finite alphabet of size $q \geq (R/\varepsilon)^d$. Let us consider the set $P^K$ as the set of words in alphabet $P$ of size $q$ of length $K$ with a Hoeffding distance. Then we notice that if there is two words $(x, y) \in P^K$ that have Hoeffding distance at least $\alpha K$ for some constant $\alpha \in (0, 1)$, then

$$\rho(x, y) = \frac{1}{K} \sum_i \rho(x_i, y_i) \geq \alpha\varepsilon.$$

Therefore, if we consider a $\alpha K$ separable set in $P^K$ in terms of Hoeffding distance, it will be automatically a $\alpha\varepsilon$-separable set in the original space $\mathcal{X}^K$. To find such a set we use the Gilbert–Varshamov bound from coding theory. As a result

$$\mathcal{P}(\alpha\varepsilon, P^K, \rho) \geq \frac{1}{\sum_{j=1}^{\lceil \alpha K \rceil} \binom{K}{j}(1 - 1/q)^j \cdot (1/q)^{K-j}}.$$

The denominator could be interpreted as follows: Let $X_1, \ldots, X_K$ be $\mathcal{B}er(1/q)$ random variables.

$$\sum_{j=1}^{\lceil \alpha K \rceil} \binom{K}{j}(1 - 1/q)^j \cdot (1/q)^{K-j} = \mathbb{P}\left[\sum_{i=1}^K X_i \geq (1 - \alpha)K\right].$$

To upper bound the last probability we can apply the Chernoff–Hoeffding theorem

$$\mathbb{P}\left[\frac{1}{K} \sum_{i=1}^K X_i \geq 1/q + (1 - \alpha - 1/q)\right] \leq \exp(-\text{kl}(1 - \alpha \| 1/q) \cdot K).$$

Take $\alpha = 1/2$, then we have

$$\text{kl}(1/2 \| 1/q) = \frac{1}{2} \log\left(\frac{q}{2}\right) + \frac{1}{2} \log\left(\frac{q}{q-1}\right) - \frac{1}{2} \log(2) \geq \frac{1}{2} \log\left(\frac{q}{4}\right).$$

Thus, we have since $d \geq 1$

$$\mathcal{P}(\varepsilon/2, P^K, \rho) \geq \exp(K/2 \cdot \log(q/4)) \geq \exp(dK/2 \cdot \log(R/(4\varepsilon))).$$

By rescaling $\varepsilon$ we conclude the statement. $\square$

**Corollary 4.** *Assume that* $\gamma \leq 1/(2A)$*. Let us define* $\mathcal{F}_h = \Delta_{A,\gamma}^S$ *and* $\mathcal{F} = \mathcal{F}_1 \times \ldots \mathcal{F}_H$*. Then Assumption 5 holds with constants* $d_h = (A-1)S/2$ *for all* $h > 1$*,* $d_1 = (A-1)$ *and* $R = 1/32$*.*

*As a result, as soon as* $H \geq 2, A \geq 2$*,* $HSA \geq 40$ *and* $n \geq 512\mathrm{e}^2 HSA$ *the following minimax lower bound holds*

$$\min_{\widehat{\pi}} \max_{\pi \in \mathcal{F}} \mathbb{E}_{\tau_1, \ldots, \tau_N \sim \pi}[\mathrm{KL}_{\mathrm{traj}}(\pi \| \widehat{\pi})] \geq \frac{HSA}{128N \log(\mathrm{e}^2/\gamma)}.$$

### B.5.3 TECHNICAL LEMMAS

**Lemma 6.** *Let* $\{(\mathcal{X}_i, \rho_i)\}_{i=1,\ldots,K}$ *be a collection of relaxed pseudometric spaces that satisfy*

$$\forall \varepsilon \in (0,1) : \log \mathcal{P}(\varepsilon, \mathcal{X}_i, \rho_i) \geq d_i \log(R/\varepsilon)$$

*for some constants* $0 \leq d_1 \leq d_2$*, Then the product space* $\mathcal{X} = \mathcal{X}_1 \times \ldots \mathcal{X}_K$ *with a pseudometric* $\rho((x_1, \ldots, x_K), (y_1, \ldots, y_K)) = \sqrt{\sum_{i=1}^K \rho_i^2(x_i, y_i)}$ *satisfies*

$$\forall \varepsilon \in (0,1) : \log \mathcal{P}(\varepsilon, \mathcal{X}, \rho) \geq \sum_{i=1}^K d_i \log(R/\varepsilon).$$

*Proof.* Let $P_1, \ldots, P_K$ be a maximal $\varepsilon$-separated set in the corresponding spaces $\mathcal{M}_1, \ldots, \mathcal{M}_K$. Then we want to show that the set $P_1 \times \ldots \times P_K$ is also $\varepsilon$-separated set in the product space.

Let $\mathbf{x} \neq \mathbf{x}'$ be two point in $P$. Then

$$\rho(\mathbf{x}, \mathbf{x}') = \sqrt{\sum_{i=1}^K \rho_i^2(x_i, x_i')} \geq \varepsilon$$

since $\mathbf{x}$ and $\mathbf{x}'$ are different in at least one coordinate. As a result, we have

$$\log \mathcal{P}(\varepsilon, \mathcal{M}_i, \rho_i)| \geq \sum_{i=1}^K \log \mathcal{P}(\varepsilon, \mathcal{M}_i, \rho_i) \geq \sum_{i=1}^K d_i \log(R/\varepsilon).$$

$\square$

**Lemma 7.** *Let* $\pi, \pi' \in \Pi_\gamma$*. Let* $\rho$ *be an averaged Hellinger distance distance defined in* (6)*. Then the following inequality holds*

$$\rho^2(\pi, \pi') \leq \mathrm{KL}_{\mathrm{traj}}(\pi \| \pi') \leq \log(\mathrm{e}^2/\gamma)\rho^2(\pi, \pi').$$

*Proof.* It is enough to show that for two measures $p, q \in \Delta_n$ such that $\min_i p_i/q_i \geq \gamma$ the following holds

$$d_{\mathcal{H}}^2(p, q) \leq \mathrm{KL}(p \| q) \leq \log(\mathrm{e}^2/\gamma)d_{\mathcal{H}}^2(p, q).$$

The lower bound holds by Lemma 2.4 of Tsybakov (2008), and the upper bound holds by Lemma 4 of Yang & Barron (1998).

$\square$

## B.6 IMITATION LEARNING GUARANTEES

In this appendix, we present guarantees that give behavior cloning procedure in the setting of imitation learning for finite MDPs and compare obtained results to (Ross & Bagnell, 2010; Rajaraman et al., 2020).

**General expert**   Using Pinsker inequality in the space of trajectories (see Lemma 9) and the fact the expert policy is $\varepsilon_\mathrm{E}$-optimal we deduce the following bound on the optimality gap of the behavior cloning policy with probability at least $1 - \delta$

$$V_1^\star(s_1) - V_1^{\pi^{\mathrm{BC}}}(s_1) \le \varepsilon_\mathrm{E} + \widetilde{\mathcal{O}}\left(\sqrt{\frac{SAH^3}{N^\mathrm{E}}}\right).$$

We remark that we obtain a similar rate as in BPI, see for example Ménard et al. (2021), where instead of observing $N^\mathrm{E}$ demonstrations we collect the same number of trajectories (observing also the rewards) by interacting sequentially with the MDPs. This seems a bit counter-intuitive since we expect to learn faster by directly observing the expert.

However, for the deterministic expert, we get an improved rate using the following variance-aware Pinkser inequality.

**Lemma 8.** *Let $\pi$ and $\pi'$ be arbitrary policies. Then the following upper bound holds*

$$V_1^\pi(s_1) - V_1^{\pi'}(s_1) \le \sqrt{4\mathbb{E}_\pi\left[\sum_{h=1}^H \mathrm{Var}_\pi Q_h^{\pi'}(s_h)\right] \cdot \mathrm{KL}_\mathrm{traj}(\pi\|\pi') + 4H\,\mathrm{KL}_\mathrm{traj}(\pi\|\pi')}.$$

*Proof.* Let us start from Lemma 11 and Lemma 32.

$$
\begin{aligned}
V_1^\pi(s_1) - V_1^{\pi'}(s_1) &= \mathbb{E}_\pi\left[\sum_{h=1}^H [\pi_h - \pi'_h]Q_h^{\pi'}(s_h)\right]\\
&\le \mathbb{E}_\pi\left[\sum_{h=1}^H \sqrt{2\mathrm{Var}_{\pi'}Q_h^{\pi'}(s_h)\cdot\mathrm{KL}(\pi_h(s_h)\|\pi'_h(s_h))} + \frac{2H}{3}\mathrm{KL}(\pi_h(s_h)\|\pi'_h(s_h))\right]\\
&\le \sqrt{2\mathbb{E}_\pi\left[\sum_{h=1}^H \mathrm{Var}_{\pi'}Q_h^{\pi'}(s_h)\right]}\cdot\sqrt{\mathrm{KL}_\mathrm{traj}(\pi\|\pi')} + \frac{2H}{3}\mathrm{KL}_\mathrm{traj}(\pi\|\pi'),
\end{aligned}
$$

where in the last line we have applied Cauchy-Schwartz inequality. Next we apply Lemma 33 and obtain

$$\mathbb{E}_\pi\left[\sum_{h=1}^H \mathrm{Var}_{\pi'}Q_h^{\pi'}(s_h)\right] \le 2\mathbb{E}_\pi\left[\sum_{h=1}^H \mathrm{Var}_\pi Q_h^{\pi'}(s_h)\right] + 4H^2\,\mathrm{KL}_\mathrm{traj}(\pi\|\pi').$$

By inequality $\sqrt{a+b} \le \sqrt{a} + \sqrt{b}$ for any $a, b \ge 0$ we have

$$V_1^\pi(s_1) - V_1^{\pi'}(s_1) \le \sqrt{4\mathbb{E}_\pi\left[\sum_{h=1}^H \mathrm{Var}_\pi Q_h^{\pi'}(s_h)\right]\cdot\mathrm{KL}_\mathrm{traj}(\pi\|\pi') + 4H\,\mathrm{KL}_\mathrm{traj}(\pi\|\pi')}.$$

$\square$

**Deterministic expert**   If we assume that the expert policy is deterministic, for example, a deterministic optimal policy, then we can improve the bound on the optimality gap since the variance term in Lemma 8 is zero,

$$V_1^\star(s_1) - V_1^{\pi^{\mathrm{BC}}}(s_1) \le \varepsilon_\mathrm{E} + \widetilde{\mathcal{O}}\left(\frac{SAH^2}{N^\mathrm{E}}\right).$$

Ross & Bagnell (2010) also consider behavior cloning with a deterministic expert and provide a bound in terms of the classification-type error of the behavior cloning policy to imitate the expert

$$V_1^\star(s_1) - V_1^{\pi^{\mathrm{BC}}}(s_1) \le \varepsilon_\mathrm{E} + e_\mathrm{E}(\pi^{\mathrm{BC}}) \text{ where } e_\mathrm{E}(\pi^{\mathrm{BC}}) = \frac{1}{H}\mathbb{E}^{\pi^{\mathrm{BC}}}\left[\sum_{h=1}^H \mathbb{1}\{\pi^\mathrm{E}(a_h|s_h) \ne 1\}\right].$$

We can easily recover our bound on the optimality gap of the behavior cloning policy from their bound by noting that $e_\mathrm{E}(\pi^{\mathrm{BC}}) \le \mathrm{KL}_\mathrm{traj}(\pi^\mathrm{E}\|\pi^{\mathrm{BC}})/H$, see Lemma 10 for a proof. In particular, we

also remark that the bound scales quadratically with the horizon $H$ but also linearly with the number of actions and states $SA$.

By comparing this bound to the lower bound in Theorem 1.1 of Rajaraman et al. (2020) we see that it is optimal in its dependence on $S, H$ and $N$. Additional dependence on a number of actions comes from the fact that our behavior cloning algorithm always outputs stochastic policy and obtains additional dependence on a number of actions.

We would like to underline that the bound in Lemma 8 directly does not give any insights on the performance of the algorithm in the case of non-deterministic optimal expert, whereas Rajaraman et al. (2020) provides $\widetilde{\mathcal{O}}(SH^2/N^{\mathrm{E}})$ guarantees using a non-regularized behavior cloning algorithm. It is connected to the fact that for the optimal policy, it is enough to determine the subset of the support of $\pi^\star(s)$ to achieve the policy with the same value.

Finally, we would like to emphasize that our approach is directly generalized to arbitrary parametric function approximation setting whereas the approach of Rajaraman et al. (2020) could be applied only in the setting of finite MDPs.

### B.6.1 TECHNICAL LEMMAS FOR IMITATION LEARNING

**Lemma 9.** *Let* $\mathcal{M} = (\mathcal{S}, \mathcal{A}, \{p_h\}_{h=1}^H, \{r_h\}_{h=1}^H, s_1)$ *be a finite MDP and let* $\pi$ *and* $\pi'$ *be two any policies in* $\mathcal{M}$. *Then*

$$V_1^\pi(s_1) - V_1^{\pi'}(s_1) \leq H\sqrt{\mathrm{KL}_{\mathrm{traj}}(\pi\|\pi')/2}.$$

*Proof.* Let us define the trajectory distribution $q^\pi(\tau)$ for $\tau = (s_1, a_1, \ldots, s_H, a_H)$. Then by the chain rule for KL-divergence we have $\mathrm{KL}(q^\pi\|q^{\pi'}) = \mathrm{KL}_{\mathrm{traj}}(\pi\|\pi')$.

Since $r_h \in [0, 1]$, we may apply a variational formula for total variation distance and Pinkser's inequality

$$\left|V_1^\pi(s_1) - V_1^{\pi'}(s_1)\right| \leq \left|\sum_{h=1}^H \mathbb{E}_\pi[r_h(s_h, a_h)] - \mathbb{E}_{\pi'}[r_h(s_h, a_h)]\right|$$

$$\leq H\,\mathrm{TV}(q^\pi, q^{\pi'}) \leq H\sqrt{\mathrm{KL}_{\mathrm{traj}}(\pi\|\pi')/2}.$$

$\square$

Let us assume that a policy $\pi$ is deterministic, then

$$e_\pi(\pi') = \frac{1}{H}\sum_{h=1}^H \mathbb{E}_\pi\left[\sum_{a\in\mathcal{A}}\pi'(a|s_h)\mathbb{1}\{a \neq \pi(s_h)\}\right].$$

Notice that

$$\sum_{a\in\mathcal{A}}\pi'(a|s_h)\mathbb{1}\{a \neq \pi(s_h)\} = \frac{1}{2}\sum_{a\in\mathcal{A}}|\pi'(a|s_h) - \pi(a|s_h)| = \mathrm{TV}(\pi(s_h), \pi'(s_h)).$$

Therefore this quantity could be decomposed as follows

$$e_\pi(\pi') = \frac{1}{H}\sum_{h=1}^H \mathbb{E}_\pi[\mathrm{TV}(\pi(s_h), \pi'(s_h))].$$

**Lemma 10.** *Let* $\pi$ *be a deterministic policy and* $\pi'$ *be any policy. Then*

$$e_\pi(\pi') \leq \frac{\mathrm{KL}_{\mathrm{traj}}(\pi\|\pi')}{H}.$$

*Proof.* If the policy $\pi$ is deterministic, then

$$\mathrm{KL}(\pi_h(s_h)\|\pi_h'(s_h)) = \log\left(\frac{1}{\pi_h'(a_h|s_h)}\right) = \log\left(\frac{1}{\pi_h'(\pi_h(s_h)|s_h)}\right)$$

$$= \log\left(\frac{1}{1 - \sum_{a\in\mathcal{A}}\pi_h'(a|s_h)\mathbb{1}\{a \neq \pi_h(s_h)\}}\right)$$

$$= -\log(1 - \mathrm{TV}(\pi_h(s_h), \pi_h'(s_h))).$$

By an inequality $\log(1 - x) \leq -x$ for any $x > 0$ we have $\mathrm{KL}(\pi_h(s_h)\|\pi_h'(s_h)) \geq \mathrm{TV}(\pi_h(s_h), \pi_h'(s_h))$. $\square$

The following lemma is known as performance-difference lemma (see e.g. Kakade & Langford (2002) for a statement in the discounted setting). We provide proof for completeness.

**Lemma 11.** *Let $\pi$ and $\pi'$ be arbitrary policies. Then the following decomposition holds*

$$V_1^\pi(s_1) - V_1^{\pi'}(s_1) = \mathbb{E}_\pi\left[\sum_{h=1}^{H}[\pi_h - \pi_h']Q_h^{\pi'}(s_h)\right].$$

*Proof.* Let us proceed by backward induction over $h \in [H]$. We want to show the following bound

$$V_h^\pi(s) - V_h^{\pi'}(s) = \mathbb{E}_\pi\left[\sum_{h'=1}^{H}[\pi_{h'} - \pi_{h'}']Q_{h'}^{\pi'}(s_{h'})|s_h = s\right].$$

For $h = H + 1$ both sides of the equation above are equal to zero. Let us assume that the statement holds for any $h' > h$. Then we have

$$V_h^\pi(s) - V_h^{\pi'}(s) = \pi Q_h^\pi(s) - \pi'Q_h^{\pi'}(s) = \pi[Q_h^\pi - Q_h^{\pi'}](s) + [\pi - \pi']Q^{\pi'}(s)$$

$$= \mathbb{E}_\pi\left[V_{h+1}^\pi(s_{h+1}) - V_{h+1}^{\pi'}(s_{h+1}) + [\pi - \pi']Q^{\pi'}(s_h)|s_h = s\right].$$

By the induction hypothesis and the tower property of mathematical expectation, we conclude the statement. $\square$

## C    PROOF FOR DEMONSTRATION-REGULARIZED RL

**Theorem** (Restatement of Theorem 2). *Assume that there are an expert policy $\pi^{\mathrm{E}}$ such that $V_1^{\star}(s_1) - V_1^{\pi^{\mathrm{E}}}(s_1) \leq \varepsilon_{\mathrm{E}}$ and a behavior cloning policy $\pi^{\mathrm{BC}}$ satisfying $\sqrt{\mathrm{KL}_{\mathrm{traj}}(\pi^{\mathrm{E}} \| \pi^{\mathrm{BC}})} \leq \varepsilon_{\mathrm{KL}}$. Let $\pi^{\mathrm{RL}}$ be $\varepsilon_{\mathrm{RL}}$-optimal policy in $\lambda$-regularized MDP with respect to $\pi^{\mathrm{BC}}$, that is, $V_{\pi^{\mathrm{BC}},\lambda,1}^{\star}(s_1) - V_{\pi^{\mathrm{BC}},\lambda,1}^{\pi^{\mathrm{RL}}} \leq \varepsilon_{\mathrm{RL}}$. Then $\pi^{\mathrm{RL}}$ fulfills*

$$V_1^{\star}(s_1) - V_1^{\pi^{\mathrm{RL}}}(s_1) \leq \varepsilon_{\mathrm{E}} + \varepsilon_{\mathrm{RL}} + \lambda \varepsilon_{\mathrm{KL}}^2.$$

*In particular, under the choice $\lambda^{\star} = \varepsilon_{\mathrm{RL}}/\varepsilon_{\mathrm{KL}}^2$, the policy $\pi^{\mathrm{BC}}$ is $(2\varepsilon_{\mathrm{RL}} + \varepsilon_{\mathrm{E}})$-optimal in the original (non-regularized) MDP.*

*Proof.* We start from the following observation that comes from the assumption on expert policy and a definition of regularized value

$$V_1^{\star}(s_1) - V_1^{\pi^{\mathrm{RL}}}(s_1) \leq \varepsilon_{\mathrm{E}} + V_1^{\pi^{\mathrm{E}}}(s_1) - V_1^{\pi^{\mathrm{RL}}}(s_1) \leq \varepsilon_{\mathrm{E}} + V_{\pi^{\mathrm{BC}},\lambda,1}^{\pi^{\mathrm{E}}}(s_1) + \lambda \, \mathrm{KL}_{\mathrm{traj}}(\pi^{\mathrm{E}} \| \pi^{\mathrm{BC}})$$
$$- V_{\pi^{\mathrm{BC}},\lambda,1}^{\pi^{\mathrm{RL}}}(s_1) - \lambda \, \mathrm{KL}_{\mathrm{traj}}(\pi^{\mathrm{RL}} \| \pi^{\mathrm{BC}}) \leq \varepsilon_{\mathrm{E}} + \varepsilon_{\mathrm{RL}} + \lambda \varepsilon_{\mathrm{KL}}^2,$$

where in the last inequality we apply assumptions on $\pi^{\mathrm{BC}}$ and $\pi^{\mathrm{RL}}$. □

Here for completeness we present the proof of Theorem 3.

**Theorem** (Restatement of Theorem 3). *For all $\varepsilon > 0$, $\delta \in (0,1)$, the* `UCBVI-Ent+`/`LSVI-UCB-Ent` *algorithms defined in Appendix D.4/Appendix E.4 are $(\varepsilon, \delta)$-PAC for the regularized BPI with sample complexity*

$$\mathcal{C}(\varepsilon, \delta) = \widetilde{\mathcal{O}}\left(\frac{H^5 S^2 A}{\lambda \varepsilon}\right) \text{ (finite)} \qquad \mathcal{C}(\varepsilon, \delta) = \widetilde{\mathcal{O}}\left(\frac{H^5 d^2}{\lambda \varepsilon}\right) \text{ (linear).}$$

*Additionally, assume that the expert policy is $\varepsilon_{\mathrm{E}} = \varepsilon/2$-optimal and satisfies Assumption 3 in the linear case. Let $\pi^{\mathrm{BC}}$ be the behavior cloning policy obtained using corresponding function sets described in Section 3. Then demonstration-regularized RL based on* `UCBVI-Ent+`/`LSVI-UCB-Ent` *with parameters $\varepsilon_{\mathrm{RL}} = \varepsilon/4$, $\delta_{\mathrm{RL}} = \delta/2$ and $\lambda = \widetilde{\mathcal{O}}\big(N^{\mathrm{E}}\varepsilon/(SAH)\big)$/$\widetilde{\mathcal{O}}\big(N^{\mathrm{E}}\varepsilon/(dH)\big)$ is $(\varepsilon, \delta)$-PAC for BPI with demonstration in finite / linear MDPs and has sample complexity of order*

$$\mathcal{C}(\varepsilon, N^{\mathrm{E}}, \delta) = \widetilde{\mathcal{O}}\left(\frac{H^6 S^3 A^2}{N^{\mathrm{E}}\varepsilon^2}\right) \text{ (finite)} \qquad \mathcal{C}(\varepsilon, N^{\mathrm{E}}, \delta) = \widetilde{\mathcal{O}}\left(\frac{H^6 d^3}{N^{\mathrm{E}}\varepsilon^2}\right) \text{ (linear).}$$

*Proof.* The first part of the statement is a combination of Theorem 5 and Theorem 6. The second part of the statement follows from an upper bound on $\varepsilon_{\mathrm{KL}}^2$ by a behavior cloning (see Appendix B.2 and Appendix B.3) and Theorem 2. □

## D  BEST POLICY IDENTIFICATION IN REGULARIZED FINITE MDPs

In this appendix, we present and analyze the `UCBVI-Ent+` algorithm for regularized BPI.

### D.1  PRELIMINARIES

We first detail the general setting of KL-regularized MDPs.

Given some reference policy $\widetilde{\pi}$ and some regularization parameter $\lambda > 0$, instead of looking at the usual value function of a policy $\pi$ we consider the trajectory Kullback-Leibler divergence regularized value function $V_{\widetilde{\pi},\lambda,1}^{\pi}(s_1) \triangleq V_1^{\pi}(s_1) - \lambda \mathrm{KL}_{\mathrm{traj}}(\pi, \widetilde{\pi})$. In this case, the value function of the policy $\pi$ is penalized for moving too far from the reference policy $\widetilde{\pi}$. Interestingly, we can compute the value of policy $\pi$ and the optimal value thanks to the regularized Bellman equations

$$
\begin{aligned}
Q_{\widetilde{\pi},\lambda,h}^{\pi}(s,a) &= r_h(s,a) + p_h V_{\widetilde{\pi},\lambda,h+1}^{\pi}(s,a)\,,\\
V_{\widetilde{\pi},\lambda,h}^{\pi}(s) &= \pi_h Q_{\widetilde{\pi},\lambda,h}^{\pi}(s) - \lambda \, \mathrm{KL}(\pi_h(s)\|\widetilde{\pi}_h(s))\,,\\
Q_{\widetilde{\pi},\lambda,h}^{\star}(s,a) &= r_h(s,a) + p_h V_{\widetilde{\pi},\lambda,h+1}^{\star}(s,a)\,,\\
V_{\widetilde{\pi},\lambda,h}^{\star}(s) &= \max_{\pi \in \Delta_A}\big\{\pi Q_{\widetilde{\pi},\lambda,h}^{\star}(s) - \lambda \, \mathrm{KL}(\pi_h(s)\|\widetilde{\pi}_h(s))\big\}\,,\\
\pi_{\widetilde{\pi},\lambda,h}^{\star}(s) &= \arg\max_{\pi \in \Delta_A}\big\{\pi Q_{\widetilde{\pi},\lambda,h}^{\star}(s) - \lambda \, \mathrm{KL}(\pi_h(s)\|\widetilde{\pi}_h(s))\big\}\,,
\end{aligned}
\tag{7}
$$

where $V_{\widetilde{\pi},\lambda,H+1}^{\pi} = V_{\widetilde{\pi},\lambda,H+1}^{\star} = 0$. Note that for $\pi$ the uniform policy we recover the entropy-regularized Bellman equations.

Next we define a convex conjugate to $\lambda \, \mathrm{KL}(\cdot\|\widetilde{\pi}_h(s))$ as $F_{\widetilde{\pi}_h(s),\lambda,h} \colon \mathbb{R}^{\mathcal{A}} \to \mathbb{R}$

$$
F_{\widetilde{\pi}_h(s),\lambda,h}(x) = \max_{\pi \in \Delta_{\mathcal{A}}}\{\langle \pi, x \rangle - \lambda \, \mathrm{KL}(\pi\|\widetilde{\pi}_h(s))\} = \lambda \log\left(\sum_{a \in \mathcal{A}} \widetilde{\pi}_h(a|s)\exp\{x_a/\lambda\}\right).
$$

and, with a sight abuse of notation extend the action of this function to the $Q$-function as follows

$$
V_{\widetilde{\pi},\lambda,h}^{\star}(s) = F_{\widetilde{\pi}_h(s),\lambda,h}(Q_{\widetilde{\pi},\lambda,h}^{\star}(s,\cdot)) = \max_{\pi \in \Delta_{\mathcal{A}}}\{\pi Q_{\widetilde{\pi},\lambda,h}^{\star}(s) - \lambda \, \mathrm{KL}(\pi\|\widetilde{\pi}_h(s))\}.
$$

Thanks to the fact that the norm of gradients of $\mathrm{KL}(\pi|\widetilde{\pi}_h(s))$ tends to infinity as $\pi$ tends to a border of simplex, we have an exact formula for the optimal policy by Fenchel-Legendre transform

$$
\pi_{\widetilde{\pi},\lambda,h}^{\star}(s) = \arg\max_{\pi \in \Delta_{\mathcal{A}}}\{\pi Q_{\widetilde{\pi},\lambda,h}^{\star}(s) - \lambda \, \mathrm{KL}(\pi\|\widetilde{\pi}_h(s))\} = \nabla F_{\widetilde{\pi}_h(s),\lambda,h}(Q_{\widetilde{\pi},\lambda,h}^{\star}(s,\cdot)).
$$

Notice that we have $\nabla F_{\widetilde{\pi}_h(s),\lambda,h}(Q_{\widetilde{\pi},\lambda,h}^{\star}(s,\cdot)) \in \Delta_{\mathcal{A}}$ since the gradient of $\Phi$ diverges on the boundary of $\Delta_{\mathcal{A}}$.

Finally, it is known that the smoothness property of $F_{\widetilde{\pi}_h(s),\lambda,h}$ plays a key role in reduced sample complexity for planning in regularized MDPs (Grill et al., 2019). For our general setting we have that since $\lambda \, \mathrm{KL}(\cdot\|\widetilde{\pi}_h(s))$ is $\lambda$-strongly convex with respect to $\|\cdot\|_1$, then $F_{\widetilde{\pi}_h(s),\lambda,h}$ is $1/\lambda$-strongly smooth with respect to the dual norm $\|\cdot\|_\infty$

$$
F_{\widetilde{\pi}_h(s),\lambda,h}(x) \le F_{\widetilde{\pi}_h(s),\lambda,h}(x') + \langle \nabla F_{\widetilde{\pi}_h(s),\lambda,h}(x'), x - x' \rangle + \frac{1}{2\lambda}\|x - x'\|_\infty^2.
$$

We notice that KL-divergence is always non-negative and, moreover, $V_{\widetilde{\pi},\lambda,h}^{\star}(s) \ge 0$ for any $s$ since the value of the reference policy $\widetilde{\pi}$ is non-negative.

### D.2  ALGORITHM DESCRIPTION

In this appendix we present the `UCBVI-Ent+` algorithm, a modification of the algorithm `UCBVI-Ent` proposed by Tiapkin et al. (2023), that achieves better rates in the tabular setting. The `UCBVI-Ent+` algorithm works by sampling trajectory according to an exploratory version of an optimistic solution of the regularized MDP and is characterized by the following rules.

**Sampling rule**  To obtain the sampling rule at episode $t$, we first compute a policy $\bar{\pi}^t$ by optimistic planning in a regularized MDP,

$$\overline{Q}_h^t(s,a) = \mathrm{clip}\Big(r_h(s,a) + \widehat{p}_h^t \overline{V}_{h+1}^t(s,a) + b_h^{p,t}(s,a), 0, H\Big),$$

$$\overline{V}_h^t(s) = \max_{\pi \in \Delta_A}\Big\{\pi\overline{Q}_h^t(s) - \lambda \,\mathrm{KL}(\pi \| \widetilde{\pi}_h(s))\Big\}, \tag{8}$$

$$\bar{\pi}_h^{t+1}(s) = \underset{\pi \in \Delta_A}{\arg\max}\Big\{\pi\overline{Q}_h^t(s) - \lambda \,\mathrm{KL}(\pi \| \widetilde{\pi}_h(s))\Big\},$$

with $\overline{V}_{H+1}^t = 0$ by convention, where $\widehat{p}^t$ is an estimate of the transition probabilities defined in Appendix D.3 and $b^{p,t}$ some bonus term, defined in (10), Appendix D.4. It takes into account an estimation error for transition probabilities. Then we define a family of policies aimed to explore actions for which $Q$-value is not well estimated at a particular step. That is, for $h' \in [0, H]$, the policy $\pi^{t,(h')}$ first follows the optimistic policy $\bar{\pi}^t$ until step $h$ where it selects an action leading to the largest confidence interval for the optimal $Q$-value,

$$\pi_h^{t,(h')}(a|s) = \begin{cases} \pi_h^{t,(h')}(a|s) = \bar{\pi}_h^t(a|s) & \text{if } h \neq h', \\ \pi_h^{t,(h')}(a|s) = \mathbb{1}\Big\{a \in \arg\max_{a' \in \mathcal{A}}(\overline{Q}_h^t(s,a') - \underline{Q}_h^t(s,a'))\Big\} & \text{if } h = h', \end{cases}$$

where $\underline{Q}^t$ is a lower bound on the optimal regularized $Q$-value function, see Appendix D.4. In particular, for $h' = 0$ we have $\pi^{t,(0)} = \bar{\pi}^t$. The sampling rule is obtained by picking up uniformly at random one policy among the family $\pi^t = \pi^{t,(h')}$, $h' \sim \mathcal{U}\mathrm{nif}[0, H]$. Note that it is equivalent to sampling from a uniform mixture policy $\pi^{\mathrm{mix},t}$ over all $h' \in [0, H]$.

**Stopping rule and decision rule**  To define the stopping rule, we first recursively build an upper-bound on the difference between the value of the optimal policy and the value of the current optimistic policy $\bar{\pi}^t$,

$$W_h^t(s,a) = \Big(1 + \frac{1}{H}\Big)\widehat{p}_h^t G_{h+1}^t(s) + b_h^{\mathrm{gap},t}(s,a),$$

$$G_h^t(s) = \mathrm{clip}\Big(\bar{\pi}_h^{t+1} W_h^t(s) + \frac{1}{2\lambda}\max_{a \in \mathcal{A}}\big(\overline{Q}_h^t(s,a) - \underline{Q}_h^t(s,a)\big)^2, 0, H\Big), \tag{9}$$

where $b_h^{\mathrm{gap},t}$ is a bonus defined in (12), Appendix D.4, $\underline{V}^t$ is a lower-bound on the optimal value function defined in Appendix D.4 and $G_{H+1}^t = 0$ by convention. Then the stopping time $\iota = \inf\{t \in \mathbb{N} : G_1^t(s_1) \leq \varepsilon\}$ corresponds to the first episode when this upper-bound is smaller than $\varepsilon$. At this episode, we return the policy $\widehat{\pi} = \bar{\pi}^\iota$.

The complete procedure is described in Algorithm 3.

---

**Algorithm 3** `UCBVI-Ent+`

---

1: **Input:** Target precision $\varepsilon$, target probability $\delta$, bonus functions $b^t, b^{t,\mathrm{KL}}$.
2: **while** true **do**
3:     Compute $\bar{\pi}^t$ by optimistic planning with (8).
4:     Compute bound on the gap $G_1^t(s,a)$ with (9).
5:     **if** $G_1^t(s_1) \leq \varepsilon$ **then break**
6:     Sample $h' \sim \mathcal{U}\mathrm{nif}[H]$ and set $\pi^t = \pi^{t,(h')}$.
7:     **for** $h \in [H]$ **do**
8:         Play $a_h^t \sim \pi_h^t(s_h^t)$
9:         Observe $s_{h+1}^t \sim p_h(s_h^t, a_h^t)$
10:     **end for**
11:     Update transition estimates $\widehat{p}^t$.
12: **end while**
13: **Output** policy $\widehat{\pi} = \bar{\pi}^t$.

---

### D.3 CONCENTRATION EVENTS

We first define an estimate of the transition kernel. The number of times the state action-pair $(s, a)$ was visited in step $h$ in the first $t$ episodes are $n_h^t(s, a) \triangleq \sum_{i=1}^t \mathbb{1}\{(s_h^i, a_h^i) = (s, a)\}$. Let $n_h^t(s'|s, a) \triangleq \sum_{i=1}^t \mathbb{1}\{(s_h^i, a_h^i, s_{h+1}^i) = (s, a, s')\}$ be the number of transitions from $s$ to $s'$ at step $h$. The empirical distribution is defined as $\widehat{p}_h^t(s'|s, a) = n_h^t(s'|s, a)/n_h^t(s, a)$ if $n_h^t(s, a) > 0$ and $\widehat{p}_h^t(s'|s, a) \triangleq 1/A$ for all $s' \in \mathcal{S}$ else.

Following the ideas of Ménard et al. (2021), we define the following concentration events. First, we define pseudo-counts as a sum of conditional expectations of the random variables that correspond to counts

$$\overline{n}_h^t(s, a) = \sum_{i=1}^t d_h^{\widetilde{\pi}^t}(s, a),$$

where $\pi^{\mathrm{mix},t}$ is a mixture policy played on $t$-th step. In particular, we have

$$d_h^{\pi^{\mathrm{mix},t}}(s, a) = \frac{1}{H+1} \sum_{h'=0}^H d_h^{\pi^{t,(h')}}(s, a),$$

where $\pi_h^{t,(h')}(s, a)$ is a greedy-modified policy $\bar{\pi}^t$ in $h'$-step:

$$\pi_h^{t,(h')}(a|s) = \begin{cases} \pi_h^{t,(h')}(a|s) = \bar{\pi}_h^t(a|s) & \text{if } h \neq h' \\ \pi_h^{t,(h')}(a|s) = \mathbb{1}\{a = \arg\max_{a' \in \mathcal{A}}(\overline{Q}_h^t(s, a') - \underline{Q}_h^t(s, a'))\} & \text{if } h = h' \end{cases},$$

Let $\beta^{\mathrm{KL}}, \beta^{\mathrm{cnt}} : (0, 1) \times \mathbb{N} \to \mathbb{R}_+$ be some functions defined later on in Lemma 12. We define the following favorable events

$$\mathcal{E}^{\mathrm{KL}}(\delta) \triangleq \left\{ \forall t \in \mathbb{N}, \forall h \in [H], \forall (s, a) \in \mathcal{S} \times \mathcal{A} : \mathrm{KL}(\widehat{p}_h^t(s, a) \| p_h(s, a)) \leq \frac{\beta^{\mathrm{KL}}(\delta, n_h^t(s, a))}{n_h^t(s, a)} \right\},$$

$$\mathcal{E}^{\mathrm{cnt}}(\delta) \triangleq \left\{ \forall t \in \mathbb{N}, \forall h \in [H], \forall (s, a) \in \mathcal{S} \times \mathcal{A} : n_h^t(s, a) \geq \frac{1}{2}\overline{n}_h^t(s, a) - \beta^{\mathrm{cnt}}(\delta) \right\},$$

We also introduce an intersection of these events of interest, $\mathcal{G}(\delta) \triangleq \mathcal{E}^{\mathrm{KL}}(\delta) \cap \mathcal{E}^{\mathrm{cnt}}(\delta)$. We prove that for the right choice of the functions $\beta^{\mathrm{KL}}, \beta^{\mathrm{cnt}}$ the above events hold with high probability.

**Lemma 12.** *For any $\delta \in (0, 1)$ and for the following choices of functions $\beta$,*

$$\beta^{\mathrm{KL}}(\delta, n) \triangleq \log(2SAH/\delta) + S\log(\mathrm{e}(1 + n)),$$
$$\beta^{\mathrm{cnt}}(\delta) \triangleq \log(2SAH/\delta),$$

*it holds that*

$$\mathbb{P}[\mathcal{E}^{\mathrm{KL}}(\delta)] \geq 1 - \delta/2, \quad \mathbb{P}[\mathcal{E}^{\mathrm{cnt}}(\delta)] \geq 1 - \delta/2,$$

*In particular, $\mathbb{P}[\mathcal{G}(\delta)] \geq 1 - \delta$.*

*Proof.* Applying Theorem 9 and the union bound over $h \in [H], (s, a) \in \mathcal{S} \times \mathcal{A}$ we get $\mathbb{P}[\mathcal{E}^{\mathrm{KL}}(\delta)] \geq 1 - \delta/2$.

By Theorem 10 and union bound, $\mathbb{P}[\mathcal{E}^{\mathrm{cnt}}(\delta)] \geq 1 - \delta/2$. The union bound over three prescribed events concludes $\mathbb{P}[\mathcal{G}_H(\delta)] \geq 1 - \delta$ and $\mathbb{P}[\mathcal{G}_B(\delta)] \geq 1 - \delta$. $\quad\square$

**Lemma 13.** *Assume conditions of Lemma 12. Then on event $\mathcal{E}^{\mathrm{KL}}(\delta)$, for any $f : \mathcal{S} \to [0, H], t \in \mathbb{N}, h \in [H], (s, a) \in \mathcal{S} \times \mathcal{A}$*

$$[p_h - \widehat{p}_h^t]f(s, a) \leq \sqrt{\frac{2H^2\beta^{\mathrm{KL}}(\delta, n_h^t(s, a))}{n_h^t(s, a)}}.$$

*Proof.* By a Hölder and Pinsker inequalities

$$[p_h - \widehat{p}_h^t]f(s,a) \le H\|p_h(s,a) - \widehat{p}_h^t(s,a)\|_1 \le H\sqrt{2\,\mathrm{KL}(\widehat{p}_h^t(s,a)\|p_h(s,a))}.$$

Applying the definition of the event $\mathcal{E}^{\mathrm{KL}}$ we conclude the statement. $\qquad\square$

**Lemma 14.** *Assume conditions of Lemma 12. Then on event $\mathcal{E}^{\mathrm{KL}}(\delta)$, for any $f: \mathcal{S} \to [0,H]$, $t \in \mathbb{N}, h \in [H], (s,a) \in \mathcal{S} \times \mathcal{A}$,*

$$[p_h - \widehat{p}_h^t]f(s,a) \le \frac{1}{H}\widehat{p}_h^t f(s,a) + 2H\left(\frac{2H\beta^{\mathrm{KL}}(\delta, n_h^t(s,a))}{n_h^t(s,a)} \wedge 1\right),$$

$$[\widehat{p}_h^t - p_h]f(s,a) \le \frac{1}{H}p_h f(s,a) + 2H\left(\frac{2H\beta^{\mathrm{KL}}(\delta, n_h^t(s,a))}{n_h^t(s,a)} \wedge 1\right).$$

*Proof.* Let us start from the first statement. We apply the second inequality of Lemma 32 and Lemma 33 to obtain

$$\begin{aligned}
[p_h - \widehat{p}_h^t]f(s,a) &\le \sqrt{2\mathrm{Var}_{p_h}[f](s,a) \cdot \mathrm{KL}(\widehat{p}_h^t\|p_h)} \\
&\le 2\sqrt{\mathrm{Var}_{\widehat{p}_h^t}[f](s,a) \cdot \mathrm{KL}(\widehat{p}_h^t\|p_h)} + 3H\,\mathrm{KL}(\widehat{p}_h^t\|p_h).
\end{aligned}$$

Since $0 \le f(s) \le H$ we get

$$\mathrm{Var}_{\widehat{p}_h^t}[f](s,a) \le \widehat{p}_h^t[f^2](s,a) \le H \cdot \widehat{p}_h^t f(s,a).$$

Finally, applying $2\sqrt{ab} \le a + b, a, b \ge 0$, we obtain the following inequality

$$(\widehat{p}_h^t - p_h)f(s,a) \le \frac{1}{H}\widehat{p}_h^t f(s,a) + 4H^2\,\mathrm{KL}(\widehat{p}_h^t\|p_h).$$

The definition of $\mathcal{E}^{\mathrm{KL}}(\delta)$ implies the part of the statement. At the same time we have a trivial bound since $f(s) \in [0,H]$

$$[p_h - \widehat{p}_h^t]f(s,a) \le 2H \le \frac{1}{H}\widehat{p}_h^t f(s,a) + 2H.$$

To prove the second statement, apply the first inequality of Lemma 32 and proceed similarly. $\qquad\square$

### D.4  CONFIDENCE INTERVALS

Similar to Azar et al. (2017); Zanette & Brunskill (2019a); Ménard et al. (2021), we define the upper confidence bound for the optimal regularized Q-function with Hoeffding bonuses.

Then we have the following sequences defined as follows

$$\overline{Q}_h^t(s,a) = \mathrm{clip}\left(r_h(s,a) + \widehat{p}_h^t \overline{V}_{h+1}^t(s,a) + b_h^{p,t}(s,a), 0, H\right),$$

$$\bar{\pi}_h^{t+1}(s) = \max_{\pi \in \Delta_{\mathcal{A}}}\{\pi\overline{Q}_h^t(s) - \lambda\,\mathrm{KL}(\pi\|\widetilde{\pi}_h(s))\},$$

$$\overline{V}_h^t(s) = \bar{\pi}_h^{t+1}\overline{Q}_h^t(s) - \lambda\,\mathrm{KL}(\bar{\pi}_h^{t+1}(s)\|\widetilde{\pi}_h(s)),$$

$$\overline{V}_{H+1}^t(s) = 0,$$

and the lower confidence bound as follows

$$\underline{Q}_h^t(s,a) = \mathrm{clip}\left(r_h(s,a) + \widehat{p}_h^t \underline{V}_h^t(s,a) - b_h^{p,t}(s,a), 0, H\right)$$

$$\underline{V}_h^t(s) = \max_{\pi \in \Delta_{\mathcal{A}}}\{\pi\underline{Q}_h^t(s) - \lambda\,\mathrm{KL}(\pi\|\widetilde{\pi}_h(s))\},$$

$$\underline{V}_{H+1}^t(s) = 0,$$

where we have two types of transition bonuses that will be specified before use. The Hoeffding bonuses are defined as follows

$$b_h^{p,t}(s,a) \triangleq \sqrt{\frac{2H^2\beta^{\mathrm{KL}}(\delta, n_h^t(s,a))]}{n_h^t(s,a)}}. \tag{10}$$

**Proposition 1.** Let $\delta \in (0, 1)$. Assume Hoeffding bonuses (10). Then on event $\mathcal{G}(\delta)$ for any $t \in \mathbb{N}$, $(h, s, a) \in [H] \times \mathcal{S} \times \mathcal{A}$ it holds

$$\underline{Q}_h^t(s, a) \leq Q_{\widetilde{\pi}, \lambda, h}^\star(s, a) \leq \overline{Q}_h^t(s, a), \qquad \underline{V}_{\lambda, h}^t(s) \leq V_{\widetilde{\pi}, \lambda, h}^\star(s) \leq \overline{V}_h^t(s).$$

*Proof.* Proceed by induction over $h$. For $h = H + 1$ the statement is trivial. Now we assume that inequality holds for any $h' > h$ for a fixed $h \in [H]$. Fix a timestamp $t \in \mathbb{N}$ and a state-action pair $(s, a)$ and assume that $\overline{Q}_h^t(s, a) < H$, i.e. no clipping occurs. Otherwise the inequality $Q_{\widetilde{\pi}, \lambda, h}^\star(s, a) \leq \overline{Q}_h^t(s, a)$ is trivial. In particular, it implies $n_h^t(s, a) > 0$.

In this case by Bellman equations (7) we have

$$[\overline{Q}_h^t - Q_{\widetilde{\pi}, \lambda, h}^\star](s, a) = \widehat{p}_h^t \overline{V}_{h+1}^t(s, a) - p_h V_{\widetilde{\pi}, \lambda, h+1}^\star(s, a) + b_h^{p, t}(s, a).$$

To show that the right-hand side is non-negative, we start from the induction hypothesis

$$[\overline{Q}_h^t - Q_{\widetilde{\pi}, \lambda, h}^\star](s, a) \geq [\widehat{p}_h^t - p_h] V_{\widetilde{\pi}, \lambda, h+1}^\star(s, a) + b_h^{p, t}(s, a).$$

The non-negativity of the expression above automatically holds from Lemma 13. To prove the second inequality on $Q$-value, we proceed exactly the same.

Finally, we have to show the inequality for $V$-values. To do it, we use the fact that $V$-value are computed by $F_{\widetilde{\pi}_h(s), \lambda, h}$ applied to $Q$-value

$$\underline{V}_{\lambda, h}^t(s) = F_{\widetilde{\pi}_h(s), \lambda, h}(\underline{Q}_h^t)(s), \ V_{\widetilde{\pi}, \lambda, h}^\star(s) = F_{\widetilde{\pi}_h(s), \lambda, h}(Q_{\widetilde{\pi}, \lambda, h}^\star)(s), \ \overline{V}_{\lambda, h}^t(s) = F_{\widetilde{\pi}_h(s), \lambda, h}(\overline{Q}_h^t)(s).$$

Notice that $\nabla F_{\widetilde{\pi}_h(s), \lambda, h}$ takes values in a probability simplex, thus, all partial derivatives of $F_{\widetilde{\pi}_h(s), \lambda, h}$ are non-negative and therefore $F_{\widetilde{\pi}_h(s), \lambda, h}$ is monotone in each coordinate. Thus, since $\underline{Q}_h^t(s, a) \leq Q_{\widetilde{\pi}, \lambda, h}^\star(s, a) \leq \overline{Q}_h^t(s, a)$, we have the same inequality $\underline{V}_h^t(s) \leq V_{\widetilde{\pi}, \lambda, h}^\star(s) \leq \overline{V}_h^t(s)$. $\square$

## D.5 SAMPLE COMPLEXITY BOUNDS

In this section, we provide guarantees for the regularization-aware gap that highly depends on the parameter $\lambda$.

Let us recall the regularization-aware gap that is defined recursively, starting from $G_{H+1}^t \triangleq 0$ and

$$W_h^t(s, a) = \left(1 + \frac{1}{H}\right) \widehat{p}_h^t G_{h+1}^t(s) + b_h^{\mathrm{gap}, t}(s, a),$$

$$G_h^t(s) = \mathrm{clip}\left(\bar{\pi}_h^{t+1} W_h^t(s) + \frac{1}{2\lambda} \max_{a \in \mathcal{A}} \left(\overline{Q}_h^t(s, a) - \underline{Q}_h^t(s, a)\right)^2, 0, H\right),$$

(11)

where the additional bonus is defined as

$$b_h^{\mathrm{gap}, t}(s, a) \triangleq \frac{4H^2 \beta^{\mathrm{KL}}(\delta, n_h^t(s, a))}{n_h^t(s, a)},$$

(12)

and the corresponding stopping time for the algorithm

$$\iota = \inf\{t \in \mathbb{N} : G_1^t(s_1) \leq \varepsilon\}.$$

(13)

The next lemma justifies this choice of the stopping rule.

**Lemma 15.** *Assume the choice of Hoeffding bonuses* (10) *and let the event* $\mathcal{G}(\delta)$ *defined in Lemma 12 holds. Then for any* $t \in \mathbb{N}$, $s \in \mathcal{S}$, $h \in [H]$

$$V_{\widetilde{\pi}, \lambda, h}^\star(s) - V_{\widetilde{\pi}, \lambda, h}^{\bar{\pi}^{t+1}}(s) \leq G_h^t(s).$$

*Proof.* Let us proceed by induction. For $h = H + 1$ the statement is trivial. Assume that for any $h' > h$ the statement holds. Also assume that $G_h^t(s) < H$, otherwise the inequality on the policy error holds trivially. In particular, it holds that $n_h^t(s, a) > 0$ for all $a \in \mathcal{A}$.

We can start analysis from understanding the policy error by applying the smoothness of $F_{\widetilde{\pi}_h(s),\lambda,h}$.

$$
\begin{aligned}
V^{\star}_{\widetilde{\pi},\lambda,h}(s) - V^{\bar{\pi}^{t+1}}_{\widetilde{\pi},\lambda,h}(s) &= F_{\widetilde{\pi}_h(s),\lambda,h}(Q^{\star}_{\widetilde{\pi},\lambda,h}(s,\cdot)) - \left(\bar{\pi}^{t+1}_h Q^{\bar{\pi}^{t+1}}_{\widetilde{\pi},\lambda,h}(s) - \lambda\,\mathrm{KL}(\bar{\pi}^{t+1}_h(s)\|\widetilde{\pi}_h(s))\right) \\
&\leq F_{\widetilde{\pi}_h(s),\lambda,h}(\overline{Q}^t_h(s,\cdot)) + \langle \nabla F_{\widetilde{\pi}_h(s),\lambda,h}(\overline{Q}^t_h(s,\cdot)), Q^{\star}_{\widetilde{\pi},\lambda,h}(s,\cdot) - \overline{Q}^t_h(s,\cdot)\rangle \\
&\quad + \frac{1}{2\lambda}\|\overline{Q}^t_h - Q^{\star}_{\widetilde{\pi},\lambda,h}\|^2_\infty(s) - \left(\bar{\pi}^{t+1}_h Q^{\bar{\pi}^{t+1}}_{\widetilde{\pi},\lambda,h}(s) - \lambda\,\mathrm{KL}(\bar{\pi}^{t+1}_h(s)\|\widetilde{\pi}_h(s))\right).
\end{aligned}
$$

Next we recall that

$$
\bar{\pi}^{t+1}_h(s) = \nabla F_{\widetilde{\pi}_h(s),\lambda,h}(\overline{Q}^t_h(s,\cdot)), \quad F_{\widetilde{\pi}_h(s),\lambda,h}(\overline{Q}^t_h)(s) = \bar{\pi}^{t+1}_h \overline{Q}^t_h(s) - \lambda\,\mathrm{KL}(\bar{\pi}^{t+1}_h(s)\|\widetilde{\pi}_h(s)),
$$

thus we have

$$
F_{\widetilde{\pi}_h(s),\lambda,h}(\overline{Q}^t_h)(s) - \left(\bar{\pi}^{t+1}_h Q^{\bar{\pi}^{t+1}}_{\widetilde{\pi},\lambda,h}(s,\cdot) - \lambda\,\mathrm{KL}(\bar{\pi}^{t+1}_h(s)\|\widetilde{\pi}_h(s))\right) = \bar{\pi}^{t+1}_h[\overline{Q}^t_h - Q^{\bar{\pi}^{t+1}}_{\widetilde{\pi},\lambda,h}](s)
$$

and, by Bellman equations

$$
\begin{aligned}
V^{\star}_{\widetilde{\pi},\lambda,h}(s) - V^{\bar{\pi}^{t+1}}_{\widetilde{\pi},\lambda,h}(s) &\leq \bar{\pi}^{t+1}_h\left[Q^{\star}_{\widetilde{\pi},\lambda,h} - Q^{\bar{\pi}^{t+1}}_{\widetilde{\pi},\lambda,h}\right](s) + \frac{1}{2\lambda}\|\overline{Q}^t_h - Q^{\star}_{\widetilde{\pi},\lambda,h}\|^2_*(s) \\
&\leq \bar{\pi}^{t+1}_h p_h\left[V^{\star}_{\widetilde{\pi},\lambda,h+1} - V^{\bar{\pi}^{t+1}}_{\widetilde{\pi},\lambda,h+1}\right](s) + \frac{1}{2\lambda}\|\overline{Q}^t_h - Q^{\star}_{\widetilde{\pi},\lambda,h}\|^2_*(s).
\end{aligned}
$$

By induction hypothesis we have

$$
V^{\star}_{\widetilde{\pi},\lambda,h}(s) - V^{\bar{\pi}^{t+1}}_{\lambda,h}(s) \leq \bar{\pi}^{t+1}_h p_h G^t_{\lambda,h+1}(s) + \frac{1}{2\lambda}\|\overline{Q}^t_h - Q^{\star}_{\widetilde{\pi},\lambda,h}\|^2_*(s).
$$

Next, we apply Lemma 14

$$
\begin{aligned}
p_h G_{\lambda,h+1}(s,a) &= \widehat{p}^t_h G_{\lambda,h+1}(s,a) + [p_h - \widehat{p}^t_h]G^t_{\lambda,h+1}(s,a) \\
&\leq \left(1 + \frac{1}{H}\right)\widehat{p}^t_h G^t_{\lambda,h+1}(s,a) + \frac{4H^2\beta^{\mathrm{KL}}(\delta, n^t_h(s,a))}{n^t_h(s,a)} \triangleq W^t_h(s,a),
\end{aligned}
$$

thus

$$
V^{\star}_{\widetilde{\pi},\lambda,h}(s) - V^{\bar{\pi}^{t+1}}_{\lambda,h}(s) \leq \bar{\pi}^{t+1}_h W^t_h(s) + \frac{1}{2\lambda}\|\overline{Q}^t_h - Q^{\star}_{\widetilde{\pi},\lambda,h}\|^2_\infty(s).
$$

Finally, by the definition of $\|\cdot\|_\infty$ and Proposition 1

$$
V^{\star}_{\widetilde{\pi},\lambda,h}(s) - V^{\bar{\pi}^{t+1}}_{\lambda,h}(s) \leq \bar{\pi}^{t+1}_h W^t_h(s) + \frac{1}{2\lambda}\max_{a\in\mathcal{A}}\left(\overline{Q}^t_h(s,a) - \underline{Q}^t_h(s,a)\right)^2 \triangleq G^t_h(s).
$$

$\square$

**Theorem 5.** *Let $\varepsilon > 0$, $\delta \in (0,1)$, $S \geq 2$ and $\lambda \leq H$. Then* UCBVI-Ent+ *algorithm with Hoeffding bonuses and a stopping rule $\iota$ (13) is $(\varepsilon,\delta)$-PAC for the best policy identification in regularized MDPs.*

*Moreover, the stopping time $\iota$ is bounded as follows*

$$
\iota = \mathcal{O}\left(\frac{H^5 SA \cdot (\log(SAH/\delta) + SL) \cdot L}{\varepsilon\lambda}\right),
$$

*where $L = \mathcal{O}(\log(SAH\log(1/\delta)/(\varepsilon\lambda)))$.*

*Proof.* To show that UCBVI-Ent+ is $(\varepsilon,\delta)$-PAC we notice that on event $\mathcal{G}(\delta)$ for $\widehat{\pi} = \pi^\iota$ by Lemma 15

$$
V^{\star}_{\widetilde{\pi},\lambda,1}(s_1) - V^{\widehat{\pi}}_{\widetilde{\pi},\lambda,1}(s_1) \leq G^\iota_1(s_1) \leq \varepsilon,
$$

and the event $\mathcal{G}(\delta)$ holds with probability at least $1 - \delta$. Next we show that the sample complexity is bounded by the quantity mentioned above.

**Step 1. Bound for $G_1^t(s_1)$** First, we start from bounding $W_h^t(s,a)$ and $G_h^t(s)$. By Lemma 14 we can define the following upper bound for $W_h^t(s,a)$

$$W_h^t(s,a) \leq \left(1 + \frac{2}{H}\right) p_h G_{h+1}^t(s,a) + \frac{8H^2 \beta^{\mathrm{KL}}(\delta, n_h^t(s,a))}{n_h^t(s,a)} \, .$$

Therefore we obtain

$$G_h^t(s) \leq \mathbb{E}_{\bar{\pi}^{t+1}}\Bigg[\left(1 + \frac{2}{H}\right) G_{h+1}^t(s_{h+1}) + \frac{8H^2 \beta^{\mathrm{KL}}(\delta, n_h^t(s_h,a_h))}{n_h^t(s_h,a_h)}$$
$$+ \frac{1}{2\lambda} \max_{a\in\mathcal{A}}\left(\overline{Q}_h^t(s_h,a) - \underline{Q}_h^t(s_h,a)\right)^2 \Big| s_h = s\Bigg],$$

By rolling out this expression

$$G_1^t(s_1) \leq \mathbb{E}_{\bar{\pi}^{t+1}}\Bigg[\sum_{h=1}^{H}\left(1 + \frac{2}{H}\right)^h \frac{8H^2 \beta^{\mathrm{KL}}(\delta, n_h^t(s_h,a_h))}{n_h^t(s_h,a_h)}$$
$$+ \left(1 + \frac{2}{H}\right)^h \frac{1}{2\lambda}\max_{a\in\mathcal{A}}\left(\overline{Q}_h^t(s_h,a) - \underline{Q}_h^t(s_h,a)\right)^2\Bigg].$$

Using the fact that $(1 + 2/H)^h \leq \mathrm{e}^2$, we have

$$G_1^t(s_1) \leq \underbrace{8\mathrm{e}^2 H^2 \mathbb{E}_{\bar{\pi}^{t+1}}\left[\sum_{h=1}^{H} \frac{\beta^{\mathrm{KL}}(\delta, n_h^t(s_h,a_h))}{n_h^t(s_h,a_h)}\right]}_{(\mathbf{A})}$$
$$+ \underbrace{\frac{\mathrm{e}^2}{2\lambda}\mathbb{E}_{\bar{\pi}^{t+1}}\left[\sum_{h=1}^{H}\max_{a\in\mathcal{A}}\left(\overline{Q}_h^t(s_h,a) - \underline{Q}_h^t(s_h,a)\right)^2\right]}_{(\mathbf{B})} \, .$$

**Term $(\mathbf{A})$.** The analysis of the term $(\mathbf{A})$ follows Ménard et al. (2021): we switch counts to pseudo-counts by Lemma 28 and obtain

$$(\mathbf{A}) \leq 32\mathrm{e}^2 H^2 \sum_{h=1}^{H} \sum_{(s,a)\in\mathcal{S}\times\mathcal{A}} d_h^{\bar{\pi}^{t+1}}(s,a) \frac{\beta^{\mathrm{KL}}(\delta, \overline{n}_h^t(s,a))}{\overline{n}_h^t(s,a) \vee 1} \, .$$

**Term $(\mathbf{B})$.** For this term we analyze each summand over $h$ separately. By Lemma 35

$$\mathbb{E}_{\bar{\pi}^{t+1}}\left[\max_{a\in\mathcal{A}}\left(\overline{Q}_h^t(s_h,a) - \underline{Q}_h^t(s_h,a)\right)^2\right] = \mathbb{E}_{\pi^{t+1,(h)}}\left[\left(\overline{Q}_h^t(s_h,a_h) - \underline{Q}_h^t(s_h,a_h)\right)^2\right].$$

Next, we analyze the expression under the square. First, we have

$$\overline{Q}_h^t(s_h,a_h) - \underline{Q}_h^t(s_h,a_h) \leq 2b_h^{p,t}(s_h,a_h) + \widehat{p}_h^t[\overline{V}_{h+1}^t - \underline{V}_{h+1}^t](s_h,a_h).$$

By Lemma 13

$$\overline{Q}_h^t(s_h,a_h) - \underline{Q}_h^t(s_h,a_h) \leq 4b_h^{p,t}(s_h,a_h) + p_h[\overline{V}_{h+1}^t - \underline{V}_{h+1}^t](s_h,a_h).$$

using $\overline{V}_{\lambda,h+1}^t(s) - \underline{V}_{\lambda,h+1}^t(s) \leq \bar{\pi}_{h+1}^t[\overline{Q}_{h+1}^t - \underline{Q}_{h+1}^t](s)$ and the definition of Hoeffding bonuses (10), thus, rolling out this recursion

$$\overline{Q}_h^t(s_h,a_h) - \underline{Q}_h^t(s_h,a_h) \leq 4H \cdot \mathbb{E}_{\bar{\pi}^{t+1}}\left[\sum_{h'=h}^{H} \sqrt{\frac{2\beta^{\mathrm{KL}}(\delta, n_{h'}^t(s_{h'},a_{h'}))}{n_{h'}^t(s_{h'},a_{h'})}}\Big| s_h\right].$$

By Lemma 28, Jensen inequality, and a change of policy $\bar{\pi}^{t+1}$ to $\pi^{t+1,(h)}$ by Lemma 35 we have

$$\overline{Q}_h^t(s_h,a_h) - \underline{Q}_h^t(s_h,a_h) \leq 4H^{3/2}\sqrt{\mathbb{E}_{\pi^{t+1,(h)}}\left[\sum_{h'=h}^{H} \frac{2\beta^{\mathrm{KL}}(\delta, \overline{n}_{h'}^t(s_{h'},a_{h'}))}{\overline{n}_{h'}^t(s_{h'},a_{h'}) \vee 1}\Big| s_h\right]}.$$

By taking the square we get

$$\mathbb{E}_{\bar{\pi}^{t+1}}\left[\max_{a\in\mathcal{A}}\left(\overline{Q}_h^t(s_h,a)-\underline{Q}_h^t(s_h,a)\right)^2\right]$$

$$\leq 16H^3\mathbb{E}_{\pi^{t+1,(h)}}\left[\mathbb{E}_{\pi^{t+1,(h)}}\left[\sum_{h'=h}^H \frac{2\beta^{\mathrm{KL}}(\delta,\overline{n}_{h'}^t(s_{h'},a_{h'}))}{\overline{n}_{h'}^t(s_{h'},a_{h'})\vee 1}\,\Big|\,s_h\right]\right].$$

The telescoping property of conditional expectation yields the final bound

$$\mathbb{E}_{\bar{\pi}^{t+1}}\left[\max_{a\in\mathcal{A}}\left(\overline{Q}_h^t(s_h,a)-\underline{Q}_h^t(s_h,a)\right)^2\right] \leq \sum_{h'=1}^H 32H^3\mathbb{E}_{\pi^{t+1,(h)}}\left[\frac{2\beta^{\mathrm{KL}}(\delta,\overline{n}_{h'}^t(s_{h'},a_{h'}))}{\overline{n}_{h'}^t(s_{h'},a_{h'})\vee 1}\right].$$

Finally, collecting bounds over all $h\in[H]$ we have

$$(\mathbf{B}) \leq \frac{16\mathrm{e}^2 H^3}{\lambda}\sum_{h=1}^H\sum_{s,a}\sum_{h'=1}^H d_{h'}^{\pi^{t,(h)}}(s,a)\frac{\beta^{\mathrm{KL}}(\delta,\overline{n}_{h'}^t(s,a))}{\overline{n}_{h'}^t(s,a)\vee 1}.$$

The final bound for an initial gap follows

$$G_1^t(s_1) \leq 32\mathrm{e}^2 H^2\sum_{h'=1}^H\sum_{(s,a)\in\mathcal{S}\times\mathcal{A}} d_{h'}^{\bar{\pi}^{t+1}}(s,a)\frac{\beta^{\mathrm{KL}}(\delta,\overline{n}_{h'}^t(s,a))}{\overline{n}_{h'}^t(s,a)\vee 1}$$

$$+ \frac{16\mathrm{e}^2 H^3}{\lambda}\sum_{h=1}^H\sum_{s,a}\sum_{h'=1}^H d_{h'}^{\pi^{t,(h)}}(s,a)\frac{\beta^{\mathrm{KL}}(\delta,\overline{n}_{h'}^t(s,a))}{\overline{n}_{h'}^t(s,a)\vee 1}.$$

Since $\lambda\leq H$, we have that $H^2\leq H^3/\lambda$. Using a convention $d_{h'}^{\bar{\pi}^{t+1}}(s,a)=d_{h'}^{\pi^{t+1,(0)}}(s,a)$ we have

$$G_1^t(s_1) \leq \frac{48\mathrm{e}^2 H^3}{\lambda}\sum_{h=0}^H\sum_{s,a}\sum_{h'=1}^H d_{h'}^{\pi^{t,(h)}}(s,a)\frac{\beta^{\mathrm{KL}}(\delta,\overline{n}_{h'}^t(s,a))}{\overline{n}_{h'}^t(s,a)\vee 1}.$$

By changing the summation order and noticing that

$$d_{h'}^{\pi^{\mathrm{mix},t}}(s,a) = \frac{1}{H+1}\sum_{h=0}^H d_h^{\pi^{t,(h)}}(s,a)$$

for $H+1\leq 2H$ we get

$$G_1^t(s_1) \leq \frac{96\mathrm{e}^2 H^4}{\lambda}\sum_{s,a}\sum_{h=1}^H d_h^{\pi^{\mathrm{mix},t}}(s,a)\frac{\beta^{\mathrm{KL}}(\delta,\overline{n}_h^t(s,a))}{\overline{n}_h^t(s,a)\vee 1}.$$

**Step 2. Sum over $t<\iota$.** Assume $\iota>0$. In the case $\iota=0$ the bound is trivially true. Notice that for any $t<\iota$ we have

$$G_{\lambda,1}^t(s_1) > \varepsilon,$$

thus, summing upper bounds on $G_{\lambda,1}^t(s_1)$ over all $t<\iota$ we have

$$\varepsilon(\iota-1) < \sum_{t=1}^{\iota-1} G_{\lambda,1}^t(s_1) \leq \frac{96\mathrm{e}^2 H^4}{\lambda}\sum_{(s,a,h)}\sum_{t=1}^{\iota-1} d_h^{\pi^{\mathrm{mix},t}}(s,a)\frac{\beta^{\mathrm{KL}}(\delta,\overline{n}_h^t(s,a))}{\overline{n}_h^t(s,a)\vee 1}.$$

Notice that $\beta^{\mathrm{KL}}(\delta,\cdot)$ is monotone and maximizes at $\iota-1$, and $d_h^{\pi^{\mathrm{mix},t+1}}(s,a)=\overline{n}_h^{t+1}(s,a)-\overline{n}_h^t(s,a)$. Thus, applying Lemma 29, we have

$$\varepsilon(\iota-1) < \frac{384\mathrm{e}^2 H^5 SA}{\lambda}\beta^{\mathrm{KL}}(\delta,\iota-1)\log(\iota).$$

Then by definition of $\beta^{\mathrm{KL}}$

$$\varepsilon(\iota-1) \leq \frac{384\mathrm{e}^4 H^5 SA}{\lambda}\cdot(\log(2SAH/\delta)+S\log(\mathrm{e}\iota))\cdot\log(\iota).$$

**Step 3. Solving the recurrence.** Define $A = 384\mathrm{e}^2 H^5 SA/(\lambda\varepsilon)$ and $B = \log(2SAH/\delta) + S$. Our goal is to upper bound solutions to the following inequality

$$\iota \leq 1 + A(S\log(\iota) + B) \cdot \log(\iota).$$

First, we obtain a loose solution by using inequality $\log(\iota) \leq \iota^\beta/\beta$ that holds for any $\iota \geq 1$. Taking $\beta = 1/3$ we have

$$\iota \leq 1 + 3A(3S \cdot \iota^{1/3} + B) \cdot \iota^{1/3}.$$

Also we may assume that $\iota \geq 2$, thus $1 \leq \iota/2$ and we achieve

$$\iota^{2/3} \leq 6A(3S\iota^{1/3} + B).$$

Solving this quadratic inequality in $\iota^{1/3}$, we have

$$\iota \leq \left( \frac{18AS + \sqrt{(18AS)^2 + 24AB}}{2} \right)^3 \leq \left( 18AS + \sqrt{24AB} \right)^3.$$

Define $L = 3\log\left(54AS + \sqrt{18AB}\right)$. Then we can easily upper bound the initial inequality as follows

$$\iota \leq 1 + A(B + SL)L.$$

$\square$

# E    BEST POLICY IDENTIFICATION IN REGULARIZED LINEAR MDPS

In this appendix, we first state some useful properties of regularized linear MDPs and describe the `LSVI-UCB-Ent` algorithm.

## E.1    GENERAL PROPERTIES OF LINEAR MDPS

Let us start with a description of how to generalize the techniques of regularized MDPs to the setup of linear function approximation (Jin et al., 2020). Again, we consider the case of KL-regularized MDPs with respect to a reference policy $\widetilde{\pi}$. In this setting the $Q$- and $V$-values could be defined through regularized Bellman equations

$$Q^{\pi}_{\widetilde{\pi},\lambda,h}(s,a) = r_h(s,a) + p_h V^{\pi}_{\widetilde{\pi},\lambda,h+1}(s,a),$$
$$V^{\pi}_{\widetilde{\pi},\lambda,h}(s) = \pi_h Q^{\pi}_{\widetilde{\pi},\lambda,h}(s) - \lambda \operatorname{KL}(\pi_h(s)\|\widetilde{\pi}_h(s)).$$

Moreover, for optimal $Q$- and $V$-functions we have

$$Q^{\star}_{\widetilde{\pi},\lambda,h}(s,a) = r_h(s,a) + p_h V^{\star}_{\widetilde{\pi},\lambda,h+1}(s,a),$$
$$V^{\star}_{\widetilde{\pi},\lambda,h}(s) = \max_{\pi \in \Delta_{\mathcal{A}}}\left\{\pi Q^{\star}_{\widetilde{\pi},\lambda,h}(s) - \lambda \operatorname{KL}(\pi\|\widetilde{\pi}_h(s))\right\}.$$

Note that the value of a policy could be arbitrarily negative, however, we know a priori that the optimal policy has non-negative value $V^{\star}_{\widetilde{\pi},\lambda,h} \in [0,H]$, since the policy $\widetilde{\pi}$ itself has non-negative value.

In particular, under this assumption, we have the following simple proposition

**Proposition 2.** For a linear MDP, for any policy $\pi$ such that $V^{\pi}_{\widetilde{\pi},\lambda,h}(s,a) \geq 0$ for any $(s,a,h) \in \mathcal{S} \times \mathcal{A} \times [H]$ there exists weights $\{w^{\pi}_h\}_{h\in[H]}$ such that for any $(s,a,h) \in \mathcal{S} \times \mathcal{A} \times [H]$ we have $Q^{\pi}_{\widetilde{\pi},\lambda,h}(s,a) = \psi(s,a)^{\mathsf{T}}w^{\pi}_h$. Moreover, for any $h \in [H]$ it holds $\|w^{\pi}_h\|_2 \leq 2H\sqrt{d}$.

*Proof.* By Bellman equations

$$Q^{\pi}_{\widetilde{\pi},\lambda,h}(s,a) = r_h(s,a) + p_h V^{\pi}_{\widetilde{\pi},\lambda,h}(s,a) = \psi(s,a)^{\mathsf{T}}\theta_h + \int_{\mathcal{S}} V^{\pi}_{\widetilde{\pi},\lambda,h}(s') \cdot \sum_{i=1}^{d} \psi(s,a)_i \mu_{h,i}(\mathrm{d}s')$$

$$= \langle \psi(s,a), \theta_h + \int_{\mathcal{S}} V^{\pi}_{\widetilde{\pi},\lambda,h}(s')\mu_h(\mathrm{d}s')\rangle.$$

To show the second part, we use Definition 2. First, we note that $\|\theta_h\| \leq \sqrt{d}$ and, at the same time

$$\int_{\mathcal{S}} V^{\pi}_{\widetilde{\pi},\lambda,h}(s')\mu_h(\mathrm{d}s') \leq H\sqrt{d}.$$

since the value is bounded by $H$.

$\square$

## E.2    ALGORITHM DESCRIPTION

In this appendix, we describe the `LSVI-UCB-Ent` algorithm for regularized BPI in linear MDPs. `LSVI-UCB-Ent` is characterized by the following rules.

**Sampling rule**  As for the `UCBVI-Ent+` algorithm we start with regularized optimistic planning under the linear function approximation

$$\overline{Q}^t_h(s,a) = \psi_h(s,a)^{\mathsf{T}}\overline{w}^t_h + b^t_h(s,a),$$
$$\overline{V}^t_h(s) = \operatorname{clip}\left(\max_{\pi\in\Delta_{\mathcal{A}}}\left\{\pi\overline{Q}^t_h(s) - \lambda\operatorname{KL}(\pi,\widetilde{\pi}_h(s))\right\}, 0, H\right), \tag{14}$$
$$\overline{\pi}^{t+1}_h(s) = \arg\max_{\pi\in\Delta_{\mathcal{A}}}\left\{\pi\overline{Q}^t_h(s) - \lambda\operatorname{KL}(\pi,\widetilde{\pi}_h(s))\right\},$$

where $b^t$ is some bonus defined as follows

$$b_h^t = \mathcal{B} \cdot \sqrt{[\psi(s,a)]^\intercal [\Lambda_h^t]^{-1} \psi(s,a)}$$

for $\mathcal{B} > 0$ a bonus scaling factor, and the parameter $w_h^t$ is obtained by least-square value iteration with Tikhonov regularization parameter $\alpha$ (Jin et al., 2020),

$$\overline{w}_h^t = \arg\min_{w \in \mathbb{R}^d} \sum_{k=1}^t \left[ r_h(s_h^k, a_h^k) + \overline{V}_{h+1}^t(s_{h+1}^k) - \psi(s_h^k, a_h^k)^\intercal w \right]^2 + \alpha \|w\|_2^2 \,.$$

We notice that there is a closed-form solution to this problem given by

$$\overline{w}_h^t = \left[ \Lambda_h^t \right]^{-1} \left[ \sum_{\tau=1}^t \psi_h^\tau \left[ r_h^\tau + \overline{V}_{h+1}^t(s_{h+1}^\tau) \right] \right],$$

where $\psi_h^\tau = \psi(s_h^\tau, a_h^\tau)$ and $\Lambda_h^t = \sum_{\tau=1}^t \psi_h^\tau [\psi_h^\tau]^\intercal + \alpha I$.

Then we also define a family of exploratory policies by, for all $h' \in [H]$,

$$\pi^{t,(h')}(a|s) = \begin{cases} \pi^{t,(h')}(a|s) = \bar{\pi}_h^t(a|s) & \text{if } h \neq h' \\ \pi^{t,(h')}(a|s) = \mathbb{1}\left\{ a \in \arg\max_{a' \in \mathcal{A}} (\overline{Q}_h^t(s,a') - \underline{Q}_h^t(s,a')) \right\} & \text{if } h = h' \end{cases}, \quad (15)$$

where $\underline{Q}^t$ is some lower bound on the optimal regularized Q-value defined as follows

$$
\begin{aligned}
\underline{w}_h^t &= \left[ \Lambda_h^t \right]^{-1} \left[ \sum_{\tau=1}^t \psi_h^\tau \left[ r_h^\tau + \underline{V}_{h+1}^t(s_{h+1}^\tau) \right] \right], \\
\underline{Q}_h^t(s,a) &= [\psi(s,a)]^\intercal \underline{w}_h^t - \mathcal{B} \cdot \sqrt{[\psi(s,a)]^\intercal [\Lambda_h^t]^{-1} \psi(s,a)}, \\
\underline{V}_h^t(s) &= \text{clip}\left( \max_{\pi \in \Delta_\mathcal{A}} \left\{ \pi \underline{Q}_h^t(s) - \lambda \, \text{KL}(\pi \| \widetilde{\pi}_h(s)) \right\}, 0, H \right),
\end{aligned}
\quad (16)
$$

The sampling rule is then obtained by picking uniformly at random a policy among the exploratory policies, $\pi^t = \pi^{t,(h')}$ for $h' \sim \mathcal{U}\text{nif}[H]$. Notice that it is equivalent to using a non-Markovian mixture policy $\pi^{\text{mix},t}$ over all $h \in [H]$. Additionally, it would be valuable to mention that computation of this policy could be done on-flight since we can compute $\overline{Q}$ and $\underline{Q}$.

**Stopping and decision rule** In the linear setting we use a simple deterministic stopping rule $\tau = T$ for a fixed parameter $T$. In the finite setting, we were able to define an adaptive stopping rule by leveraging a certain Bernstein-like inequality on the gaps (see Lemma 14). However, it is not clear how to adapt such inequality to the linear setting.

As decision rule `LSVI-UCB-Ent` returns the non-Markovian policy $\widehat{\pi}$, the uniform mixture over the optimistic policies $\{\bar{\pi}^t\}_{t \in [T]}$ The complete procedure is described in Algorithm 4.

---

**Algorithm 4** `LSVI-UCB-Ent`

---

1: **Input:** Number of episodes $T$, bonus function $b^t$, Tikhonov regularization parameter $\alpha$.
2: **for** $t \in [T]$ **do**
3:     Compute $\bar{\pi}^t$ by regularized optimistic planning with (14).
4:     Sample $h' \sim \mathcal{U}\text{nif}\{1,\ldots,H\}$ and set $\pi^t = \pi^{t,(h')}$
5:     **for** $h \in [H]$ **do**
6:         Play $a_h^t \sim \pi_h^t(s_h^t)$
7:         Observe $s_{h+1}^t \sim p_h(s_h^t, a_h^t)$
8:     **end for**
9: **end for**
10: **Output** $\widehat{\pi}$ the uniform mixture over $\{\pi^t\}_{t \in [T]}$.

---

### E.3 Concentration Events

In this section, the required concentration events for a proof of sample complexity for `LSVI-UCB-Ent` will be described. First, we define several important objects. Let $\psi_h^\tau = \psi(s_h^\tau, a_h^\tau)$ for any $\tau \in \mathbb{N}, h \in [H]$ and define

$$\Lambda_h^t = \alpha I_d + \sum_{\tau=1}^{t} \psi_h^\tau [\psi_h^\tau]^\intercal, \quad \overline{\Lambda}_h^t = \alpha I_d + \sum_{\tau=1}^{t} \mathbb{E}_{\pi^{\mathrm{mix},\tau}} [\psi(s_h, a_h)[\psi(s_h, a_h)]^\intercal | s_1],$$

where $\widetilde{\pi}^t$ is a uniform mixture policy of $\pi^{t,(h')}$ defined in (15) over all $h' \in \{0, \ldots, H\}$.

Let $\beta^{\mathrm{conc}} \colon (0, 1) \times \mathbb{N} \times \mathbb{R}_+ \times \mathbb{R}_+ \to \mathbb{R}_+$ and $\beta^{\mathrm{cnt}} \colon (0, 1) \times \mathbb{N} \to \mathbb{R}_+$ be some functions defined later on in Lemma 16. We define the following favorable events for any fixed values of bonus scaling $\mathcal{B} > 0$ and Ridge coefficient $\alpha \geq 1$ that will be specified later.

$$\mathcal{E}^{\mathrm{conc}}(\delta, \mathcal{B}) \triangleq \Bigg\{ \forall t \in \mathbb{N}, \forall h \in [H] :$$

$$\left\| \sum_{\tau=1}^{t} \psi_h^\tau \left\{ \overline{V}_{h+1}^t(s_{h+1}^\tau) - p_h \overline{V}_{h+1}^t(s_h^\tau, a_h^\tau) \right\} \right\|_{[\Lambda_h^t]^{-1}} \leq 2dH \sqrt{\beta^{\mathrm{conc}}(\delta, t, \mathcal{B})}$$

$$\left\| \sum_{\tau=1}^{t} \psi_h^\tau \left\{ \underline{V}_{h+1}^t(s_{h+1}^\tau) - p_h \underline{V}_{h+1}^t(s_h^\tau, a_h^\tau) \right\} \right\|_{[\Lambda_h^t]^{-1}} \leq 2dH \sqrt{\beta^{\mathrm{conc}}(\delta, t, \mathcal{B})} \Bigg\},$$

$$\mathcal{E}^{\mathrm{cnt}}(\delta) \triangleq \Bigg\{ \forall t \in \mathbb{N}, \forall h \in [H] : \quad \Lambda_h^t \succcurlyeq \frac{1}{2} \overline{\Lambda}_h^t - \beta^{\mathrm{cnt}}(\delta, t) I_d \Bigg\},$$

We also introduce an intersection of these events of interest, $\mathcal{G}(\delta, \mathcal{B}) \triangleq \mathcal{E}^{\mathrm{conc}}(\delta, \mathcal{B}) \cap \mathcal{E}^{\mathrm{cnt}}(\delta)$. We prove that for the right choice of the functions $\beta^{\mathrm{conc}}, \beta^{\mathrm{cnt}}$ the above events hold with high probability.

**Lemma 16.** *Let $\mathcal{B}, \alpha \geq 1$ be fixed. For any $\delta \in (0, 1)$ and for the following choices of functions $\beta$,*

$$\beta^{\mathrm{conc}}(\delta, t, \mathcal{B}) \triangleq 2 \log\left( \frac{H(1 + t^2)}{\delta} \right) + 5 + \log\left( 1 + 8d^{1/2} t^2 \cdot \left( \frac{\mathcal{B}}{Hd} \right)^2 \right),$$

$$\beta^{\mathrm{cnt}}(\delta, t) \triangleq 4 \log(8\mathrm{e}H(2t + 1)/\delta) + 4d \log(3t) + 3,$$

*for any fixed $\alpha \geq 1$ it holds that*

$$\mathbb{P}[\mathcal{E}^{\mathrm{conc}}(\delta, \mathcal{B})] \geq 1 - \delta/2, \quad \mathbb{P}[\mathcal{E}^{\mathrm{cnt}}(\delta)] \geq 1 - \delta/2,$$

*In particular, $\mathbb{P}[\mathcal{G}(\delta, \mathcal{B})] \geq 1 - \delta$.*

*Proof.* Let us fix $h \in [H]$. Then for all $t \in \mathbb{N}$ by Lemma 17 we have $\|w_h^t\|_2 \leq 2H\sqrt{dt/\alpha}$ and by a construction of $\Lambda_h^t$ we have $\lambda_{\min}(\Lambda_h^t) \geq \alpha$. Therefore, combination of Lemmas 25 and 26 for any fixed $\varepsilon > 0$ we have with probability at least $1 - \delta/H$

$$\left\| \sum_{\tau=1}^{t} \psi_h^\tau \left\{ \overline{V}_{h+1}^t(s_{h+1}^\tau) - p_h \overline{V}_{h+1}^t(s_h^\tau, a_h^\tau) \right\} \right\|_{[\Lambda_h^t]^{-1}}^2 \leq 4H^2 \left[ \frac{d}{2} \log\left( \frac{H(t + \alpha)}{\alpha \delta} \right) + d \log\left( 1 + \frac{8Hd^{1/2} t^{1/2}}{\varepsilon \alpha^{1/2}} \right) \right.$$

$$\left. + d^2 \log\left( 1 + \frac{8d^{1/2} \mathcal{B}^2}{\alpha \varepsilon^2} \right) \right] + \frac{8t^2 \varepsilon^2}{\alpha}.$$

Next we take $\varepsilon = Hd/t$ and obtain by using $\alpha \geq 1$ and $d \geq 1$

$$\left\| \sum_{\tau=1}^{t} \psi_h^\tau \left\{ \overline{V}_{h+1}^t(s_{h+1}^\tau) - p_h \overline{V}_{h+1}^t(s_h^\tau, a_h^\tau) \right\} \right\|_{[\Lambda_h^t]^{-1}}^2 \leq 4H^2 d^2 \left[ 2 \log\left( \frac{H(1 + t^2)}{\delta} \right) + 5 \right.$$

$$\left. + \log\left( 1 + 8d^{1/2} t^2 \cdot \left( \frac{\mathcal{B}}{Hd} \right)^2 \right) \right].$$

Taking the square root we conclude the first half of the statement, the second half is exactly the same, since Lemma 25 gives a bound uniformly over all value functions.

By Theorem 27 and union bound over $h \in [H]$, $\mathbb{P}[\mathcal{E}^{\mathrm{cnt}}(\delta)] \geq 1 - \delta/2$. The union bound over two prescribed events concludes $\mathbb{P}[\mathcal{G}(\delta, \mathcal{B})] \geq 1 - \delta$. □

The proof of the following lemma remains exactly the same as in Jin et al. (2020).

**Lemma 17.** *[Lemma B.2 by Jin et al. (2020)] For any $(t, h) \in \mathbb{N} \times [H]$ the weights $w_h^t$ generated by* `LSVI-UCB-Ent` *satisfies*
$$\|w_h^t\|_2 \leq 2H\sqrt{dt/\alpha}.$$

### E.4 CONFIDENCE INTERVALS

In this section, we provide the confidence intervals on the optimal Q-function that is required for the proof of sample complexity of `LSVI-UCB-Ent`.

We start from the specification of the required values of $\alpha$ and $\mathcal{B}$.

$$\alpha \triangleq 2(\beta^{\mathrm{cnt}}(\delta, T) + 1), \qquad \mathcal{B} = 32dH\sqrt{\log\left(\frac{24edHT}{\delta}\right)}, \tag{17}$$

where $\beta^{\mathrm{cnt}}$ is defined in Lemma 16.

**Proposition 3.** *Let $\alpha$ and $\mathcal{B}$ satisfy (17). Then on the event $\mathcal{G}(\delta)$ defined in Lemma 16 we have*
$$\langle \psi(s, a), \overline{w}_h^t \rangle - Q_{\widetilde{\pi}, \lambda, h}^{\star}(s, a) = p_h[\overline{V}_{h+1}^t - V_{\widetilde{\pi}, \lambda, h+1}^{\star}](s, a) + \overline{\Delta}_h^t,$$
$$\langle \psi(s, a), \underline{w}_h^t \rangle - Q_{\widetilde{\pi}, \lambda, h}^{\star}(s, a) = p_h[\underline{V}_{h+1}^t - V_{\widetilde{\pi}, \lambda, h+1}^{\star}](s, a) + \underline{\Delta}_h^t,$$
*where $\overline{\Delta}_h^t(s, a)$ and $\underline{\Delta}_h^t(s, a)$ satisfies*
$$\max\{|\overline{\Delta}_h^t(s, a)|, |\underline{\Delta}_h^t(s, a)|\} \leq \mathcal{B}\sqrt{\langle \psi(s, a), [\Lambda_h^t]^{-1}\psi(s, a)\rangle}.$$

*Proof.* We provide the proof only for the first equation since proof of one statement completely reassembles the other. By Proposition 2 and Bellman equations we have
$$Q_{\widetilde{\pi}, \lambda, h}^{\star}(s, a) = \langle \psi(s, a), w_h^{\star} \rangle = r_h(s, a) + p_h V_{\widetilde{\pi}, \lambda, h+1}^{\star}(s, a),$$

therefore
$$\overline{w}_h^t - w_h^{\star} = [\Lambda_h^t]^{-1}\left[\sum_{\tau=1}^t \psi_h^\tau[r_h^\tau + \overline{V}_{h+1}^t(s_{h+1}^\tau)] - \sum_{\tau=1}^t \psi_h^\tau \langle \psi(s_h^\tau, a_h^\tau), w_h^{\star}\rangle - \alpha w_h^{\star}\right]$$

$$= \underbrace{-\alpha[\Lambda_h^t]^{-1}w_h^{\star}}_{\xi_1} + \underbrace{[\Lambda_h^t]^{-1}\sum_{\tau=1}^t \psi_h^\tau\left[\overline{V}_{h+1}^t(s_{h+1}^\tau) - p_h\overline{V}_{h+1}^t(s_h^\tau, a_h^\tau)\right]}_{\xi_2}$$

$$+ \underbrace{[\Lambda_h^t]^{-1}\sum_{\tau=1}^t \psi_h^\tau p_h\left[\overline{V}_{h+1}^t - V_{\widetilde{\pi}, \lambda, h+1}^{\star}\right](s_h^\tau, a_h^\tau)}_{\xi_3}.$$

Next, we analyze the last term $\xi_3$. By Definition 2 we have
$$p_h\left[\overline{V}_{h+1}^t - V_{\widetilde{\pi}, \lambda, h+1}^{\star}\right](s_h^\tau, a_h^\tau) = \left\langle \psi_h^\tau, \int_{\mathcal{S}}[\overline{V}_{h+1}^t - V_{\widetilde{\pi}, \lambda, h+1}^{\star}](s')\mu_h(\mathrm{d}s')\right\rangle,$$

thus
$$\xi_3 = [\Lambda_h^t]^{-1}\left(\sum_{\tau=1}^t \psi_h^\tau[\psi_h^\tau]^{\mathsf{T}} + \alpha I_d - \alpha I_d\right)\left[\int_{\mathcal{S}}[\overline{V}_{h+1}^t - V_{\widetilde{\pi}, \lambda, h+1}^{\star}](s')\mu_h(\mathrm{d}s')\right]$$

$$= \int_{\mathcal{S}}[\overline{V}_{h+1}^t - V_{\widetilde{\pi}, \lambda, h+1}^{\star}](s')\mu_h(\mathrm{d}s') \underbrace{-\alpha[\Lambda_h^t]^{-1}\int_{\mathcal{S}}[\overline{V}_{h+1}^t - V_{\widetilde{\pi}, \lambda, h+1}^{\star}](s')\mu_h(\mathrm{d}s')}_{\xi_4}.$$

As a result, moving to $Q$-values directly we have

$$\langle \psi(s,a), \overline{w}_h^t \rangle - Q_h^\pi(s,a) = \langle \psi(s,a), \overline{w}_h^t - w_h^\star \rangle$$
$$= p_h[\overline{V}_{h+1}^t - V_{\tilde{\pi},\lambda,h+1}^\star](s,a) + \underbrace{\langle \psi(s,a), \xi_1 + \xi_2 + \xi_4 \rangle}_{\overline{\Delta}_h^t(s,a)}.$$

Next, we compute an upper bound for $\overline{\Delta}_h^t(s,a)$. We start from the first term, where we apply Cauchy-Schwartz inequality and the second statement of Proposition 2

$$|\langle \psi(s,a), \xi_1 \rangle| = |\langle \psi(s,a), \alpha w_h^\star \rangle_{[\Lambda_h^t]^{-1}}| \le \alpha \|\psi(s,a)\|_{[\Lambda_h^t]^{-1}} \cdot \|w_h^\star\|_{[\Lambda_h^t]^{-1}}$$
$$\le 2H\sqrt{d\alpha}\|\psi(s,a)\|_{[\Lambda_h^t]^{-1}},$$

where we have used that $\|[\Lambda_h^t]^{-1}\|_2 \le 1/\alpha$. For the third term, we apply exactly the same construction and obtain the same upper bound. For the second term, we also apply Cauchy-Schwartz inequality and Lemma 16

$$|\langle \psi(s,a), \xi_1 \rangle| \le \|\psi(s,a)\|_{[\Lambda_h^t]^{-1}} \left\| \sum_{\tau=1}^t \psi_h^\tau \left[ \overline{V}_{h+1}^t(s_{h+1}^\tau) - p_h \overline{V}_{h+1}^t(s_h^\tau, a_h^\tau) \right] \right\|_{[\Lambda_h^t]^{-1}}$$
$$\le 2dH\sqrt{\beta^{\mathrm{conc}}(\delta,t,\mathcal{B})}\|\psi(s,a)\|_{[\Lambda_h^t]^{-1}}.$$

Thus, we have

$$|\overline{\Delta}_h^t(s,a)| \le \left[ 2dH\sqrt{\beta^{\mathrm{conc}}(\delta,T)} + 4H\sqrt{2d(\beta^{\mathrm{cnt}}(\delta,T)+1)} \right] \sqrt{[\psi(s,a)]^\top [\Lambda_h^t]^{-1}\psi(s,a)}.$$

The only part is to show that for our particular choice of $\mathcal{B}$ it holds

$$2dH\sqrt{\beta^{\mathrm{conc}}(\delta,T,\mathcal{B})} + 4H\sqrt{2d(\beta^{\mathrm{cnt}}(\delta,T)+1)} \le \mathcal{B}.$$

First, we notice that

$$4H\sqrt{2d(\beta^{\mathrm{cnt}}(\delta,T)+1)} \le 16Hd\sqrt{\log\left(\frac{24\mathrm{e}HT}{\delta}\right)} \le \mathcal{B}/2.$$

Thus, it is enough to show that

$$\beta^{\mathrm{conc}}(\delta,T,\mathcal{B}) = 2\log\left(\frac{H(1+T^2)}{\delta}\right) + 5 + \log\left(1 + 8d^{1/2}T^2\left(\frac{\mathcal{B}}{dH}\right)^2\right) \le \frac{1}{16}\left(\frac{\mathcal{B}}{dH}\right)^2.$$

First, we notice that since $T \ge 1$ and $\delta \in (0,1)$ then

$$2\log\left(\frac{H(1+T^2)}{\delta}\right) + 5 \le 4\log\left(\frac{2TH\mathrm{e}^2}{\delta}\right),$$

and also, using the inequality $\log(1+x) \le x$ for any $x \ge 0$

$$\log\left(1 + 8d^{1/2}T^2\left(\frac{\mathcal{B}}{dH}\right)^2\right) \le \log\left(1 + \frac{1}{32}\left(\frac{\mathcal{B}}{dH}\right)^2\right) + \log\left(32 \cdot 8 \cdot d^{1/2}T^2\right)$$
$$\le \frac{1}{32}\left(\frac{\mathcal{B}}{dH}\right)^2 + 2\log(16 \cdot dT).$$

Thus, it is enough for $\mathcal{B}$ to satisfy the following inequalities

$$\log\left(\frac{24\mathrm{e}HT}{\delta}\right) \le \left(\frac{\mathcal{B}}{32dH}\right)^2, \quad 4\log\left(\frac{2TH\mathrm{e}^2}{\delta}\right) \le \left(\frac{\mathcal{B}}{8dH}\right)^2, \quad 2\log(16dT) \le \left(\frac{\mathcal{B}}{8dH}\right)^2.$$

It is clear that the choice $\mathcal{B}$ defined in (17) satisfies all required inequalities. $\qquad\square$

**Corollary 5** (Confidence intervals validity). *Let constant $\alpha$ and $\mathcal{B}$ defined in (17). Then on the event $\mathcal{G}(\delta,\mathcal{B})$ we have $\overline{Q}_h^t(s,a) \ge Q_{\tilde{\pi},\lambda,h}^\star(s,a) \ge \underline{Q}_h^t(s,a)$ and $\overline{V}_h^t(s) \ge V_{\tilde{\pi},\lambda,h}^\star(s) \ge \underline{V}_h^t(s)$ for any $t \in [T], h \in [H], (s,a) \in \mathcal{S} \times \mathcal{A}$.*

*Proof.* Let us prove using backward induction over $h \in [H]$. For $h = H + 1$ this statement is trivially true. Let us assume that the statement holds for any $h' > h$. Thus by Proposition 3

$$\overline{Q}_h^t(s, a) - Q_{\widetilde{\pi}, \lambda, h}^\star(s, a) = \overline{\Delta}_h^t(s, a) + \mathcal{B}\sqrt{[\psi(s, a)]^\intercal [\Lambda_h^t]^{-1} \psi(s, a)} + p_h \left[ \overline{V}_{h+1}^t - V_{h+1}^\star \right](s, a).$$

Notice that $\overline{\Delta}_h^t(s, a) + \mathcal{B}\sqrt{[\psi(s, a)]^\intercal [\Lambda_h^t]^{-1} \psi(s, a)} \geq 0$ and by induction hypothesis $\overline{V}_{h+1}^t - V_{h+1}^\star \geq 0$ for any $s'$. Thus, we have proven the required statement for $Q$-values. To show it for $V$-values, we notice that if upper clipping in the definition $\overline{V}$ in (14) occurs, then the statement trivially holds. Otherwise, we have

$$\overline{V}_h^t(s) - V_{\widetilde{\pi}, \lambda, h}^\star(s) \geq F_{\widetilde{\pi}_h(s), \lambda, h}(\overline{Q}_h^t(s)) - F_{\widetilde{\pi}_h(s), \lambda, h}(Q_{\widetilde{\pi}, \lambda, h}^\star(s)).$$

However, the function $F_{\widetilde{\pi}_h(s), \lambda, h}$ is monotone since its gradients lies in a probability simplex. Thus, $\overline{V}_h^t(s) - V_{\widetilde{\pi}, \lambda, h}^\star(s) \geq 0$. Using exactly the same reasoning we may show the lower confidence bound. $\qquad\square$

### E.5 SAMPLE COMPLEXITY BOUNDS

In this section, we provide the sample complexity result of the `LSVI-UCB-Ent` algorithm. We start from a general result that does not depend on the properties of linear MDPs, however, depends on the properties of the algorithms.

**Lemma 18.** *Let constants $\alpha, \mathcal{B}$ be defined by (17). Then on event $\mathcal{G}(\delta, \mathcal{B})$ defined in Lemma 16 for any $t \in \mathbb{N}$, $s \in \mathcal{S}, h \in [H]$*

$$V_{\widetilde{\pi}, \lambda, h}^\star(s) - V_{\widetilde{\pi}, \lambda, h}^{\bar{\pi}^{t+1}}(s) \leq \frac{1}{2\lambda} \mathbb{E}_{\bar{\pi}^{t+1}} \left[ \sum_{h=1}^{H} \max_{a \in \mathcal{A}} \left( \overline{Q}_h^t - \underline{Q}_h^t \right)^2 (s_h, a) \right].$$

*Proof.* Let us proceed by induction. For $h = H + 1$ the statement is trivial. Assume that for any $h' > h$ the statement holds. Also assume that $G_h^t(s) < H$, otherwise the inequality on the policy error holds trivially. In particular, it holds that $n_h^t(s, a) > 0$ for all $a \in \mathcal{A}$.

We can start analysis from understanding the policy error by applying the smoothness of $F_{\widetilde{\pi}_h(s), \lambda, h}$.

$$V_{\widetilde{\pi}, \lambda, h}^\star(s) - V_{\widetilde{\pi}, \lambda, h}^{\bar{\pi}^{t+1}}(s) = F_{\widetilde{\pi}_h(s), \lambda, h}(Q_{\widetilde{\pi}, \lambda, h}^\star(s, \cdot)) - \left( \bar{\pi}_h^{t+1} Q_{\widetilde{\pi}, \lambda, h}^{\bar{\pi}^{t+1}}(s, \cdot) - \lambda \operatorname{KL}(\bar{\pi}_h^{t+1}(s) \| \widetilde{\pi}_h(s)) \right)$$

$$\leq F_{\widetilde{\pi}_h(s), \lambda, h}(\overline{Q}_h^t(s, \cdot)) + \langle \nabla F_{\widetilde{\pi}_h(s), \lambda, h}(\overline{Q}_h^t(s, \cdot)), Q_{\widetilde{\pi}, \lambda, h}^\star(s, \cdot) - \overline{Q}_h^t(s, \cdot) \rangle$$

$$+ \frac{1}{2\lambda} \| \overline{Q}_h^t - Q_{\widetilde{\pi}, \lambda, h}^\star \|_\infty^2(s) - \left( \bar{\pi}_h^{t+1} Q_{\widetilde{\pi}, \lambda, h}^{\bar{\pi}^{t+1}}(s, \cdot) - \lambda \operatorname{KL}(\bar{\pi}_h^{t+1}(s) \| \widetilde{\pi}_h(s)) \right).$$

Next we recall that

$$\bar{\pi}_h^{t+1}(s) = \nabla F(\overline{Q}_h^t(s, \cdot)), \quad F(\overline{Q}_h^t)(s) = \bar{\pi}_h^{t+1} \overline{Q}_h^t(s) - \lambda \operatorname{KL}(\bar{\pi}_h^{t+1}(s) \| \widetilde{\pi}_h(s)),$$

thus we have

$$F(\overline{Q}_h^t)(s) - \left( \bar{\pi}_h^{t+1} Q_{\widetilde{\pi}, \lambda, h}^{\bar{\pi}^{t+1}}(s, \cdot) - \lambda \operatorname{KL}(\bar{\pi}_h^{t+1}(s) \| \widetilde{\pi}_h(s)) \right) = \bar{\pi}_h^{t+1} [\overline{Q}_h^t - Q_{\widetilde{\pi}, \lambda, h}^{\bar{\pi}^{t+1}}](s)$$

and, by Bellman equations

$$V_{\widetilde{\pi}, \lambda, h}^\star(s) - V_{\widetilde{\pi}, \lambda, h}^{\bar{\pi}^{t+1}}(s) \leq \bar{\pi}_h^{t+1} \left[ Q_{\widetilde{\pi}, \lambda, h}^\star - Q_{\widetilde{\pi}, \lambda, h}^{\bar{\pi}^{t+1}} \right](s) + \frac{1}{2\lambda} \| \overline{Q}_h^t - Q_{\widetilde{\pi}, \lambda, h}^\star \|_\infty^2(s)$$

$$\leq \bar{\pi}_h^{t+1} p_h \left[ V_{\widetilde{\pi}, \lambda, h+1}^\star - V_{\widetilde{\pi}, \lambda, h+1}^{\bar{\pi}^{t+1}} \right](s) + \frac{1}{2\lambda} \| \overline{Q}_h^t - Q_{\widetilde{\pi}, \lambda, h}^\star \|_\infty^2(s).$$

Next, we start by changing the norm and using the properties of $\overline{Q}$ and $\underline{Q}$ (see Corollary 5)

$$\| \overline{Q}_h^t - Q_{\widetilde{\pi}, \lambda, h}^\star \|_\infty^2(s) = \max_{a \in \mathcal{A}} \left( \overline{Q}_h^t(s, a) - Q_{\widetilde{\pi}, \lambda, h}^\star(s, a) \right)^2 \leq \max_{a \in \mathcal{A}} \left( \overline{Q}_h^t(s, a) - \underline{Q}_h^t(s, a) \right)^2.$$

$$\qquad\square$$

**Theorem 6.** *Let $\varepsilon > 0, \delta \in (0,1)$. Then* `LSVI-UCB-Ent` *algorithm with a choice of parameters described in* (17) *is $(\varepsilon, \delta)$-PAC for the best policy identification in regularized MDPs after*

$$T = \widetilde{\mathcal{O}}\left(\frac{H^5 d^2}{\lambda \varepsilon}\right)$$

*iterates.*

*Proof.* First, we notice that the definition of the output policy $\widehat{\pi}$ as a mixture policy over all $\bar{\pi}^t$ allows us to define the following

$$V^\star_{\widetilde{\pi},\lambda,1}(s_1) - V^{\widehat{\pi}}_{\widetilde{\pi},\lambda,1}(s_1) = \frac{1}{T}\sum_{i=1}^{T}\{V^\star_{\widetilde{\pi},\lambda,1}(s_1) - V^{\bar{\pi}^t}_{\widetilde{\pi},\lambda,1}(s_1)\},$$

therefore it is enough to compute only the average regret of the presented procedure. In the sequel we assume the event $\mathcal{G}(\delta, \mathcal{B})$ for $\mathcal{B}$ defined in (17).

**Step 1. Study of sub-optimality gap**  Let us fix $t \in [T]$, then by Lemma 18 we have

$$V^\star_{\widetilde{\pi},\lambda,1}(s_1) - V^{\bar{\pi}^{t+1}}_1(s_1) \le \frac{1}{2\lambda}\sum_{h=1}^{H}\mathbb{E}_{\bar{\pi}^{t+1}}\left[\max_{a\in\mathcal{A}}\left(\overline{Q}^t_h - \underline{Q}^t_h\right)^2(s_h,a)\right].$$

Next, we analyze each term separately, starting from the difference between Q-values inside. Let us fix $h \in [H]$, then by Proposition 3

$$\begin{aligned}\overline{Q}^t_h(s,a) - \underline{Q}^t_h(s,a) &= [\overline{Q}^t_h - Q^\star_{\widetilde{\pi},\lambda,h}](s,a) - [\underline{Q}^t_h - Q^\star_{\widetilde{\pi},\lambda,h}](s,a) \\ &\le p_h\left[\overline{V}^t_{h+1} - \underline{V}^t_{h+1}\right](s,a) + 4\mathcal{B}\|\psi(s,a)\|_{[\Lambda^t_h]^{-1}}.\end{aligned} \qquad (18)$$

Next, we have for any $h' \in [H]$ and any $s' \in \mathcal{S}$

$$\left[\overline{V}^t_{h'} - \underline{V}^t_{h'}\right](s') \le \bar{\pi}^{t+1}_h\left[\overline{Q}^t_{h'} - \underline{Q}^t_{h'}\right](s'),$$

therefore we can roll-out the equation (18) and obtain

$$\overline{Q}^t_h(s,a) - \underline{Q}^t_h(s,a) \le 4\mathcal{B}\mathbb{E}_{\bar{\pi}^{t+1}}\left[\sum_{h'=h}^{H}\|\psi(s_{h'},a_{h'})\|_{[\Lambda^t_{h'}]^{-1}}\Big|(s_h,a_h)=(s,a)\right].$$

Next, we apply Lemma 35 for any fixed $h \in [H]$

$$\mathbb{E}_{\bar{\pi}^{t+1}}\left[\max_{a\in\mathcal{A}}\left(\overline{Q}^t_h - \underline{Q}^t_h\right)^2(s_h,a)|s_1\right] = \mathbb{E}_{\pi^{t+1,(h)}}\left[\left(\overline{Q}^t_h - \underline{Q}^t_h\right)^2(s_h,a_h)|s_1\right]$$

and for any $h' \ge h$

$$\mathbb{E}_{\bar{\pi}^{t+1}}\left[\|\psi(s_{h'},a_{h'})\|_{[\Lambda^t_{h'}]^{-1}}\Big|(s_h,a_h)=(s,a)\right] = \mathbb{E}_{\pi^{t+1,(h)}}\left[\|\psi(s_{h'},a_{h'})\|_{[\Lambda^t_{h'}]^{-1}}\Big|(s_h,a_h)=(s,a)\right].$$

Therefore, applying Jensen's inequality to conditional measure and the tower property of conditional expectation

$$\begin{aligned}\mathbb{E}_{\bar{\pi}^{t+1}}\left[\max_{a\in\mathcal{A}}\left(\overline{Q}^t_h - \underline{Q}^t_h\right)^2(s_h,a)|s_1\right] &\le 16\mathcal{B}^2\mathbb{E}_{\pi^{t+1,(h)}}\left[\left(\sum_{h'=h}^{H}\|\psi(s_{h'},a_{h'})\|_{[\Lambda^t_{h'}]^{-1}}\right)^2|s_1\right] \\ &\le 16\mathcal{B}^2 H\sum_{h'=h}^{H}\mathbb{E}_{\pi^{t+1,(h)}}\left[\|\psi(s_{h'},a_{h'})\|^2_{[\Lambda^t_{h'}]^{-1}}|s_1\right] \\ &\le 16\mathcal{B}^2 H\sum_{h'=1}^{H}\mathbb{E}_{\pi^{t+1,(h)}}\left[\|\psi(s_{h'},a_{h'})\|^2_{[\Lambda^t_{h'}]^{-1}}|s_1\right].\end{aligned}$$

Summing over all $h$ and recalling $\widetilde{\pi}^{t+1}$ as a mixture policy of all $\pi^{t+1,(h)}$

$$\mathbb{E}_{\overline{\pi}^{t+1}}\left[\max_{a\in\mathcal{A}}\left(\overline{Q}_h^t - \underline{Q}_h^t\right)^2(s_h,a)|s_1\right] \leq 16\mathcal{B}^2 H \sum_{h'=1}^{H}\sum_{h=1}^{H}\mathbb{E}_{\pi^{t+1,(h)}}\left[\|\psi(s_{h'},a_{h'})\|^2_{[\Lambda_{h'}^t]^{-1}}|s_1\right]$$

$$= 16\mathcal{B}^2 H^2 \sum_{h'=1}^{H}\mathbb{E}_{\pi^{\mathrm{mix},t+1}}\left[\|\psi(s_{h'},a_{h'})\|^2_{[\Lambda_{h'}^t]^{-1}}|s_1\right].$$

**Step 2. Summing sub-optimality gaps**   Next, we sum all sub-optimality gaps and obtain

$$\sum_{t=1}^{T}\{V^\star_{\widetilde{\pi},\lambda,1}(s_1) - V^{\overline{\pi}^t}_{\widetilde{\pi},\lambda,1}(s_1)\} \leq H + \sum_{t=1}^{T-1}\{V^\star_{\widetilde{\pi},\lambda,1}(s_1) - V_1^{\overline{\pi}^{t+1}}(s_1)\}$$

$$\leq H + \frac{16\mathcal{B}^2 H^2}{\lambda}\sum_{h'=1}^{H}\sum_{t=1}^{T-1}\mathbb{E}_{\pi^{\mathrm{mix},t+1}}\left[\|\psi(s_{h'},a_{h'})\|^2_{[\Lambda_{h'}^t]^{-1}}|s_1\right].$$

Next, we apply the definition of event $\mathcal{E}^{\mathrm{cnt}}(\delta)$

$$\Lambda_h^t \succcurlyeq \frac{1}{2}\overline{\Lambda}_h^t - \beta^{\mathrm{cnt}}(\delta,T)I_d \Rightarrow [\Lambda_h^t]^{-1} \preccurlyeq 2\left[\overline{\Lambda}_h^t - 2\beta^{\mathrm{cnt}}(\delta,T)I_d\right]^{-1}.$$

Notice that by the choice of $\alpha = 2(\beta^{\mathrm{cnt}}(\delta,t)+1)$ we have

$$\widetilde{\Lambda}_h^t \triangleq \overline{\Lambda}_h^t - 2\beta^{\mathrm{cnt}}(\delta,T)I_d = 2I_d + \sum_{\tau=1}^{t}\mathbb{E}_{\pi^{\mathrm{mix},\tau}}[\psi(s_h,a_h)[\psi(s_h,a_h)]^\mathsf{T}].$$

Thus, we may apply Lemma 31

$$\sum_{t=1}^{T-1}\mathbb{E}_{\widetilde{\pi}^{t+1}}\left[\|\psi(s_{h'},a_{h'})\|^2_{[\Lambda_{h'}^t]^{-1}}|s_1\right] \leq 2\sum_{t=1}^{T-1}\mathbb{E}_{\pi^{\mathrm{mix},t+1}}\left[\|\psi(s_{h'},a_{h'})\|^2_{[\widetilde{\Lambda}_{h'}^t]^{-1}}|s_1\right]$$

$$\leq 4\log\det\left(\widetilde{\Lambda}_{h'}^T\right).$$

To upper bound the determinant, we upper bound the operator norm using the triangle inequality

$$\|\widetilde{\Lambda}_h^T\|_2 = \left\|2I_d + \sum_{\tau=1}^{T-1}\mathbb{E}_{\widetilde{\pi}^\tau}[\psi(s_h,a_h)[\psi(s_h,a_h)]^\mathsf{T}]\right\|_2 \leq 2 + (T-1),$$

therefore, combining with a definition of $\mathcal{B}$ given in (17) we have

$$\sum_{t=1}^{T}\{V^\star_{\widetilde{\pi},\lambda,1}(s_1) - V^{\overline{\pi}^t}_{\widetilde{\pi},\lambda,1}(s_1)\} \leq H + \frac{64\cdot 32^2 d^2 H^5\cdot\log(T+1)}{\lambda}\cdot\log\left(\frac{24\mathrm{e}dHT}{\delta}\right),$$

yielding

$$V^\star_{\widetilde{\pi},\lambda,1}(s_1) - V^{\widehat{\pi}}_{\widetilde{\pi},\lambda,1}(s_1) \leq \frac{H}{T} + \frac{64\cdot 32^2 d^2 H^5\cdot\log(T+1)}{\lambda T}\cdot\log\left(\frac{24\mathrm{e}dHT}{\delta}\right).$$

In particular, this implies that

$$T = \widetilde{\mathcal{O}}\left(\frac{H^5 d^2}{\lambda\varepsilon}\right)$$

is enough to obtain $\varepsilon$-accurate policy. $\qquad\qquad\qquad\qquad\qquad\qquad\qquad\qquad\qquad\qquad\qquad\qquad\square$

## F  DEMONSTRATION-REGULARIZED PREFERENCE-BASED LEARNING

### F.1  MAXIMUM LIKELIHOOD ESTIMATION FOR REWARD MODEL

In this section, we discuss the maximum likelihood estimation problem for the reward estimation, following Zhan et al. (2023a). Let $\mathcal{G}$ be a function class of reward functions that satisfies the following assumption

**Assumption 6.** For a function class $\mathcal{G}$ we assume that the true reward belongs to it: $r^\star \in \mathcal{G}$. Additionally, for the following function family $\mathcal{Q} = \{q_r(\tau_0, \tau_1) = \sigma(r(\tau_1) - r(\tau_0)) : r \in \mathcal{G}\}$ equipped with an $\ell_\infty$-norm has a finite *bracketing dimension*, that means there is a $d_r > 0$ and $R_r > 0$ such that

$$\forall \varepsilon \in (0, 1) : \log \mathcal{N}_{[]}(\varepsilon, \mathcal{Q}, \|\cdot\|_\infty) \le d_\mathcal{G} \log(R_\mathcal{G}/\varepsilon).$$

The bracketing numbers are commonly used in statistics for MLE, M-estimation, and, more generally, in the empirical processes theory, see van de Geer (2000). Related to our setting, this assumption is satisfied in the setting of tabular MDPs with a dimension $d_\mathcal{G} = SAH$ and linear MDPs with dimensions $d_\mathcal{G} = dH$, see Lemmas 19-20.

For each $r \in \mathcal{G}$ and a pair of trajectories $(\tau_0, \tau_1)$ define the induced preference model as follows

$$q_r(\tau_0, \tau_1) \triangleq \sigma(r(\tau_1) - r(\tau_0)).$$

To measure the complexity of the reward class $\mathcal{G}$ we will use the bracketing numbers of the function class $\mathcal{Q} = \{q_r : \mathcal{T} \times \mathcal{T} \to [0, 1] : r \in \mathcal{F}_r\}$, where $\mathcal{T} = (\mathcal{S} \times \mathcal{A})^H$ is a space of all trajectories. See Definition 7 for the definition of bracketing numbers.

Given the dataset of preferences $\mathcal{D}^{\mathrm{RM}} = \{(\tau_0^k, \tau_1^k, o^k)\}_{k=1}^{N^{\mathrm{RM}}}$, we define the maximum likelihood estimate of the reward model as follows

$$\hat{r} = \arg\max_{r \in \mathcal{G}} \left\{ \sum_{k=1}^{N^{\mathrm{RM}}} o^k \log q_r(\tau_0^k, \tau_1^k) + (1 - o^k) \log(1 - q_r(\tau_0^k, \tau_1^k)) \right\}. \tag{19}$$

The following result is a standard result on MLE estimation and, generally, M-estimation, see van de Geer (2000). The proof heavily uses PAC-Bayes techniques by Zhang (2006), see also Agarwal et al. (2020) for non-i.i.d. extension. We notice that a similar result could be extracted from Lemma 2 by Zhan et al. (2023a), however, we did not find the proof of exactly this statement in their paper or references within.

**Proposition 4.** Let Assumptions 4-6 hold. Let $\mathcal{D}^{\mathrm{RM}}$ be a preference dataset and assume that trajectories $\tau_0^k$ and $\tau_1^k$ were generated i.i.d. by following the policy $\pi$. Then for any $\delta \in (0, 1)$ with probability at least $1 - \delta$ the following bound for the MLE reward estimate $\hat{r}$ given by solution to 19 holds

$$\mathbb{E}_{\tau_0, \tau_1 \sim q^\pi} \left[ \left( q_\star(\tau_0, \tau_1) - q_{\hat{r}}(\tau_0, \tau_1) \right)^2 \right] \le \frac{2 + 2d_\mathcal{G} \log(R_\mathcal{G}/N^{\mathrm{RM}}) + \log(1/\delta)}{N^{\mathrm{RM}}},$$

where $q^\pi$ is a distribution over trajectories induced by policy $\pi$

*Proof.* In the sequel, we drop the superscript from $\mathcal{D}^{\mathrm{RM}}$ and $N^{\mathrm{RM}}$ to simplify the notation.

Let us consider a maximal set of $\varepsilon$-brackets $B$ of size $\mathcal{N}_{[]}(\varepsilon, \mathcal{Q}, \|\cdot\|_\infty)$ and apply Lemma 2.1 by Zhang (2006) (or Lemma 21 by Agarwal et al. (2020)), where consider brackets $[\ell, u]$ from $B$ as parameters $\theta$, prior $\pi$ is a uniform over $B$, and the density $w_\mathcal{D}([\ell, u])$ is equal to a (properly weighted) Dirac measure on a bracket $[\ell^\star, u^\star]$ that contains the MLE estimate (ties are resolved arbitrary). It implies that for any function $\mathcal{L} : B \times \mathcal{D} \to \mathbb{R}$ it holds

$$\mathbb{E}_\mathcal{D} \left[ \exp \left\{ \mathcal{L}(\mathcal{D}, [\ell^\star, u^\star]) - \log \mathbb{E}_{\mathcal{D}'} \left[ e^{\mathcal{L}(\mathcal{D}', [\ell^\star, u^\star])} \right] - \log \mathcal{N}_{[]}(\varepsilon, \mathcal{Q}, \|\cdot\|_\infty) \right\} \right] \le 1,$$

where $\mathcal{D}'$ is an independent copy of the dataset $\mathcal{D}$ and the KL-divergence between a Dirac measure on an MLE bracket and the uniform distribution is computed exactly. Since we can control the exponential moment, by a simple Chernoff argument we have with probability at last $1 - \delta$

$$-\log \mathbb{E}_{\mathcal{D}'} \left[ e^{\mathcal{L}(\mathcal{D}', [l^\star, u^\star])} \right] \le -\mathcal{L}(\mathcal{D}, [l^\star, u^\star]) + \log \mathcal{N}_{[]}(\varepsilon, \mathcal{Q}, \|\cdot\|_\infty) + \log(1/\delta).$$

Next we choose $\mathcal{L}(\mathcal{D}, [\ell, u])$ as log-likelihood ratio

$$\mathcal{L}(\mathcal{D}, [\ell, u]) = -\frac{1}{2}\sum_{k=1}^{N}\left\{\frac{o^k \log q_\star(\tau_0^k, \tau_1^k) + (1 - o^k)\log(1 - q_\star(\tau_0^k, \tau_1^k))}{o^k \log u(\tau_0^k, \tau_1^k) + (1 - o^k)\log(1 - \ell(\tau_0^k, \tau_1^k))}\right\}.$$

By the choice of a brackets $[\ell, u]$ it holds $\ell^\star(\tau_0, \tau_1) \le q_{\hat{r}}(\tau_0, \tau_1) \le u^\star(\tau_0, \tau_1)$. Using the fact that $\hat{r}$ is a solution to (19)

$$-\mathcal{L}(\mathcal{D}, [\ell^\star, u^\star]) = \frac{1}{2}\sum_{k=1}^{N}\left\{\frac{o^k \log q_\star(\tau_0^k, \tau_1^k) + (1 - o^k)\log(1 - q_\star(\tau_0^k, \tau_1^k))}{o^k \log u(\tau_0^k, \tau_1^k) + (1 - o^k)\log(1 - \ell(\tau_0^k, \tau_1^k))}\right\}$$

$$\le \frac{1}{2}\sum_{k=1}^{N}\left\{\frac{o^k \log q_\star(\tau_0^k, \tau_1^k) + (1 - o^k)\log(1 - q_\star(\tau_0^k, \tau_1^k))}{o^k \log q_{\hat{r}}(\tau_0^k, \tau_1^k) + (1 - o^k)\log(1 - q_{\hat{r}}(\tau_0^k, \tau_1^k))}\right\} \le 0.$$

At the same time

$$-\log\mathbb{E}_{\mathcal{D}'}\left[e^{\mathcal{L}(\mathcal{D}', [l^\star, u^\star])}\right] = -N\log\mathbb{E}\left[\exp\left\{-\frac{1}{2}\frac{o\log q_\star(\tau_0, \tau_1) + (1 - o)\log(1 - q_\star(\tau_0, \tau_1))}{o\log u^\star(\tau_0, \tau_1) + (1 - o)\log(1 - \ell^\star(\tau_0, \tau_1))}\right\}\right],$$

where in the last expectation $\tau_0, \tau_1 \sim q^\pi, o \sim \mathcal{B}er(q_\star(\tau_0, \tau_1))$.

By Fubini's theorem, we have

$$-\frac{1}{N}\log\mathbb{E}_{\mathcal{D}'}\left[e^{\mathcal{L}(\mathcal{D}', [l^\star, u^\star])}\right] = -\log\mathbb{E}_{\tau_0, \tau_1}\left[\sqrt{q_\star(\tau_0, \tau_1)u(\tau_0, \tau_1)}\right.$$

$$\left. + \sqrt{(1 - q_\star(\tau_0, \tau_1))(1 - \ell(\tau_0, \tau_1))}\right].$$

Next, we study the expression under the square root. By the definition of the $\varepsilon$-bracket we have $\ell^\star(\tau_0, \tau_1) \le q_{\hat{r}}(\tau_0, \tau_1) \le u^\star(\tau_0, \tau_1)$ and $u^\star(\tau_0, \tau_1) - \ell^\star(\tau_0, \tau_1) \le \varepsilon$, therefore

$$\sqrt{q_\star(\tau_0, \tau_1)u(\tau_0, \tau_1)} \le \sqrt{q_\star(\tau_0, \tau_1)(q_{\hat{r}}(\tau_0, \tau_1) + \varepsilon)} \le \sqrt{q_\star(\tau_0, \tau_1)q_{\hat{r}}(\tau_0, \tau_1)} + \sqrt{\varepsilon}.$$

and the similar bound for the second term. Applying inequality $-\log(x) \ge 1 - x$ we have

$$-\frac{1}{N}\log\mathbb{E}_{\mathcal{D}'}\left[e^{\mathcal{L}(\mathcal{D}', [l^\star, u^\star])}\right] \ge 1 - \mathbb{E}_{\tau_0, \tau_1}\left[\sqrt{q_\star(\tau_0, \tau_1)q_{\hat{r}}(\tau_0, \tau_1)}\right.$$

$$\left. + \sqrt{(1 - q_\star(\tau_0, \tau_1))(1 - q_{\hat{r}}(\tau_0, \tau_1))}\right] - 2\sqrt{\varepsilon}.$$

By the properties of the Hellinger distance $d_{\mathcal{H}}$ (see Section 2.4 and Lemma 2.3 by Tsybakov 2008) we have

$$1 - \mathbb{E}_{\tau_0, \tau_1}\left[\sqrt{q_\star(\tau_0, \tau_1)q_{\hat{r}}(\tau_0, \tau_1)} + \sqrt{(1 - q_\star(\tau_0, \tau_1))(1 - q_{\hat{r}}(\tau_0, \tau_1))}\right]$$

$$= \frac{1}{2}\mathbb{E}_{\tau_0, \tau_1}\left[d_{\mathcal{H}}^2(\mathcal{B}er(q_\star(\tau_0, \tau_1)), \mathcal{B}er(q_{\hat{r}}(\tau_0, \tau_1)))\right] \ge \mathbb{E}_{\tau_0, \tau_1}\left[(q_\star(\tau_0, \tau_1) - q_{\hat{r}}(\tau_0, \tau_1))^2\right].$$

Overall, we obtain

$$\mathbb{E}_{\tau_0, \tau_1}\left[(q_\star(\tau_0, \tau_1) - q_{\hat{r}}(\tau_0, \tau_1))^2\right] \le 2\sqrt{\varepsilon} + \frac{\log\mathcal{N}_{[]}(\varepsilon, \mathcal{Q}, \|\cdot\|_\infty) + \log(1/\delta)}{N}.$$

Taking $\varepsilon = 1/N^2$ and applying the upper bound on the bracketing number by Assumption 6 we conclude the statement. $\square$

And also we have a simple corollary of this result that shows convergence of the reward models.

**Theorem 7.** *Let Assumptions 4-6 hold. Let $\mathcal{D}^{\mathrm{RM}}$ be a preference dataset and assume that trajectories $\tau_0^k$ and $\tau_1^k$ were generated i.i.d. by following the policy $\pi$. Then for any $\delta \in (0, 1)$ with probability at least $1 - \delta$ the following bound holds*

$$\mathbb{E}_{\tau_0, \tau_1 \sim q^\pi}\left[\left(r_\star(\tau_1) - r_\star(\tau_0) - \hat{r}(\tau_1) + \hat{r}(\tau_0)\right)^2\right] \le \frac{2\zeta^2 d_{\mathcal{G}}\log(R_{\mathcal{G}}/N^{\mathrm{RM}}) + \zeta^2\log(e^2/\delta)}{N^{\mathrm{RM}}},$$

*where $\zeta = 1/(\inf_{x \in [-H, H]}\sigma'(x))$ the non-linearity measure of the link function $\sigma$ defined in Assumption 4.*

**Remark 7.** We additionally notice that since two trajectories are i.i.d., we have

$$\mathbb{E}_{\tau_0,\tau_1 \sim q^\pi}\left[\left([r^\star(\tau_1) - r^\star(\tau_0)] - [\hat{r}(\tau_1) - \hat{r}(\tau_0)]\right)^2\right] = 2\mathrm{Var}_{q^\pi}[r^\star(\tau_1) - \hat{r}(\tau_1)].$$

In particular, it means that if the reward function is estimated up to a constant shift, then the MLE estimation error is zero.

**Remark 8.** In the setting of a sigmoid link function $\sigma(x) = 1/(1 + \exp(-x))$ we have $\zeta = \exp\{\Theta(H)\}$, yields the exponential dependence on the reward scaling. This could be avoided by using a more problem-dependent way to obtain a bound on rewards given bound on the induced preference model.

*Proof.* Follows from the application of Proposition 4 and the following application of mean-value theorem for any two fixed $\tau_0, \tau_1$

$$[r^\star(\tau_1) - r^\star(\tau_0)] - [\hat{r}(\tau_1) - \hat{r}(\tau_0)] = \sigma^{-1}(q_\star(\tau_0,\tau_1)) - \sigma^{-1}(q_{\hat{r}}(\tau_0,\tau_1))$$
$$= (\sigma^{-1})'(\xi)[q_\star(\tau_0,\tau_1) - q_{\hat{r}}(\tau_0,\tau_1)],$$

where $\xi$ is a point between $q_\star(\tau_0,\tau_1)$ and $q_{\hat{r}}(\tau_0,\tau_1)$. The observation $(\sigma^{-1})'(\xi) = 1/\sigma'(\sigma^{-1}(\xi))$ yields the statement. $\qquad\square$

Next, we compute the required quantities $d_{\mathcal{G}}$ for the case of finite and linear MDPs for a choice of $\sigma = 1/(1 + \exp(-x))$ as a sigmoid function.

**Lemma 19.** *Let a reward function $\{r_h(s,a)\}_{h\in[H]}$ be an arbitrary function $r_h \colon \mathcal{S} \times \mathcal{A} \to [0,1]$. Let us define $\mathcal{G} = \{r(\tau) = \sum_{h=1}^H r_h(s_h,a_h)\}$. Then Assumption 6 holds with constants $d_{\mathcal{G}} = HSA$ and $R_{\mathcal{G}} = 3H/2$.*

*Proof.* Let us define a functional class of interest $\mathcal{Q} = \{q_r(\tau_1,\tau_2) = \sigma(r(\tau_1) - r(\tau_2)) \mid r \in \mathcal{G}\}$ Since $\sigma$ is a monotonically increasing function that satisfies $\sigma'(x) \leq 1/4$. Thus, by combination of Lemma 21 and Lemma 22 we have

$$\mathcal{N}_{[]}(\varepsilon, \mathcal{Q}, \|\cdot\|_\infty) \leq \mathcal{N}_{[]}(2\varepsilon, \mathcal{G}, \|\cdot\|_\infty).$$

Next we define a function classes $\mathcal{F}_h = \{r_h \colon \mathcal{S} \times \mathcal{A} \to [0,1]\}$ of one-step rewards. By Lemma 23 it holds

$$\mathcal{N}_{[]}(2\varepsilon, \mathcal{G}, \|\cdot\|_\infty) \leq \prod_{h=1}^H \mathcal{N}_{[]}(2\varepsilon/H, \mathcal{F}_h, \|\cdot\|_\infty).$$

Then we can associate a function space $\mathcal{F}_h$ with a parameters $\Theta_h = [0,1]^{SA}$ and by Lemma 24 and a standard results in bounding of covering numbers of balls in normed spaces, see van Handel (2016),

$$\mathcal{N}_{[]}(\varepsilon, \mathcal{F}_h, \|\cdot\|_\infty) \leq \mathcal{N}(\varepsilon/2, [0,1]^{SA}, \|\cdot\|_\infty) \leq (3/\varepsilon)^{SA}.$$

As a result, we have

$$\log \mathcal{N}_{[]}(\varepsilon, \mathcal{Q}, \|\cdot\|_\infty) \leq HSA \log(3H/(2\varepsilon)).$$

$\qquad\square$

**Lemma 20.** *Let a reward function $\{r_h(s,a)\}_{h\in[H]}$ be parametrized as $r_h(s,a) = \psi(s,a)^\top \theta_h$ for $\psi \colon \mathcal{S} \times \mathcal{A} \to \mathbb{R}^d$ that satisfies $\|\psi(s,a)\|_2 \leq 1$, and $\theta_h \in \Theta_h$ for $\Theta_h = \{\theta_h \in \mathbb{R}^d \mid \|\theta_h\| \leq \sqrt{d}\}$. Let us define $\mathcal{G} = \{r(\tau) = \sum_{h=1}^H r_h(s_h,a_h)\}$. Then Assumption 6 holds with constants $d_{\mathcal{G}} = dH$ and $R_{\mathcal{G}} = 3H\sqrt{d}/2$.*

*Proof.* Let us define one-step rewards as follows $\mathcal{F}_h = \{r_h(s,a) = \psi(s,a)^\top \theta_h \mid \theta_h \in \mathbb{R}^d, \|\theta_h\| \leq \sqrt{d}\}$. Then following exactly the same reasoning as in Lemma 19

$$\mathcal{N}_{[]}(\varepsilon, \mathcal{Q}, \|\cdot\|_\infty) \leq \prod_{h=1}^H \mathcal{N}_{[]}(2\varepsilon/H, \mathcal{F}_h, \|\cdot\|_\infty).$$

Next we notice that $\mathcal{F}_h$ is Lipschtiz in $\ell_2$-norm with respect to parameters $\theta_h$ with a constant 1:

$$|r_h(s,a) - r_h'(s,a)| = |\psi(s,a)^\top(\theta_h - \theta_h')| \leq \|\theta_h - \theta_h'\|_2.$$

Therefore, since $\Theta_h$ is a ball of radius $\sqrt{d}$ we obtain

$$\mathcal{N}_{[]}(\varepsilon, \mathcal{F}_h, \|\cdot\|_\infty) \leq \mathcal{N}(\varepsilon/2, \Theta_h, \|\cdot\|_2) \leq (3\sqrt{d}/\varepsilon)^d.$$

As a result, we have

$$\log \mathcal{N}_{[]}(\varepsilon, \mathcal{Q}, \|\cdot\|_\infty) \leq dH \log(3H\sqrt{d}/(2\varepsilon)).$$

$\square$

### F.2  Properties of Bracketing Numbers

In this section, we provide a list of elementary properties of bracketing numbers for completeness. See Section 7 of Dudley (2014) for additional information.

**Lemma 21.** *Let $(\mathcal{G}, \|\cdot\|_\infty)$ be a normed space of functions over a set $\mathcal{X}$ that takes values in the interval $I$ and let $\sigma\colon I \to [0,1]$ be a monotonically increasing link function that satisfies $\sup_{x\in I} \sigma'(x) \leq C$ for $C > 0$. Then for $\mathcal{F} = \sigma \circ \mathcal{G} = \{f = \sigma \circ g \mid g \in \mathcal{G}\}$ it holds for any $\varepsilon > 0$*

$$\mathcal{N}_{[]}(\varepsilon, \mathcal{F}, \|\cdot\|_\infty) \leq \mathcal{N}_{[]}(\varepsilon/C, \mathcal{G}, \|\cdot\|_\infty).$$

*Proof.* Let $G$ be a minimal set of $\varepsilon$ brackets that covers $\mathcal{G}$. Then let us take a set of brackets $F = \{[\sigma \circ \ell, \sigma \circ u] : [l, u] \in G\}$. The set of $F$ consists of brackets since $\sigma$ is monotone, and it covers all the space $\mathcal{F}$. Additionally, we have

$$|\sigma \circ u(x) - \sigma \circ \ell(x)| = \sigma'(\xi)|(u(x) - \ell(x))| \leq C\varepsilon,$$

therefore the set $F$ consists of $C\varepsilon$-brackets. By rescaling we conclude the statement. $\square$

**Lemma 22.** *Let $(\mathcal{G}, \|\cdot\|_\infty)$ be a normed space of functions over a set $\mathcal{X}$ and let us define a set of functions over $\mathcal{X} \times \mathcal{X}$ as $\mathcal{F} = \{f(x, x') = g(x) - g(x') \mid g \in \mathcal{G}\}$. Then it holds*

$$\mathcal{N}_{[]}(\varepsilon, \mathcal{F}, \|\cdot\|_\infty) \leq \mathcal{N}_{[]}(\varepsilon/2, \mathcal{G}, \|\cdot\|_\infty).$$

*Proof.* Let $G$ be a minimal set of $\varepsilon$ brackets that covers $\mathcal{G}$. Let us define the following set of brackets

$$F = \{[f_\ell(x, x') = \ell(x) - u(x'), f_u(x, x') = u(x) - \ell(x')] \mid [\ell, u] \in G\}.$$

We may check that elements cover all the set $\mathcal{F}$. Let us take $f \in \mathcal{F}$ and the corresponding $g \in \mathcal{G}$. Then let us take a bracket $[\ell, u]$ that covers $g$. In this case, we have

$$f_\ell(x, x') = \ell(x) - u(x') \leq g(x) - g(x') \leq u(x) - \ell(x') = f_u(x, x').$$

At the same time, we have

$$\|f_\ell - f_u\|_\infty \leq 2\|u - \ell\|_\infty \leq 2\varepsilon.$$

Thus, $F$ is a set of $2\varepsilon$-brackets that covers $\mathcal{F}$. $\square$

**Lemma 23.** *Let $\{\mathcal{G}_k\}_{k=1}^K$ be a sequence of spaces of functions over a set $\mathcal{X}$ equipped with a norm $\|\cdot\|_\infty$ and let us define a set $\mathcal{F}$ of functions over $\mathcal{X}^K$ as follows*

$$\mathcal{F} = \left\{ f(x_1, \ldots, x_k) = \sum_{k=1}^K g_k(x_k) \mid g_k \in \mathcal{G} \right\}.$$

*Then the following bound holds*

$$\mathcal{N}_{[]}(\varepsilon, \mathcal{F}, \|\cdot\|_\infty) \leq \prod_{k=1}^K \mathcal{N}_{[]}(\varepsilon/K, \mathcal{G}, \|\cdot\|_\infty).$$

*Proof.* For any $k \in [K]$ let $G_k$ be a minimal set of $\varepsilon$ brackets that covers $\mathcal{G}_k$. Then we construct the set $F$ as follows

$$F = \left\{ \left[ f_\ell(x) = \sum_{k=1}^{K} \ell_k(x_k), f_u(x) = \sum_{k=1}^{k} u_k(x_k) \right] \mid \forall k \in [K] : [\ell_k, u_k] \in \mathcal{G}_k \right\}.$$

It holds that $|F| \leq \prod_{k=1}^{K} |G_k|$ and also it is clear that $F$ consists of brackets and covers all the set $\mathcal{F}$. Additionally, we notice that

$$\|f_\ell - f_u\|_\infty \leq \sum_{k=1}^{K} \|\ell_k - u_k\|_\infty \leq K\varepsilon.$$

By rescaling we conclude the statement. $\qquad\square$

**Lemma 24.** *Let $\Theta$ be a set of parameters and let $\mathcal{F} = \{f_\theta(x) \mid \theta \in \Theta\}$ be a set of functions over $\mathcal{X}$. Assume that $f_\theta(x)$ is $L$-Lipschitz in $\theta$ with respect to an arbitrary norm $\|\cdot\|$:*

$$\forall x \in \mathcal{X} : |f_\theta(x) - f_{\theta'}(x)| \leq L\|\theta - \theta'\|.$$

*Then we have*

$$\mathcal{N}_{[]}(\varepsilon, \mathcal{F}, \|\cdot\|_\infty) \leq \mathcal{N}(\varepsilon/(2L), \Theta, \|\cdot\|).$$

*Proof.* Let $X$ be a $\varepsilon$-covering of $\Theta$ and let us define a set $F = \{[\ell = f_\theta - \varepsilon L, u = f_\theta + \varepsilon L] \mid \theta \in X\}$. Let us show that $F$ consists of brackets. Let $f_\theta \in \mathcal{F}$, then there is $\hat{\theta} \in X$. By Lipchitzness

$$\forall x \in \mathcal{X} : |f_\theta(x) - f_{\hat{\theta}}(x)| \leq L\|\theta - \hat{\theta}\| = L\varepsilon,$$

therefore a bracket that corresponding $\ell$ and $u$ indeed satisfy $\ell(x) \leq f_\theta(x) \leq u(x)$ for any $x \in \mathcal{X}$. Also, we notice that $\|\ell - u\|_\infty = 2\varepsilon L$, therefore by rescaling we conclude the statement. $\qquad\square$

### F.3 PROOF FOR DEMONSTRATION-REGULARIZED RLHF

During this section we assume that the MDP is finite, i.e. $|\mathcal{S}| < +\infty$ to simplify the manipulations with the trajectory space. However, the state space could be arbitrarily large.

In this section, we provide the proof for the demonstration-regularized RLHF pipeline defined in Algorithm 2. We start from the general oracle version of this inequality. For a reward function $r$ let the value with respect to this value be defined as follows

$$V_h^\pi(s; r) = \mathbb{E}_\pi \left[ \sum_{h'=h}^{H} r_{h'}(s_{h'}, a_{h'}) \mid s_h = s \right], \quad V_{\widetilde{\pi}, \lambda, h}^\pi(s; r) = V_h^\pi(s; r) - \lambda \mathrm{KL}_{\mathrm{traj}}(\widetilde{\pi}\|\pi).$$

A similar definition holds for $Q$-values. Additionally, let us define the following coefficient defined by Zhan et al. (2023a) that additionally shows the closeness of $\pi^\mathrm{E}$ and $\pi^\mathrm{BC}$ in terms of the family of reward functions.

$$C_r(\mathcal{G}, \pi^\mathrm{E}, \pi^\mathrm{BC}) \triangleq \max \left\{ 0, \sup_{r \in \mathcal{G}} \frac{\mathbb{E}_{\tau_0 \sim \pi^\mathrm{E}, \tau_1 \sim \pi^\mathrm{BC}}[r^\star(\tau_0) - r^\star(\tau_1) - (r(\tau_0) - r(\tau_1))]}{\sqrt{\mathbb{E}_{\tau_0, \tau_1 \sim \pi^\mathrm{BC}}\left[ (r^\star(\tau_0) - r^\star(\tau_1) - (r(\tau_0) - r(\tau_1)))^2 \right]}} \right\}. \quad (20)$$

We notice that the denominator could be written as

$$\mathbb{E}_{\tau^0, \tau^1 \sim \pi^\mathrm{BC}}\left[ (r^\star(\tau_0) - r^\star(\tau_1) - (r(\tau_0) - r(\tau_1)))^2 \right] = 2\mathrm{Var}_{q^{\pi^\mathrm{BC}}}[r^\star - \hat{r}].$$

**Theorem 8.** *Let us assume that there is an underlying reward function $r^\star$ such that*

1. *There is an expert policy $\pi^\mathrm{E}$ such that*

$$V_1^\star(s_1; r^\star) - V_1^{\pi^\mathrm{E}}(s_1; r^\star) \leq \varepsilon_\mathrm{E};$$

2. *There is a behavior cloning policy $\pi^{\mathrm{BC}}$ that satisfies*

$$\sqrt{\mathrm{KL}_{\mathrm{traj}}(\pi^{\mathrm{E}}\|\pi^{\mathrm{BC}})} \leq \varepsilon_{\mathrm{KL}};$$

3. *There is an estimate of reward function $\hat{r}$ that satisfies*

$$\sqrt{\mathrm{Var}_{q^{\pi^{\mathrm{BC}}}}[r^{\star}(\tau) - \hat{r}(\tau)]} \leq \varepsilon_{\mathrm{RM}};$$

*Let $\pi^{\mathrm{RL}}$ be a $\varepsilon_{\mathrm{RL}}$-optimal policy in the $\lambda$-regularized MDP for $\pi^{\mathrm{BC}}$ and using $\hat{r}$ as rewards:*

$$V^{\star}_{\pi^{\mathrm{BC}},\lambda,1}(s_1;\hat{r}) - V^{\pi^{\mathrm{RL}}}_{\pi^{\mathrm{BC}},\lambda,1}(s_1;\hat{r}) \leq \varepsilon_{\mathrm{RL}}.$$

*Then we have the following optimality guarantees for $\pi^{\mathrm{RL}}$ in the unregularized MDP equipped with the true reward function*

$$V^{\star}_1(s_1;r^{\star}) - V^{\pi^{\mathrm{RL}}}_1(s_1;r^{\star}) \leq \frac{\varepsilon^2_{\mathrm{RM}}}{\lambda} + 2\varepsilon_{\mathrm{E}} + 2\varepsilon_{\mathrm{RL}} + 2\lambda\varepsilon^2_{\mathrm{KL}} + 3\varepsilon_{\mathrm{RM}}\varepsilon_{\mathrm{KL}} + \frac{4H}{3}\varepsilon^2_{\mathrm{KL}}.$$

*Moreover, if we assume that $C_r(\mathcal{G}, \pi^{\mathrm{E}}, \pi^{\mathrm{BC}})$ is finite, we have*

$$V^{\star}_1(s_1;r^{\star}) - V^{\pi^{\mathrm{RL}}}_1(s_1;r^{\star}) \leq \frac{\varepsilon^2_{\mathrm{RM}}}{\lambda} + 2\varepsilon_{\mathrm{E}} + 2\varepsilon_{\mathrm{RL}} + 2\lambda\varepsilon^2_{\mathrm{KL}} + 3C_r(\mathcal{G}, \pi^{\mathrm{E}}, \pi^{\mathrm{BC}}) \cdot \varepsilon_{\mathrm{RM}}.$$

*Additionally, under the choice $\lambda^{\star}$ as a positive solution to the equation $2\varepsilon^2_{\mathrm{KL}}(\lambda^{\star})^2 = 2\lambda\varepsilon_{\mathrm{RL}} + \varepsilon^2_{\mathrm{RM}}$ we have the following bound*

$$V^{\star}_1(s_1;r^{\star}) - V^{\pi^{\mathrm{RL}}}_1(s_1;r^{\star}) \leq 2\varepsilon_{\mathrm{E}} + 4\varepsilon_{\mathrm{RL}} + 6\varepsilon_{\mathrm{KL}}\varepsilon_{\mathrm{RM}} + \min\{2H\varepsilon^2_{\mathrm{KL}}, C_r \cdot \varepsilon_{\mathrm{RM}}\},$$

*where $C_r = C_r(\mathcal{G}, \pi^{\mathrm{E}}, \pi^{\mathrm{BC}})$ is a concentrability coefficient.*

*Proof.* We start from the following decomposition, using the assumption on the expert policy

$$V^{\star}_1(s_1;r^{\star}) - V^{\pi^{\mathrm{RL}}}_1(s_1;r^{\star}) \leq V^{\pi^{\mathrm{E}}}_1(s_1;r^{\star}) - V^{\pi^{\mathrm{RL}}}_1(s_1;r^{\star}) + \varepsilon_{\mathrm{E}}.$$

Next, we change the reward function as follows

$$V^{\pi^{\mathrm{E}}}_1(s_1;r^{\star}) - V^{\pi^{\mathrm{RL}}}_1(s_1;r^{\star}) = \underbrace{V^{\pi^{\mathrm{E}}}_1(s_1;\hat{r}) - V^{\pi^{\mathrm{RL}}}_1(s_1;\hat{r})}_{(\mathbf{A})} + \underbrace{V^{\pi^{\mathrm{E}}}_1(s_1;r^{\star} - \hat{r}) - V^{\pi^{\mathrm{RL}}}_1(s_1;r^{\star} - \hat{r})}_{(\mathbf{B})}.$$

We start from the analysis of the term $(\mathbf{A})$. By properties of the behavior cloning policy and the BPI policy $\pi^{\mathrm{RL}}$

$$(\mathbf{A}) = V^{\pi^{\mathrm{E}}}_{\pi^{\mathrm{BC}},\lambda,1}(s_1;\hat{r}) + \lambda\,\mathrm{KL}_{\mathrm{traj}}(\pi^{\mathrm{E}}\|\pi^{\mathrm{BC}}) - V^{\pi^{\mathrm{RL}}}_{\pi^{\mathrm{BC}}}(s_1;\hat{r}) - \lambda\,\mathrm{KL}_{\mathrm{traj}}(\pi^{\mathrm{RL}}\|\pi^{\mathrm{BC}})$$

$$\leq V^{\star}_{\pi^{\mathrm{BC}},\lambda,1}(s_1;\hat{r}) - V^{\pi^{\mathrm{RL}}}_{\pi^{\mathrm{BC}}}(s_1;\hat{r}) + \lambda\varepsilon^2_{\mathrm{KL}} \leq \varepsilon_{\mathrm{RL}} + \lambda\varepsilon^2_{\mathrm{KL}}.$$

Next, we have to analyze the second term $(\mathbf{B})$. We decompose it as follows

$$(\mathbf{B}) = \underbrace{V^{\pi^{\mathrm{E}}}_1(s_1;r^{\star} - \hat{r}) - V^{\pi^{\mathrm{BC}}}_1(s_1;r^{\star} - \hat{r})}_{(\mathbf{C})} + \underbrace{V^{\pi^{\mathrm{BC}}}_1(s_1;r^{\star} - \hat{r}) - V^{\pi^{\mathrm{RL}}}_1(s_1;r^{\star} - \hat{r})}_{(\mathbf{D})}.$$

We start from the analysis of $(\mathbf{D})$ because the analysis of term $(\mathbf{C})$ depends on the concentrability assumption.

For the term $(\mathbf{D})$ we can apply the second part of Lemma 32 since the space of the trajectories is finite

$$(\mathbf{D}) = \mathbb{E}_{q^{\pi^{\mathrm{BC}}}}[r^{\star}(\tau) - \hat{r}(\tau)] - \mathbb{E}_{q^{\pi^{\mathrm{RL}}}}[r^{\star}(\tau) - \hat{r}(\tau)]$$

$$\leq \sqrt{2\mathrm{Var}_{q^{\pi^{\mathrm{BC}}}}[r^{\star}(\tau) - \hat{r}(\tau)] \cdot \mathrm{KL}_{\mathrm{traj}}(\pi^{\mathrm{RL}}\|\pi^{\mathrm{BC}})} \leq \sqrt{\varepsilon^2_{\mathrm{RM}} \cdot \mathrm{KL}_{\mathrm{traj}}(\pi^{\mathrm{RL}}\|\pi^{\mathrm{BC}})},$$

where we applied an assumption on the reward estimate. Next, we have to estimate the trajectory KL-divergence between $\pi^{\mathrm{RL}}$ and $\pi^{\mathrm{BC}}$.

Let us apply the $\varepsilon$-optimality of the policy $\pi^{\mathrm{RL}}$ with respect to reward $\hat{r}$ in the regularized MDP

$$V_1^{\pi^{\mathrm{E}}}(s_1; \hat{r}) - \lambda \mathrm{KL}_{\mathrm{traj}}(\pi^{\mathrm{E}} \| \pi^{\mathrm{BC}}) \le V_1^{\pi^{\mathrm{RL}}}(s_1; \hat{r}) + \varepsilon_{\mathrm{RL}} - \lambda \mathrm{KL}_{\mathrm{traj}}(\pi^{\mathrm{RL}} \| \pi^{\mathrm{BC}}).$$

By rerranging the terms we have

$$\begin{aligned}
\mathrm{KL}_{\mathrm{traj}}(\pi^{\mathrm{RL}} \| \pi^{\mathrm{BC}}) &\le \frac{1}{\lambda} \Big( V_1^{\pi^{\mathrm{RL}}}(s_1; \hat{r}) - V_1^{\pi^{\mathrm{E}}}(s_1; \hat{r}) + \varepsilon_{\mathrm{RL}} \Big) + \varepsilon_{\mathrm{KL}}^2 \\
&\le \frac{\varepsilon_{\mathrm{E}} + \varepsilon_{\mathrm{RL}}}{\lambda} + \frac{1}{\lambda} \Big( V_1^{\pi^{\mathrm{RL}}}(s_1; \hat{r} - r^\star) - V_1^{\pi^{\mathrm{E}}}(s_1; \hat{r} - r^\star) \Big) + \varepsilon_{\mathrm{KL}}^2 \\
&= \frac{1}{\lambda} \Big( V_1^{\pi^{\mathrm{E}}}(s_1; r^\star - \hat{r}) - V_1^{\pi^{\mathrm{RL}}}(s_1; r^\star - \hat{r}) \Big) + \varepsilon_{\mathrm{KL}}^2 + \frac{\varepsilon_{\mathrm{E}} + \varepsilon_{\mathrm{RL}}}{\lambda}.
\end{aligned}$$

We notice that $(\mathbf{B}) \triangleq V_1^{\pi^{\mathrm{E}}}(s_1; r^\star - \hat{r}) - V_1^{\pi^{\mathrm{RL}}}(s_1; r^\star - \hat{r})$, then we have the following recursion

$$(\mathbf{B}) \le (\mathbf{C}) + \sqrt{\frac{\varepsilon_{\mathrm{RM}}^2}{\lambda} \cdot ((\mathbf{B}) + (\varepsilon_{\mathrm{E}} + \varepsilon_{\mathrm{RL}} + \lambda \varepsilon_{\mathrm{KL}}^2))}.$$

Let us denote $t^2 = (\mathbf{B}) + \varepsilon_{\mathrm{E}} + \varepsilon_{\mathrm{RL}} + \lambda \varepsilon_{\mathrm{KL}}^2$, $a = \sqrt{\varepsilon_{\mathrm{RM}}^2/\lambda}$, $b = (\mathbf{C}) + \varepsilon_{\mathrm{E}} + \varepsilon_{\mathrm{RL}} + \lambda \varepsilon_{\mathrm{KL}}^2$. Then we have the standard quadratic inequality in $t$, the maximal solution of which could be upper bounded as $t^2 \le a^2 + 2b$ which implies

$$(\mathbf{B}) \le \frac{\varepsilon_{\mathrm{RM}}^2}{\lambda} + \varepsilon_{\mathrm{E}} + \varepsilon_{\mathrm{RL}} + \lambda \varepsilon_{\mathrm{KL}}^2 + 2(\mathbf{C}).$$

Next, we analyze the term $(\mathbf{C})$.

**Without concentrability assumption** By the first part of Lemma 32 we have

$$\begin{aligned}
(\mathbf{C}) &= \mathbb{E}_{q^{\pi^{\mathrm{E}}}}[r^\star(\tau) - \hat{r}(\tau)] - \mathbb{E}_{q^{\pi^{\mathrm{BC}}}}[r^\star(\tau) - \hat{r}(\tau)] \\
&\le \sqrt{2 \mathrm{Var}_{q^{\pi^{\mathrm{BC}}}} \cdot \mathrm{KL}_{\mathrm{traj}}(\pi^{\mathrm{E}} \| \pi^{\mathrm{BC}})} + \frac{2H}{3} \mathrm{KL}_{\mathrm{traj}}(\pi^{\mathrm{E}} \| \pi^{\mathrm{BC}}) \le \sqrt{2} \cdot \varepsilon_{\mathrm{RM}} \varepsilon_{\mathrm{KL}} + \frac{2H}{3} \varepsilon_{\mathrm{KL}}^2.
\end{aligned}$$

Overall, we have the final rates

$$V_1^\star(s_1; r^\star) - V_1^{\pi^{\mathrm{RL}}}(s_1; r^\star) \le \frac{\varepsilon_{\mathrm{RM}}^2}{\lambda} + 2\varepsilon_{\mathrm{E}} + 2\varepsilon_{\mathrm{RL}} + 2\lambda \varepsilon_{\mathrm{KL}}^2 + 3\varepsilon_{\mathrm{RM}} \varepsilon_{\mathrm{KL}} + \frac{4H}{3} \varepsilon_{\mathrm{KL}}^2.$$

**With concentrability assumption** Now we assume that $C_r(\mathcal{G}, \pi^{\mathrm{E}}, \pi^{\mathrm{BC}})$ is finite. In this situation, we can upper bound $(\mathbf{C})$ as follows

$$\begin{aligned}
(\mathbf{C}) &= \mathbb{E}_{\tau_0 \sim q^{\pi^{\mathrm{E}}}, \tau_1 \sim q^{\pi^{\mathrm{BC}}}}[r^\star(\tau_0) - r^\star(\tau_1) + \hat{r}(\tau_1) - \hat{r}(\tau_0)] \\
&\le \sqrt{2 C_r^2(\mathcal{G}, \pi^{\mathrm{E}}, \pi^{\mathrm{BC}}) \cdot \mathrm{Var}_{q^{\pi^{\mathrm{BC}}}}[r^\star - \hat{r}]} \le \sqrt{2} \cdot C_r(\mathcal{G}, \pi^{\mathrm{E}}, \pi^{\mathrm{BC}}) \cdot \varepsilon_{\mathrm{RM}}.
\end{aligned}$$

Overall, we have the final rates

$$V_1^\star(s_1; r^\star) - V_1^{\pi^{\mathrm{RL}}}(s_1; r^\star) \le \frac{\varepsilon_{\mathrm{RM}}^2}{\lambda} + 2\varepsilon_{\mathrm{E}} + 2\varepsilon_{\mathrm{RL}} + 2\lambda \varepsilon_{\mathrm{KL}}^2 + 3C_r(\mathcal{G}, \pi^{\mathrm{E}}, \pi^{\mathrm{BC}}) \cdot \varepsilon_{\mathrm{RM}}.$$

**Under the choice of $\lambda^\star$** To show the last part of the statement, we choose $\lambda^\star$ as a solution to the following quadratic equation $2(\lambda^\star)^2 \varepsilon_{\mathrm{KL}}^2 = 2\lambda^\star \varepsilon_{\mathrm{RL}} + \varepsilon_{\mathrm{RM}}^2$. In particular, its positive solution is equal to

$$\lambda^\star = \frac{\varepsilon_{\mathrm{RL}} + \sqrt{\varepsilon_{\mathrm{RL}}^2 + 2\varepsilon_{\mathrm{KL}}^2 \varepsilon_{\mathrm{RM}}^2}}{2\varepsilon_{\mathrm{KL}}^2}.$$

Applying firstly the quadratic formula and then the exact formula above we have

$$\frac{\varepsilon_{\mathrm{RM}}^2}{\lambda^\star} + 2\varepsilon_{\mathrm{RL}} + 2\lambda^\star \varepsilon_{\mathrm{KL}}^2 = 4\lambda^\star \varepsilon_{\mathrm{KL}}^2 = 2\varepsilon_{\mathrm{RL}} + 2\sqrt{\varepsilon_{\mathrm{RL}}^2 + 2\varepsilon_{\mathrm{KL}}^2 \varepsilon_{\mathrm{RM}}^2} \le 4\varepsilon_{\mathrm{RL}} + 3\varepsilon_{\mathrm{KL}} \varepsilon_{\mathrm{RM}}.$$

Combining this bound with the bound on performance we have

$$V_1^\star(s_1; r^\star) - V_1^{\pi^{\mathrm{RL}}}(s_1; r^\star) \le 2\varepsilon_{\mathrm{E}} + 4\varepsilon_{\mathrm{RL}} + 6\varepsilon_{\mathrm{KL}} \varepsilon_{\mathrm{RM}} + \min\{2H\varepsilon_{\mathrm{KL}}^2, C_r \cdot \varepsilon_{\mathrm{RM}}\},$$

where $C_r = C_r(\mathcal{G}, \pi^{\mathrm{E}}, \pi^{\mathrm{BC}})$ is a concentrability coefficient.

$\square$

# G   DEVIATION INEQUALITIES

## G.1   DEVIATION INEQUALITY FOR CATEGORICAL DISTRIBUTIONS

Next, we state the deviation inequality for categorical distributions by Jonsson et al. (2020, Proposition 1). Let $(X_t)_{t\in\mathbb{N}^\star}$ be i.i.d. samples from a distribution supported on $\{1,\ldots,m\}$, of probabilities given by $p\in\Delta_{m-1}$, where $\Delta_{m-1}$ is the probability simplex of dimension $m-1$. We denote by $\widehat{p}_n$ the empirical vector of probabilities, i.e., for all $k\in\{1,\ldots,m\}$,

$$\widehat{p}_{n,k} \triangleq \frac{1}{n}\sum_{\ell=1}^{n}\mathbb{1}\{X_\ell = k\}.$$

Note that an element $p\in\Delta_{m-1}$ can be seen as an element of $\mathbb{R}^{m-1}$ since $p_m = 1 - \sum_{k=1}^{m-1}p_k$. This will be clear from the context.

**Theorem 9.** *For all $p\in\Delta_{m-1}$ and for all $\delta\in[0,1]$,*

$$\mathbb{P}(\exists n\in\mathbb{N}^\star,\, n\,\mathrm{KL}(\widehat{p}_n,p) > \log(1/\delta) + (m-1)\log(e(1 + n/(m-1)))) \le \delta.$$

## G.2   DEVIATION INEQUALITY FOR SEQUENCE OF BERNOULLI RANDOM VARIABLES

Below, we state the deviation inequality for Bernoulli distributions by Dann et al. (2017, Lemma F.4). Let $\mathcal{F}_t$ for $t\in\mathbb{N}$ be a filtration and $(X_t)_{t\in\mathbb{N}^\star}$ be a sequence of Bernoulli random variables with $\mathbb{P}(X_t = 1|\mathcal{F}_{t-1}) = P_t$ with $P_t$ being $\mathcal{F}_{t-1}$-measurable and $X_t$ being $\mathcal{F}_t$-measurable.

**Theorem 10.** *For all $\delta > 0$,*

$$\mathbb{P}\left(\exists n:\ \sum_{t=1}^{n}X_t < \sum_{t=1}^{n}P_t/2 - \log\frac{1}{\delta}\right) \le \delta.$$

## G.3   DEVIATION INEQUALITY FOR BOUNDED DISTRIBUTIONS

Below, we state the self-normalized Bernstein-type inequality by Domingues et al. (2021b). Let $(Y_t)_{t\in\mathbb{N}^\star}$, $(w_t)_{t\in\mathbb{N}^\star}$ be two sequences of random variables adapted to a filtration $(\mathcal{F}_t)_{t\in\mathbb{N}}$. We assume that the weights are in the unit interval $w_t\in[0,1]$ and predictable, i.e. $\mathcal{F}_{t-1}$ measurable. We also assume that the random variables $Y_t$ are bounded $|Y_t|\le b$ and centered $\mathbb{E}[Y_t|\mathcal{F}_{t-1}] = 0$. Consider the following quantities

$$S_t \triangleq \sum_{s=1}^{t}w_sY_s,\quad V_t \triangleq \sum_{s=1}^{t}w_s^2\cdot\mathbb{E}\big[Y_s^2|\mathcal{F}_{s-1}\big],\quad\text{and}\quad W_t \triangleq \sum_{s=1}^{t}w_s$$

and let $h(x)\triangleq(x+1)\log(x+1) - x$ be the Cramér transform of a Poisson distribution of parameter 1.

**Theorem 11** (Bernstein-type concentration inequality)**.** *For all $\delta > 0$,*

$$\mathbb{P}\left(\exists t\ge 1, (V_t/b^2 + 1)h\left(\frac{b|S_t|}{V_t + b^2}\right)\ge \log(1/\delta) + \log(4e(2t+1))\right)\le\delta.$$

*The previous inequality can be weakened to obtain a more explicit bound: if $b\ge 1$ with probability at least $1-\delta$, for all $t\ge 1$,*

$$|S_t|\le\sqrt{2V_t\log(4e(2t+1)/\delta)} + 3b\log(4e(2t+1)/\delta).$$

## G.4   DEVIATION INEQUALITY FOR VECTOR-VALUED SELF-NORMALIZED PROCESSES

Next, we state Lemma D.4 by Jin et al. (2020). For any symmetric positive definite matrix $A$ we define $\|x\|_A = \sqrt{x^\intercal A x}$.

**Lemma 25** (Jin et al. (2020))**.** *Let $\{s_\tau\}_{\tau=1}^{\infty}$ be a stochastic process on the state space $\mathcal{S}$ adapted to a filtration $\{\mathcal{F}_\tau\}_{\tau=0}^{\infty}$. Let $\{X_\tau\}_{\tau=0}^{\infty}$ be an $\mathbb{R}^d$-valued stochastic process where $\psi_\tau$ is $\mathcal{F}$-predictable ($X_\tau$ is $\mathcal{F}_{\tau-1}$ measurable) and $\|X_\tau\|\le 1$. Let $\Lambda_t = \alpha I_d + \sum_{\tau=1}^{t}X_t X_t^\intercal$ and let $\mathcal{V}$ be a family of*

*function over the state-space $\mathcal{S}$ such that $\forall V \in \mathcal{V}, \forall s \in \mathcal{S} : 0 \leq V(s) \leq H$. Then for any $\delta \in (0, 1)$ with probability at least $1 - \delta$*

$$\forall t \in \mathbb{N}, \forall V \in \mathcal{V} : \left\| \sum_{\tau=1}^{t} X_\tau \{V(s_\tau) - \mathbb{E}_{\tau-1}[V(s_\tau)]\} \right\|_{\Lambda_t^{-1}}^2 \leq 4H^2 \left[ \frac{d}{2} \log\left( \frac{t+\alpha}{\alpha} \cdot \frac{|\mathcal{N}_\varepsilon|}{\delta} \right) \right] + \frac{8t^2\varepsilon^2}{\alpha},$$

*where $\mathcal{N}_\varepsilon$ is a minimal $\varepsilon$-cover of $\mathcal{V}$ with respect to the distance $\rho(V, V') = \sup_{s \in \mathcal{S}} |V(s) - V'(s)|$.*

Also, we state the result on the covering dimension of the class of bonus function, see Lemma D.6 by Jin et al. (2020) for a similar result.

**Lemma 26.** *Let $\mathcal{V}$ be a class of functions over $\mathcal{S}$ such that*

$$\forall s \in \mathcal{S} : V(s) = \text{clip}\left( \max_{\pi \in \Delta_\mathcal{A}} \left\{ \pi \left[ w^\intercal \psi(s, \cdot) + \beta \sqrt{\psi(s, \cdot)^\intercal \Lambda^{-1} \psi(s, \cdot)} \right] - \lambda \Phi_{h,s}(\pi) \right\}, 0, H \right)$$

*is parameterized by a tuple $(w, \beta, \Lambda)$ such that $\|w\| \leq L, \beta \in [0, B], \lambda_{\min}(\Lambda) \geq \alpha$. Assume $\|\psi(s, a)\| \leq 1$ for any $(s, a) \in \mathcal{S} \times \mathcal{A}$. Then the covering number $|\mathcal{N}_\varepsilon|$ of the function space $\mathcal{V}$ with respect to the distance $\rho(V, V') = \sup_{s \in \mathcal{S}} |V(s) - V'(s)|$ satisfies*

$$\log \mathcal{N}(\varepsilon, \mathcal{V}, \rho) \leq d \log\left( 1 + \frac{4L}{\varepsilon} \right) + d^2 \log\left( 1 + \frac{8d^{1/2}B^2}{\alpha\varepsilon^2} \right).$$

*Proof.* Following the approach of Jin et al. (2020), we reparametrize the following set by setting $A = \beta^2 \Lambda^{-1}$ and obtain

$$V(s) = \text{clip}\left( \max_{\pi \in \Delta_\mathcal{A}} \left\{ \pi \left[ w^\intercal \psi(s, \cdot) + \sqrt{\psi(s, \cdot)^\intercal A \psi(s, \cdot)} \right] - \lambda \Phi_{h,s}(\pi) \right\}, 0, H \right),$$

for $\|w\|_2 \leq L, \|A\|_2 \leq B^2 \alpha^{-1}$. Next, let $V_1, V_2 \in \mathcal{V}$ be two functions that corresponds to parameters $(w_1, A_1)$ and $(w_2, A_2)$. Then, using non-expanding property of $\text{clip}(\cdot, 0, H)$ and $\max_{\pi \in \Delta_\mathcal{A}} \{\cdot\}$ we have

$$\rho(V_1, V_2) \leq \sup_{s \in \mathcal{S}} \max_{\pi \in \Delta_\mathcal{A}} \pi \left[ (w_1^\intercal \psi(s, \cdot) + \sqrt{\psi(s, \cdot)^\intercal A_1 \psi(s, \cdot)}) - (w_2^\intercal \psi(s, \cdot) + \sqrt{\psi(s, \cdot)^\intercal A_2 \psi(s, \cdot)}) \right]$$

$$\leq \sup_{s,a \in \mathcal{S} \times \mathcal{A}} \left[ [w_1 - w_2]^\intercal \psi(s, a) + \sqrt{\psi(s, a)^\intercal A_1 \psi(s, a)}) - \sqrt{\psi(s, a)^\intercal A_2 \psi(s, a)} \right]$$

$$\leq \sup_{\psi:\|\psi\|\leq 1} |\langle w_1 - w_2, \psi \rangle| + \sup_{\psi:\|\psi\|\leq 1} \sqrt{|\psi^\intercal(A_1 - A_2)\psi|}$$

$$\leq \|w_1 - w_2\|_2 + \sqrt{\|A_1 - A_2\|_F}.$$

The rest of the proof follows Lemma D.6 by Jin et al. (2020) and uses the result on covering numbers of Euclidean balls in $\mathbb{R}^d$. $\qquad\square$

## G.5 DEVIATION INEQUALITY FOR SAMPLE COVARIANCE MATRICES

The following result generalizes Theorem 10 in the case of linear MDPs and generalized counters.

Let $\{X_t\}_{t=1}^\infty$ be a sequence of random vectors of dimension $d$ adapted to a filtration $\{\mathcal{F}_t\}_{t=1}^\infty$ such that $\|X_t\|_2 \leq 1$ a.s.. Define a sequence of positive semi-definite matrices $A_t = \mathbb{E}[X_t X_t^\intercal | \mathcal{F}_{t-1}]$. Notice that $\|A_t\|_2 = \sigma_{\max}(A_t) \leq 1$. Also define $\Lambda_t = \lambda I_d + \sum_{j=1}^t X_j X_j^\intercal$ and $\overline{\Lambda}_t = \lambda I_d + \sum_{j=1}^t A_j$.

**Lemma 27.** *Let $\delta \in (0, 1)$. Then the following event*

$$\mathcal{E}^{\text{cnt}'}(\delta) = \left\{ \forall t \geq 1 : \Lambda_t \succcurlyeq \frac{1}{2}\overline{\Lambda}_t - \beta(\delta, t) I_d \right\}$$

*under the choice $\beta(\delta, t) = 4 \log(4e(2t + 1)/\delta) + 4d \log(3t) + 3$ holds with probability at least $1 - \delta$.*

*Proof.* Let us fix a vector $v$ from a unit sphere $\|v\|_2 = 1$. Next, note that

$$v^\mathsf{T}\Lambda_t v - v^\mathsf{T}\overline{\Lambda}_t v = \sum_{j=1}^{t} \underbrace{\langle v, X_j\rangle^2 - \mathbb{E}\big[\langle v, X_j\rangle^2 | \mathcal{F}_{j-1}\big]}_{\Delta M_j}.$$

Notice that $\Delta M_j$ is a martingale-difference sequence that satisfied $|\Delta M_j| \leq 2$ and

$$\sum_{j=1}^{t} \mathbb{E}\big[\Delta M_j^2 | \mathcal{F}_{j-1}\big] = \sum_{j=1}^{t} \mathbb{E}\big[\Delta M_j^2 | \mathcal{F}_{j-1}\big] \leq \sum_{j=1}^{t} \mathbb{E}\big[\langle v, X_j\rangle^4 | \mathcal{F}_{j-1}\big] \leq v^\mathsf{T}\overline{\Lambda}_t v.$$

Therefore, applying self-normalized Bernstein inequality (Theorem 11) we have that with probability at least $1 - \delta$ for all $t \geq 1$

$$|v^\mathsf{T}\Lambda_t v - v^\mathsf{T}\overline{\Lambda}_t v| \leq \sqrt{2 v^\mathsf{T}\overline{\Lambda}_t v \cdot \log(4\mathrm{e}(2t+1)/\delta)} + 3\log(4\mathrm{e}(2t+1)/\delta).$$

Next, inequality $2ab \leq a^2 + b^2$ implies that

$$|v^\mathsf{T}\Lambda_t v - v^\mathsf{T}\overline{\Lambda}_t v| \leq \frac{1}{2} v^\mathsf{T}\overline{\Lambda}_t v + 4\log(4\mathrm{e}(2t+1)/\delta).$$

Let us denote by $\mathcal{N}_\varepsilon$ a $\varepsilon$-net over a unit sphere of dimension $d$. By union bound over this net we have with probability at least $1 - \delta$

$$\forall \hat{v} \in \mathcal{N}_\varepsilon : |\hat{v}^\mathsf{T}\Lambda_t \hat{v} - v^\mathsf{T}\overline{\Lambda}_t \hat{v}| \leq \frac{1}{2}\hat{v}^\mathsf{T}\overline{\Lambda}_t \hat{v} + 4\log(4\mathrm{e}(2t+1)/\delta) + 4\log(|\mathcal{N}_\varepsilon|).$$

Let $v$ be an arbitrary vector on a unit sphere and let $\hat{v} \in \mathcal{N}_\varepsilon$ be the closest vector to $v$ in the $\varepsilon$-net. Then for any matrix $A \in \mathbb{R}^{d \times d}$ we have

$$|v^\mathsf{T} A v - \hat{v}^\mathsf{T} A \hat{v}| \leq |v^\mathsf{T} A(v - \hat{v})| + |(v - \hat{v})^\mathsf{T} A \hat{v}| \leq 2\varepsilon \|A\|_2.$$

Next we notice that $\|\Lambda_t - \overline{\Lambda}_t\|_2 \leq 2t$ and $\|\overline{\Lambda}_t\| \leq t$. Then we have

$$\forall v \in \mathcal{S}^d : |v^\mathsf{T}\Lambda_t v - v\overline{\Lambda}_t v| \leq \frac{1}{2} v^\mathsf{T}\overline{\Lambda}_t v + 4\log(4\mathrm{e}(2t+1)/\delta) + 4\log(|\mathcal{N}_\varepsilon|) + 3t\varepsilon.$$

Finally, we have the upper bound on covering number $|\mathcal{N}_\varepsilon| \leq (3/\varepsilon)^d$ and, taking $\varepsilon = 1/t$ for all fixed $t \geq 1$ we have

$$\forall v \in \mathcal{S}^d : |v^\mathsf{T}\Lambda_t v - v\overline{\Lambda}_t v| \leq \frac{1}{2} v^\mathsf{T}\overline{\Lambda}_t v + 4\log(4\mathrm{e}(2t+1)/\delta) + 4d\log(3t) + 3.$$

Thus, with probability at least $1 - \delta$ we have for all $t \geq 1$

$$\frac{3}{2}\overline{\Lambda}_t + \beta(\delta, t)I_d \succcurlyeq \Lambda_t \succcurlyeq \frac{1}{2}\overline{\Lambda}_t - \beta(\delta, t)I_d.$$

where $\beta(\delta, t) = 4\log(4\mathrm{e}(2t+1)/\delta) + 4d\log(3t) + 3$.

$\square$

# H  TECHNICAL LEMMAS

## H.1  COUNTS TO PSEUDO-COUNTS

Here we state Lemma 8 and Lemma 9 by Ménard et al. (2021).

**Lemma 28.** *On event* $\mathcal{E}^{\mathrm{cnt}}$, *for any* $\beta(\delta, \cdot)$ *such that* $x \mapsto \beta(\delta, x)/x$ *is non-increasing for* $x \geq 1$, $x \mapsto \beta(\delta, x)$ *is non-decreasing* $\forall h \in [H], (s, a) \in \mathcal{S} \times \mathcal{A}$,

$$\forall t \in \mathbb{N}^\star, \; \frac{\beta(\delta, n_h^t(s, a))}{n_h^t(s, a)} \wedge 1 \leq 4 \frac{\beta(\delta, \bar{n}_h^t(s, a))}{\bar{n}_h^t(s, a) \vee 1}.$$

**Lemma 29.** *For* $T \in \mathbb{N}^\star$ *and* $(u_t)_{t \in \mathbb{N}^\star}$, *for a sequence where* $u_t \in [0, 1]$ *and* $U_t \triangleq \sum_{l=1}^t u_\ell$, *we get*

$$\sum_{t=0}^T \frac{u_{t+1}}{U_t \vee 1} \leq 4 \log(U_{T+1} + 1).$$

## H.2  COUNTS TO PSEUDO-COUNTS IN LINEAR MDPS

Let $\{X_t\}_{t=1}^\infty$ be a sequence of random vectors of dimension $d$ adapted to a filtration $\{\mathcal{F}_t\}_{t=1}^\infty$ such that $\|X_t\|_2 \leq 1$ a.s.. Define a sequence of positive semi-definite matrices $A_t = \mathbb{E}[X_t X_t^\intercal | \mathcal{F}_{t-1}]$. Notice that $\|A_t\|_2 = \sigma_{\max}(A_t) \leq 1$. Also define $\Lambda_t = \lambda I_d + \sum_{j=1}^t X_j X_j^\intercal$ and $\overline{\Lambda}_t = \lambda I_d + \sum_{j=1}^t A_j$.

**Lemma 30.** *Let* $A \succeq 0$ *be a positive semi-definite matrix such that* $\|A\|_2 \leq 1$. *Then*

$$\log \det(I + A) \leq \mathrm{Tr}(A) \leq 2 \log \det(I + A).$$

*Proof.* Follows from eigendecomposition for $A$ and numeric inequality $\log(1+x) \leq x \leq 2 \log(1+x)$ for all $x \in [0, 1]$. $\square$

The next result generalized Lemma D.2 by Jin et al. (2020); see also (Abbasi-Yadkori et al., 2011). Also, it could be treated as a generalization of Lemma 29 for linear MDPs.

**Lemma 31.** *Let* $\|A_t\|_2 \leq 1$ *for all* $t \geq 1$. *Then for any* $T \geq 1$

$$\log \frac{\det(\overline{\Lambda}_T)}{\det(\overline{\Lambda}_0)} \leq \sum_{t=1}^T \mathbb{E}\big[X_t^\intercal [\overline{\Lambda}_{t-1}]^{-1} X_t | \mathcal{F}_{t-1}\big] \leq 2 \log \frac{\det(\overline{\Lambda}_T)}{\det(\overline{\Lambda}_0)}.$$

*Proof.* First, we notice that $\overline{\Lambda}_{t-1}$ is $\mathcal{F}_{t-1}$-measurable, thus

$$\mathbb{E}\big[X_t^\intercal [\overline{\Lambda}_{t-1}]^{-1} X_t | \mathcal{F}_{t-1}\big] = \mathbb{E}\big[\mathrm{Tr}\big([\overline{\Lambda}_{t-1}]^{-1} X_t X_t^\intercal\big) | \mathcal{F}_{t-1}\big] = \mathrm{Tr}\big([\overline{\Lambda}_{t-1}]^{-1} \mathbb{E}[X_t X_t^\intercal | \mathcal{F}_{t-1}]\big)$$
$$= \mathrm{Tr}\big([\overline{\Lambda}_{t-1}]^{-1} A_t\big) = \mathrm{Tr}\Big([\overline{\Lambda}_{t-1}]^{-1/2} A_t [\overline{\Lambda}_{t-1}]^{-1/2}\Big).$$

Notice that $\Sigma_t = [\overline{\Lambda}_{t-1}]^{-1/2} A_t [\overline{\Lambda}_{t-1}]^{-1/2}$ is positive semi-definite matrix. Then by Lemma 30

$$\log \det(I_d + \Sigma_t) \leq \mathbb{E}\big[X_t^\intercal [\overline{\Lambda}_{t-1}]^{-1} X_t | \mathcal{F}_{t-1}\big] \leq 2 \log \det(I_d + \Sigma_t).$$

At the same time, we have

$$\det(\overline{\Lambda}_t) = \det(\overline{\Lambda}_{t-1} + A_t) = \det(\overline{\Lambda}_{t-1}) \cdot \det\Big(I_d + [\overline{\Lambda}_{t-1}]^{-1/2} A_t [\overline{\Lambda}_{t-1}]^{-1/2}\Big).$$

Thus by telescoping property

$$\log \frac{\det(\overline{\Lambda}_T)}{\det(\overline{\Lambda}_0)} \leq \sum_{t=1}^T \mathbb{E}\big[X_t^\intercal [\overline{\Lambda}_{t-1}]^{-1} X_t | \mathcal{F}_{t-1}\big] \leq 2 \log \frac{\det(\overline{\Lambda}_T)}{\det(\overline{\Lambda}_0)}.$$

$\square$

### H.3 ON THE BERNSTEIN INEQUALITY

We restate here a Bernstein-type inequality by Talebi & Maillard (2018).

**Lemma 32** (Corollary 11 by Talebi & Maillard, 2018). *Let $p, q \in \Delta_S$, where $\Delta_S$ denotes the probability simplex of dimension $S$. For all functions $f : \mathcal{S} \mapsto [0, b]$ defined on $\mathcal{S}$,*

$$pf - qf \leq \sqrt{2\mathrm{Var}_q(f)\,\mathrm{KL}(p,q)} + \frac{2}{3}b\,\mathrm{KL}(p,q)\,,$$

$$qf - pf \leq \sqrt{2\mathrm{Var}_q(f)\,\mathrm{KL}(p,q)}\,.$$

*where use the expectation operator defined as $pf \triangleq \mathbb{E}_{s \sim p} f(s)$ and the variance operator defined as $\mathrm{Var}_p(f) \triangleq \mathbb{E}_{s \sim p}\big(f(s) - \mathbb{E}_{s' \sim p}f(s')\big)^2 = p(f - pf)^2$.*

**Lemma 33.** *Let $p, q \in \Delta_S$ and a function $f : \mathcal{S} \mapsto [0, b]$, then*

$$\mathrm{Var}_q(f) \leq 2\mathrm{Var}_p(f) + 4b^2\,\mathrm{KL}(p,q)\,,$$

$$\mathrm{Var}_p(f) \leq 2\mathrm{Var}_q(f) + 4b^2\,\mathrm{KL}(p,q).$$

**Lemma 34.** *For $p, q \in \Delta_S$, for $f, g : \mathcal{S} \mapsto [0, b]$ two functions defined on $\mathcal{S}$, we have that*

$$\mathrm{Var}_p(f) \leq 2\mathrm{Var}_p(g) + 2bp|f - g| \quad and$$

$$\mathrm{Var}_q(f) \leq \mathrm{Var}_p(f) + 3b^2\|p - q\|_1,$$

*where we denote the absolute operator by $|f|(s) = |f(s)|$ for all $s \in \mathcal{S}$.*

### H.4 CHANGE OF POLICY

Let $\pi$ and $\pi'$ be two Markovian policies and let $\pi^{(h')}$ for $h' \in [H]$ be a family of policies defined as follows

$$\pi_h^{(h')}(s) = \begin{cases} \pi_h(s) & h \neq h', \\ \pi'_h(s) & h = h'. \end{cases}$$

**Lemma 35.** *For any measurable function $f \colon \mathcal{S} \to \mathbb{R}$ and any $h \leq h'$ it holds*

$$\mathbb{E}_\pi[f(s_h)|s_1] = \mathbb{E}_{\pi^{(h')}}[f(s_h)|s_1].$$

*Moreover, for any measurable $g \colon \mathcal{S} \times \mathcal{A} \to \mathbb{R}$ and any $h \geq h'$*

$$\mathbb{E}_\pi[g(s_h, a_h)|(s_{h'}, a_{h'}) = (s, a)] = \mathbb{E}_{\pi^{(h')}}[g(s_h, a_h)|(s_{h'}, a_{h'}) = (s, a)].$$

*Proof.* At first, we recall that by definition of a kernel $p_h$ we have for any measurable $f$ :

$$\mathbb{E}_\pi[f(s_{h+1})|s_h, a_h] = p_h f(s_h, a_h), \quad \mathbb{E}_\pi[g(s_h, a_h)|s_h] = \pi_h g(s_h). \tag{21}$$

We show the first statement by induction over $h \leq h'$. For $h = 1$ the statement is trivial. Next, we assume that it holds for $h$ and we have to show for $h + 1 \leq h'$. By the tower property of conditional expectation and (21)

$$\mathbb{E}_\pi[f(s_{h+1})|s_1] = \mathbb{E}_\pi[\mathbb{E}_\pi[f(s_{h+1})|s_h, a_h]|s_1] = \mathbb{E}_\pi[p_h f(s_h, a_h)|s_1]$$
$$= \mathbb{E}_\pi[\mathbb{E}_\pi[p_h f(s_h, a_h)|s_h]|s_1] = \mathbb{E}_\pi[\pi_h[p_h f](s_h)|s_1].$$

We notice that since $h < h'$ then $\pi_h = \pi_h^{(h')}$. Then we can apply the induction hypothesis to a function $\pi_h^{(h')}[p_h f](s_h)$

$$\mathbb{E}_\pi[f(s_{h+1})|s_1] = \mathbb{E}_\pi[\pi_h[p_h f](s_h)|s_1] = \mathbb{E}_\pi[\pi_h^{(h')}[p_h f](s_h)|s_1] = \mathbb{E}_{\pi^{(h')}}[f(s_{h+1})|s_1].$$

To show the second statement, we use induction over all $h \geq h'$. For $h = h'$ the statement is trivial. Next, we assume that it holds for $h \geq h'$ and we have to show it for $h + 1$. Again, using the tower property

$$\mathbb{E}_\pi[g(s_{h+1}, a_{h+1})|s_{h'}, a_{h'}] = \mathbb{E}_\pi[\mathbb{E}_\pi[g(s_{h+1}, a_{h+1})|s_{h+1}]|s_{h'}, a_{h'}] = \mathbb{E}_\pi[\pi_{h+1}g(s_{h+1})|s_{h'}, a_{h'}].$$

We notice that $\pi_{h+1} = \pi_{h+1}^{(h')}$ since $h \geq h'$. Thus, again applying the tower property, (21), and induction hypothesis

$$
\begin{aligned}
\mathbb{E}_\pi[g(s_{h+1}, a_{h+1})|s_{h'}, a_{h'}] &= \mathbb{E}_\pi[\mathbb{E}_\pi[\pi_{h+1}^{(h')} g(s_{h+1})|s_h, a_h]|s_{h'}, a_{h'}] \\
&= \mathbb{E}_\pi[p_h[\pi_{h+1}^{(h')} g](s_h, a_h)|s_{h'}, a_{h'}] \\
&= \mathbb{E}_{\pi^{(h')}}[p_h[\pi_{h+1}^{(h')} g](s_h, a_h)|s_{h'}, a_{h'}] = \mathbb{E}_{\pi^{(h')}}[g(s_{h+1}, a_{h+1})|s_{h'}, a_{h'}].
\end{aligned}
$$

$\square$

