# OpenReview forum: "Demonstration-Regularized RL"
_ICLR.cc/2024/Conference — ICLR 2024 poster_

### Official Review · Reviewer_UXmS · 2023-10-23

**Soundness:** 3 good
**Presentation:** 3 good
**Contribution:** 3 good
**Rating:** 6
**Confidence:** 2

**Summary:**

This paper studies two new hybrid setting which are novel/uncommon in the literature: (i) demonstration regularized RL and (ii) Demonstration Regularized RLHF.

In (i) both expert demonstrations from an $\epsilon$-optimal policy and online access to an MDP with reward function are possible. In (ii) the reward function is not available but it can be inferred thanks to a Preference Based Model introduced in Assumption 4.

**Strengths:**

Both the newly introduced settings are interesting and matches practical situations.

**Weaknesses:**

There are several technical weaknesses in my opinion.

The main weakness is in my opinion that the setting seems

1) The lower bound in Theorem 2 turns unfortunately vacuous in the limit of $\gamma \rightarrow 0$.

2) It is unclear why the class of linear policies at the third line of Section 3.2 is considered to be not learnable. In fact, under this choice [1] proves in their Theorem 5 that it is possible to recover an $\epsilon$-suboptimal policy compared to the expert with behavioural cloning.

3) Corollary 3 requires the expert to be $\mathcal{O}(\epsilon)$ optimal but I would expect that, given the reward knowledge, it should be possible to prove a sample complexity bound without the assumption on the $\mathcal{O}(\epsilon)$ optimality of the expert.
To see this think to the case of any BPI algorithm which requires no expert at all to learn an $\epsilon$-optimal policy.

4) I think that Lemma 11 should be referred as the standard performance difference lemma.

5) Just before Corollary 3, it is said  that "UCBVI-Ent+ algorithm for regularized BPI. It is a modification of the algorithm UCBVI-Ent by
Tiapkin et al. (2023) with improvement sample complexity". However, it is not explained which is the crucial difference between the two algorithms. In particular, also the settings are different because UCBVI-Ent+ uses reward information while UCBVI-Ent can be used only for maximum entropy exploration and not to solve Regularized MDPs but this difference is not explained in the main text.


[1] ( Rajamaran et al., 2021 ) On the Value of Interaction and Function Approximation in Imitation Learning.

**Questions:**

Q1) Is it possible to prove a bound which does not require the assumption that the expert is $\epsilon$ optimal ?

Q2) Why the regularization is needed in the tabular case but not in the linear one ? I am referring to Section 3.2

Q3) How can UCBVI-Ent+ achieve $\mathcal{O}(\epsilon^{-1})$ sample complexity according to Theorem 5 while UCBVI-Ent achieves a worst sample complexity of $\mathcal{O}(\epsilon^{-2})$ ?

Q4) What are the definitions of $\pi^{t,(h)}$ and $\tilde{\pi}^t$ in Algorithm 3?

Q5) In the setting of Demonstration Regularized RLHF is it necessary to have the coefficients defined in equation 3 in the bound ?

---

> ### Author Response · Authors · 2023-11-14
> **Official Comment by Authors, Part 1**
>
> We would like to thank reviewer UXmS for the careful reading and the constructive feedback.  Please find below our response to the main points raised in the review.
>
> - The lower bound in Theorem 2 turns unfortunately vacuous in the limit of $\gamma \to 0$.
>
> A fixed $\gamma$ enables us to establish a general lower bound by constraining the space of policies we aim to learn. Thus, the minimax lower bound with respect to the space of all policies is lower-bounded by the minimax lower bound with respect to the space of restricted policies. Therefore, choosing any fixed $\gamma$ (for example, $\gamma = 1/N$) allows us to deduce a valid and reasonable lower bound.
>
> - It is unclear why the class of linear policies at the third line of Section 3.2 is considered to be not learnable. In fact, under this choice [1] proves in their Theorem 5 that it is possible to recover an $\varepsilon$-suboptimal policy compared to the expert with behavioural cloning.
>
> It is important to emphasize that the behavior cloning defined in our paper aims not to find an $\varepsilon$-optimal policy but to reconstruct the initial behavioral policy in trajectory KL-distance. To make the algorithm implementable, defining a hypothesis class with policies that are differentiable or at least continuous with respect to the learnable parameters becomes necessary. In the setting of purely greedy policies, the mentioned linear class of policies lacks these properties, thus preventing the efficient minimization of the log-loss. We will add an additional discussion and a reference to [1].
>
>
> - Corollary 3 requires the expert to be $O(\varepsilon)$-optimal but I would expect that, given the reward knowledge, it should be possible to prove a sample complexity bound without the assumption on the optimality of the expert. To see this think to the case of any BPI algorithm which requires no expert at all to learn an $O(\varepsilon)$-optimal policy.
>
>
> We acknowledge that the assumption of nearly optimal experts might be considered strong. If it is known a priori that the expert is far from optimal, a viable approach is to simply set $\lambda=0$ and employ a stopping rule from UCBVI-BPI (Ménard et al. (2021)).
>
> However, it's important to emphasize that the primary goal of this paper is to provide an analysis of an algorithm actively used in practice, where these assumptions are deemed reasonable.
>
>
> - I think that Lemma 11 should be referred as the standard performance difference lemma.
>
> We agree that Lemma 11 is already known in literature, so we will add the corresponding reference.
>
> - Just before Corollary 3, it is said that "UCBVI-Ent+ algorithm for regularized BPI. It is a modification of the algorithm UCBVI-Ent by Tiapkin et al. (2023) with improvement sample complexity". However, it is not explained which is the crucial difference between the two algorithms. In particular, also the settings are different because UCBVI-Ent+ uses reward information while UCBVI-Ent can be used only for maximum entropy exploration and not to solve Regularized MDPs but this difference is not explained in the main text.
>
> We add a separate comment (available to all reviewers) regarding the key differences between UCBVI-Ent and UCBVI-Ent+.

---

> > ### Author Response · Authors · 2023-11-14
> > **Official Comment by Authors, Part 2**
> >
> > - Q1) Is it possible to prove a bound which does not require the assumption that the expert is $\varepsilon$-optimal ?
> >
> > We believe that it is possible (in the setup of RL with demonstrations) since a usual BPI algorithm allows us to provide an optimal policy without any additional data. However, the resulting algorithm using  data from a highly suboptimal expert may be significantly different from the proposed one. So we focus on a simple and implementable approach.
> >
> > - Q2) Why the regularization is needed in the tabular case but not in the linear one ? I am referring to Section 3.2
> >
> > Regularization in the linear setting is unnecessary because we add constraints on the parameters of the class. Alternatively, one can introduce ridge regularization with an appropriate regularization parameter and make no direct restrictions on the regression coefficients. In fact, these two approaches are nearly equivalent.
> >
> > - Q3) How can UCBVI-Ent+ achieve $O(\varepsilon^{-1})$ sample complexity according to Theorem 5 while UCBVI-Ent achieves a worst sample complexity of $O(\varepsilon^{-2})$?
> >
> > We have included a separate comment detailing the distinctions between UCBVI-Ent and UCBVI-Ent+. The critical feature that enables UCBVI-Ent+ to achieve $O(\varepsilon^{-1})$ sample complexity is the exploitation of the strong convexity of the regularizer. However, to implement this effectively, it is necessary to introduce additional time randomization to the policy played during the episodes. We believe that this is a novel aspect of our work on the algorithmic side.
> >
> > - Q4) What are the definitions of $\pi^{t, (h)}$ and $\tilde{pi}$ in Algorithm 3?
> >
> > The definition of $\pi^{t, (h’)}$ is given after sampling rules (9) in Appendix D.2: $\pi^{t, (h’)}_h$ is equal to an regularized-greedy optimistic policy given by optimistic planning for all steps $h \not = h’$ and $\pi^{t, (h’)}_h$ is a greedy with respect to a difference between upper and lower estimate for optimal Q-value for a step $h = h’$. In particular, this randomization is crucial to achieve $\mathcal{O(\varepsilon^{-1})}$ sample complexity. Regarding $\tilde{pi}$, it is a misprint and the final policy should be $\bar{\pi}^t$, thank you for showing it to us.
> >
> > - Q5) In the setting of Demonstration Regularized RLHF is it necessary to have the coefficients defined in equation 3 in the bound ?
> >
> > As it is stated in Corollary 4, to achieve $O(\varepsilon) $ quality it is enough to have either large enough $N^{\text{E}}$ (independent on a coefficient defined in (3)) or large enough $N^{\text{RM}}$ depending on the coefficient defined in (3). If we assume only large enough $N^{\text{RM}}$, then the dependence on $C_r$ seems unavoidable due to the  lower bounds in Theorem 3 by Zhan et al. (2023a).

---

> > > ### Comment · Reviewer_UXmS · 2023-11-15
> > >
> > > Dear authors,
> > >
> > > Thanks for your detailed response !
> > >
> > > At the moment I don't feel like rising my score. I think there is value in the current submission but I think that the presentation of the results do not match the acceptance standard. Since this is a theoretical paper it would be important to convey the main new proof techniques to attain the improved results.
> > >
> > > However, this is not done in the main text but only in the very long Appendix.
> > >
> > > Be sure however that I will discuss with the other reviewers to check whether they share my viewpoint or not.
> > >
> > > Best,
> > >
> > > Reviewer UXmS

---

> > > > ### Author Response · Authors · 2023-11-18
> > > >
> > > > We are happy that you think that there is a value in the current submission and we are happy that we answered all your questions.
> > > > We added a separate comment devoted to theoretical techniques that we used and also we tried to explain our presentation choice. Also we would like to underline that the main contribution of this paper is not novel proof techniques but general ideas that generalizes the best policy identification and imitation learning with a simple and implementable algorithmic technique. If you can suggest a better way to introduce the results given a 9 pages constraint we would be happy to try to implement this organization.

---

> > > > > ### Comment · Reviewer_UXmS · 2023-11-20
> > > > >
> > > > > Dear Authors,
> > > > >
> > > > > Thanks for adding some discussions about the linear expert condition and explaining the differences between UCBVI-Ent and UCBVI-Ent+.
> > > > >
> > > > > I understand that the 9 pages limit can be restrictive for all your contributions. However, I think that the main text should be structures to convey a clearer presentation of maybe fewer contribution.
> > > > >
> > > > > To this end, my suggestion would be to first state the result in Corollary 3 using the existing results in Tiapkin et al. 2023 (UCBVI-Ent) and shows which would be the result in that case.
> > > > >
> > > > > Then, you can explain the algorithm UCBVI-Ent+ (adding the pseudocode in the main text) and show the improvement in sample complexity compared to the one obtained using UCBVI-Ent and explain that the reason for the improvement is the fact that UCBVI-Ent can leverage the strong convexity of the KL divergence.
> > > > >
> > > > > The current paragraph in red is still bit difficult to parse in my opinion and the pseudocode for UCBVI-Ent+ is needed.
> > > > >
> > > > > Probably, implementing these changes will require to move the RLHF part to the Appendix and just mention this extension in the main text.
> > > > >
> > > > > Implementing these changes would make the paper way easier to appreciate in my opinion.
> > > > >
> > > > > Finally, I have a question. Would have been possible to adapt RL-Explore-Ent to the regularised MDP setting to attain the same sample complexity without the need of introducing a new algorithm in this work ?
> > > > >
> > > > > Best,
> > > > > Reviewer UXmS

---

> > > > > > ### Author Response · Authors · 2023-11-20
> > > > > >
> > > > > > We would like to thank reviewer UXmS for the valuable suggestions on paper organization.
> > > > > >
> > > > > > To implement these suggestions, we reorganized Section 4 and included a description of the UCBVI-Ent+ algorithm as well as discussion on why the UCBVI-Ent algorithm cannot provide any acceleration using the expert demonstrations. The discussions and all the changes are highlighted in red.
> > > > > >
> > > > > > To free up some space, we removed a lower bound statement for behavior cloning and moved the definition of a coefficient $C_r$ to Appendix.
> > > > > >
> > > > > > - Would have been possible to adapt RL-Explore-Ent to the regularized MDP setting to attain the same sample complexity without the need of introducing a new algorithm in this work ?
> > > > > >
> > > > > > We do not think that it is possible to improve the sample complexity of RL-Explore-Ent algorithm from $O(H^8 S^4 A / (\lambda \varepsilon)$ to $O(H^5 S^2 A / (\lambda \varepsilon))$ without significant algorithmic changes. It may be possible to improve the dependence in $S$ for the RL-Explore-Ent bound by switching for a different base regret minimizer algorithm than Euler, but improving the dependance in $H$ seems much more challenging since Euler is optimal in its dependence on $H$. Furthermore, note that even if we manage to match the bound of UCBVI-Ent+, we would still prefer not to use the RL-Explore-Ent algorithm because of its very large space complexity. Indeed RL-Explore-Ent needs to store $O(H^8S^4/\epsilon)$ policies in order to construct the mixture while the space complexity of UCBVI-Ent+ is only of order $O(HS^2A)$. Another reason why we used UCBVI-Ent instead of RL-Explore-Ent is that it is easier to adapt to the linear setting. In general, reward-free algorithms in the linear setting become much more complicated, in contrast to our simple randomization idea.

---

> > > > > > > ### Comment · Reviewer_UXmS · 2023-11-20
> > > > > > >
> > > > > > > dear authors,
> > > > > > >
> > > > > > > thanks a lot for the revision !
> > > > > > >
> > > > > > > i think it solves my concerns about the presentation because now the randomization idea is well presented and it is easier to understand how lambda should be set to obtain the bound for the demonstration regularized RL setting.
> > > > > > >
> > > > > > > as a result i will rise my score to 6.
> > > > > > >
> > > > > > > I think that the main remaining limitation is the assumption that the expert is epsilon optimal. Hopefully the authors will address this question in some future work.
> > > > > > >
> > > > > > > Best,
> > > > > > > Reviewer

---

### Official Review · Reviewer_Muxw · 2023-10-31

**Soundness:** 3 good
**Presentation:** 3 good
**Contribution:** 3 good
**Rating:** 6
**Confidence:** 3

**Summary:**

This paper studies demonstration-regularized RL where an agent is supposed to find a near optimal policy given an offline dataset that is collected from an expert policy. The paper theoretically shows that given $N^E$ expert samples, the sample complexity of finding a $\epsilon$-optimal policy reduces by a factor of $1/N^E$ in both tabular and linear MDPs. Moreover, the paper extends the proposed method to RLHF and theoretically justify the efficiency of it.

**Strengths:**

1. The paper provides comprehensive theoretical results on various settings in demonstration-regularized RL.
2. The results are nice and show a strong benefit using expert demonstrations.

**Weaknesses:**

1. I prefer that there is a separated "related works" section such that the presentation is clear.
2. The contributions from the algorithm design part seem not significant. The algorithm is a combination of imitation learning and regularized RL.
3. The results highly depends on the performance of the expert policy. However, in real life applications, obtaining expert demonstrations is usually expensive, and there might be far less offline demonstrations that that considered in this paper. Specifically, from Corollary 3, it seems that the benefit occurs when $N^E>H^3SA$, which is close to the typical sample complexity in standard RL, which might be too much in real applications.

**Questions:**

While the paper provides the lower bound for the imitation learning, is there any lower bound for the regularized RL?

---

> ### Author Response · Authors · 2023-11-14
>
> We would like to thank reviewer Muxw for the careful reading and the constructive feedback.  Please find below our response to the main points raised in the review.
>
> - The contributions from the algorithm design part seem not significant. The algorithm is a combination of imitation learning and regularized RL.
>
> We are particularly interested in algorithms that are scalable and easily implementable. The simplicity of our algorithm enables its straightforward implementation beyond tabular and linear settings. Our primary focus in this work is to analyze the existing practical pipeline and demonstrate its provable efficiency.
>
> On the other hand, our algorithm, UCBVI-Ent+, introduces a novel time step randomization to control the widths of the confidence intervals, as described in a separate comment available to all reviewers. In our view, this represents a significant algorithmic contribution, enabling the achievement of optimal complexity bounds.
>
>
> - The results highly depends on the performance of the expert policy. However, in real life applications, obtaining expert demonstrations is usually expensive, and there might be far less offline demonstrations that that considered in this paper. Specifically, from Corollary 3, it seems that the benefit occurs when $N^E > H^3 SA$, which is close to the typical sample complexity in standard RL, which might be too much in real applications.
>
>
> Regarding Corollary 3, it is important to emphasize that with significantly less expert data, reconstructing the expert policy with a small error becomes impossible, as indicated in the lower bound (Theorem 2). The application of behavior cloning techniques makes no sense with such a limited amount of data. However, with a reasonable amount of expert data, our approach demonstrates convergence to the optimal policy, surpassing the behavior cloning outcome.
>
> In the RLHF setup, it's crucial to highlight that even a relatively small amount of expert data results in a straightforward and practical algorithm. This algorithm delivers a policy which is close  to the optimal one when provided with a reasonably large preference dataset. In contrast to existing theoretical algorithms for RLHF, our approach distinguishes itself by not necessitating the solution of complex min-max problems to achieve pessimism. Instead, it aligns more closely with approaches actively applied in practical settings.

---

> > ### Comment · Reviewer_Muxw · 2023-11-21
> >
> > Thanks for the response. After reading the revision and other general answers, I appreciate the contribution from the behavior cloning and the corresponding lower bound. In addition, other contribution also seems good. I have decided to raise my score to 6.

---

### Official Review · Reviewer_zApY · 2023-11-04

**Soundness:** 4 excellent
**Presentation:** 4 excellent
**Contribution:** 4 excellent
**Rating:** 8
**Confidence:** 2

**Summary:**

The paper studied demonstration-regularized reinforcement learning (RL), where the learner first performs behavioral cloning on expert-generated demonstrations using maximum likelihood estimation. Then during the online interaction with the underlying environment, the learner penalizes deviation of the learned policy from the one learned in the behavioral cloning phase. The paper provided a theoretical analysis of these two phases. For behavioral cloning, the authors show that the KL divergence between the learned policy and the expert policy decreases linearly as the number of demonstrations grows. This holds for both tabular MDPs and linear MDPs under certain assumptions. Then based on this result, the authors further studied the regularized online learning scenario and the RLHF setting. In both cases, the authors are able to prove a fast convergence rate for the proposed algorithms.

**Strengths:**

(1) The paper performed a strong and solid theoretical study of behavioral cloning for both tabular and linear MDPs. The authors proved that the KL-divergence between the learner policy and the expert policy decreases linearly as the number of demonstrations grows. The authors also complemented the above positive results with a lower bound on the convergence rate. In terms of the dependency on the number of demonstrates, the upper bound and lower bound match. This is a nice and great result. Based on that, the authors further performed analysis on their proposed demonstration-regularized RL algorithms and the RLHF algorithms, and both achieved surprisingly fast convergence rates. Overall, the paper has made significant technical contributions.

(2) The paper studied a very novel, interesting, yet challenging problem. The topic is of particular interest to the theoretical RL community, and I can forsee that the results of this paper significantly push the frontiers of RL theory and will drive more research along this line.

**Weaknesses:**

(1) It would be great to include some empirical studies, although this is purely a theory paper.

**Questions:**

(1) Can you provide some empirical results to validate the theoretical findings of this paper?

---

> ### Author Response · Authors · 2023-11-14
>
> We thank reviewer zApY for the positive and encouraging feedback. We will add numerical results for our tabular algorithm.

---

### Official Review · Reviewer_XPKe · 2023-11-06

**Soundness:** 3 good
**Presentation:** 3 good
**Contribution:** 2 fair
**Rating:** 6
**Confidence:** 4

**Summary:**

The authors propose KL-regularized online RL algorithms and provide an upper bound on the sample complexity of this algorithm in tabular MDP and linear MDP settings.

**Strengths:**

The authors provide a thorough analysis of the algorithms, to justify the efficiency of the RL algorithms with access to an expert dataset.

**Weaknesses:**

1. There is limited discussion on the relationship between the pure online learning version of LSVI-UCB and UCBVI+.
2. The paper can be better organized, there are too many references pointing towards the appendix.

**Questions:**

I am not an expert in RL theory, I am a bit confused by the results presented in Corollary3 and Theorem6, where the bound presented in Corollary3 depends on $N^E$, whereas the bound in Theorem6 does not. Intuitively, I suppose the sample complexity would eventually depend on the size of the expert dataset. Can authors explain this?

---

> ### Author Response · Authors · 2023-11-14
>
> We would like to thank reviewer XPKe for the careful reading and the constructive feedback.  Please find below our response to the main points raised in the review.
>
> - Limited discussion on LSVI-UCB-Ent and UCBVI-Ent algorithms;
>
> Unfortunately, the limited discussion on the regularized BPI algorithms is a byproduct of constrained space. We've included a general comment highlighting the novel features of UCBVI-Ent+ and LSVI-Ent algorithms compared to their non-regularized counterparts. Additionally, we present an adaptation of the UCBVI-Ent algorithm for general regularized MDPs in a separate comment dedicated to all the reviewers.
>
> Moreover, it's important to stress that the primary contribution of the paper lies not solely in the regularized BPI algorithms but in their combination with behavior cloning techniques. This combination illustrates the convergence properties of simple and implementable algorithms that are already widely utilized in practice.
>
> - The paper can be better organized, there are too many references pointing towards the appendix.
>
> We would greatly appreciate any suggestions from the reviewer regarding the reorganization of the main results of the paper, considering the constraint of a 9-page limit.
>
> - Why Theorem 6 does not depend on the size of the expert dataset $N^{\mathrm{E}}$?
>
> Theorem 6 is devoted to the sample complexity of best policy identification for KL-regularized MDPs with respect to any reference policy and any regularization coefficient $\lambda > 0$. To specialize these bounds to Corollary 3, we need to take a behavior-cloning policy $\pi^{\mathrm{BC}}$ as a reference one and choose the regularization parameter  $\lambda$ depending on the number of expert trajectories $N^{\mathrm{E}}$ (choice $\lambda^\star$ from Theorem 3) and the desired accuracy $\varepsilon$ as $\mathcal{O}(N^{\mathrm{E}} \varepsilon / (SAH))$ (as described in Corollary 3).

---

> > ### Author Response · Authors · 2023-11-20
> >
> > We would like to acknowledge reviewer XPKe that we included the description of algorithm UCBVI-Ent+ in the main text as well as a discussion on its difference with UCBVI-Ent and RL-Explore-Ent. Additionally, we reduced the number of references towards statements in the Appendix by including the sample complexity result for UCBVI-Ent+ and LSVI-UCB-Ent in the main text. All the changes are highlighted in red.

---

> > > ### Comment · Reviewer_XPKe · 2023-11-21
> > >
> > > Thank you for the clarification and the revision. I would like to raise my rating to 6.

---

### Author Response · Authors · 2023-11-14
**Clarification on a difference between UCBVI-Ent+ and UCBVI-Ent**

In this general comment we would like to make a general clarifications regarding BPI in regularized MDPs. First of all, we would like to acknowledge that each time we refer to UCBVI-Ent algorithm we refer to its version for general regularized MDPs, as it is mentioned in Appendix D.2 of Tiapkin et al. (2023).

Next, we would like to emphasize the differences between UCBVI-Ent and UCBVI-Ent+  allowing the latter algorithm to achieve sample complexity of order $\mathcal{O}(\varepsilon^{-1})$ which improves upon $\mathcal{O}(\varepsilon^{-2})$.

The UCBVI-Ent+ algorithm operates by sampling trajectories based on an exploratory version of the regularized-greedy policy. This exploratory policy corresponds to an optimistic solution of the regularized MDP, distinguishing it from the UCBVI and UCBVI-Ent algorithms, which employ the regularized-greedy policy without additional exploration.
The unique aspect of the UCBVI-Ent+ algorithm involves a novel randomization over the horizon, denoted as $H$. Specifically, at a randomly chosen time step, the policy acts greedily concerning a gap between upper and lower bounds on the optimal Q-value. The formal definition of this exploration policy, denoted as $\pi^{t,(h')}$, can be found in Appendix D.2 after equation (9). The compressed form of the played policy is referred to as $\pi^{\mathrm{mix}, t}$.


Importantly, this additional randomization introduces control over a gap value $G^t$, as defined in equation (10). Notably, $G^t$ scales *quadratically* with the width of a confidence interval on the optimal Q-value. This quadratic scaling is crucial for achieving fast rates, specifically $\mathcal{O}(\varepsilon^{-1})$. This efficiency is akin to the reward-free algorithm RL-Explore-Ent by Tiapkin et al. (2023), but with an improved dependence on $H$ and $S$. It's worth highlighting that the use of the standard greedy policy would not enable us to control  the expected width of the corresponding confidence intervals. This control is essential for exploiting the strong smoothness of our regularized MDP, ultimately leading to the achievement of fast rates.

LSVI-Ent algorithm also actively uses an explorative version of regularized-greedy policy, that allows us to control the quadratic quantity that appears in Lemma 19. As a result, the final rate for a linear case also scales with $\mathcal{O}(\varepsilon^{-1})$ instead of $\mathcal{O}(\varepsilon^{-2})$ in non-regularized setup.

---

### Author Response · Authors · 2023-11-18
**Comment on proof techniques, theoretical contribution, and presenetation choices**

We would like to emphasize a particular proof and algorithmic techniques used in our results and explain why we do not include some of the results in the main text. Unfortunately, due to space constraints we cannot include all these discussions in full length in the main body of the paper.
In a new rebuttal revision of the paper we have added a brief discussion on Bernstein conditions for behavior cloning, discussion on assumption for behavior cloning in the linear setting, and an important algorithm difference between UCBVI-Ent and UCBVI-Ent+ that allows to improve the convergence guarantees. It is highlighted by a red color.

- In Section B, we showed $O(1/N)$ convergence rates for behavior cloning. To achieve them, we first switched to  smoothed version of the  KL-divergence (see e.g. Hazan et al. (2019) for a similar idea) and then verified the so-called Bernstein condition (see e.g. Bartlett & Mendelson, 2006): we showed that the variance of the smoothed version of the policy log-likelihood ratio is controllable by just KL-divergence itself up to some logarithmic multiplicative factors and second-order additive factors. However, since the focus of our paper is mainly on reinforcement learning with demonstrations and preferences, we decided to not include the proof in the main text.

- In Section C, we introduced the setting of RL with demonstrations  and state the results on its convergence relying on the analysis of UCBVI-Ent+ and LSVI-Ent algorithms. We decided to focus on a novel demonstration-regularized setting instead of introducing the regularized BPI algorithms since the effect of fast rates is already studied in Tiapkin et al. (2023). This focus allowed us to  clearly demonstrate the usefulness of regularization in RL in more practical scenarios from a theoretical viewpoint. Nethertheless, we included a novel algorithmic idea of UCBVI-Ent+ into the main text of  the revised version.

- In Section D, we introduced RLHF with demonstrations and proposed a simple algorithm that shares similarity with real-world algorithms that have already been applied to fine-tuning of large language models (see Remark 6). To show the theoretical properties of these demonstration-regularized algorithms, we have to 1) show the theoretical properties of MLE estimate of the underlying reward function and 2) upper bound the policy error by applying Bernstein-type inequality introduced by Talebi & Maillard, (2018) and by verifying the Bernstein condition.

    * Theoretical properties of MLE estimate of reward function were already studied in the literature, see Lemma 2 by Zhan et al. (2023a) and we decided to not include them in the main text. However, we did not find the proof of the  statement we exactly need  in references and provided the proof for completeness in Appendix F.

    *  Regarding the bounding the policy error, we did not include the proof since it is very technical and heavily relies on the existing Bernstein-type inequalities in terms of KL-divergence by Talebi & Maillard, (2018).

However, if the reviewers have a different viewpoint and might suggest a better organization of the achieved results, we would be  happy to implement it.


Hazan, E., Kakade, S., Singh, K., & Van Soest, A. (2019, May). Provably efficient maximum entropy exploration. In International Conference on Machine Learning (pp. 2681-2691). PMLR.

Bartlett, P. L., & Mendelson, S. (2006). Empirical minimization. Probability theory and related fields, 135(3), 311-334.

---

### Meta-Review · Area_Chair_Ct85 · 2023-12-09

**Metareview:**

## Overall Assessment:
The paper presents a significant theoretical contribution to the field of Reinforcement Learning (RL), particularly in the domain of demonstration-regularized reinforcement learning and reinforcement learning with human feedback (RLHF). The authors have theoretically quantified how expert demonstrations can improve the sample efficiency of RL, offering new insights into the utility of expert demonstrations in RL and RLHF settings.

## Strengths:
- Theoretical Depth: The paper offers a rigorous theoretical analysis, including proofs and convergence guarantees for behavior cloning procedures under general assumptions.
- Novelty: The demonstration-regularized reinforcement learning framework and its application in RLHF are novel contributions, offering fresh perspectives on leveraging expert demonstrations in RL.
- Practical Relevance: The approach aligns with practical applications, particularly in settings where expert demonstrations are available, making the findings relevant for real-world RL applications.

## Weaknesses:
- Presentation and Clarity: Multiple reviewers pointed out issues with the presentation, particularly the heavy reliance on appendices and lack of clarity in explaining proof techniques and algorithmic contributions in the main text.
- Assumptions and Limitations: The assumption of near-optimal expert demonstrations and the lack of empirical validation were noted as limitations, although these are somewhat mitigated by the theoretical nature of the work.

## Reviewer Consensus:
While there were initial concerns regarding the presentation and the assumptions made in the paper, the authors addressed most of these through revisions and clarifications. The reviewers appreciated the theoretical contributions and the clarity improvements made in response to feedback. The consensus leans towards acceptance, recognizing the paper’s contribution to the theoretical understanding of RL with expert demonstrations.

**Justification For Why Not Higher Score:**

The decision to accept the paper is based on its strong theoretical contributions and the successful addressal of concerns raised during the review process. However, it is recommended that future work should aim to relax some of the assumptions, particularly regarding the optimality of expert demonstrations, and include empirical validations to strengthen the practical applicability of the proposed methods.

**Justification For Why Not Lower Score:**

The paper has sufficient theoretical contribution. The authors have also addressed the reviewers’ concerns.

---

### Decision · Program_Chairs · 2024-01-16

Accept (poster)